# Simulation and sensitivity analysis for cloud and precipitation measurements via spaceborne millimeter wave radar

Leilei Kou[1], Zhengjian Lin[2], Haiyang Gao[1], Shujun Liao[2], Piman Ding[3]

[1] Collaborative Innovation Center on Forecast and Evaluation of Meteorological Disasters, Key Laboratory for Aerosol-Cloud-Precipitation of China Meteorological Administration, Nanjing University of Information Science and Technology, Nanjing 210044, China
[2] School of Atmospheric Physics, Nanjing University of Information Science and Technology, Nanjing 210044, China
[3] Shanghai Institute of Satellite Engineering, Shanghai 201109, China

*Correspondence to*: Leilei Kou (cassie320@163.com)

**Abstract.** This study presents a simulation framework for cloud and precipitation measurements via spaceborne millimeter wave radar composed of eight sub modules. To demonstrate the influence of the assumed physical parameters and improve the microphysical modeling of the hydrometeors, we first conducted a sensitivity analysis. The results indicated that the radar reflectivity was highly sensitive to the particle size distribution (PSD) parameter of the median volume diameter and particle density parameter, which can cause reflectivity variations of several to more than 10 dB. The variation in the prefactor of the mass-power relations that related to riming degree may result in an uncertainty of approximately 30–45 %. The particle shape and orientation also had a significant impact on the radar reflectivity. The spherical assumption may result in an average overestimation of the reflectivity by approximately 4–14 %, dependent on the particle type, shape, and orientation. Typical weather cases were simulated using improved physical modeling accounting for the particle shapes, typical PSD parameters corresponding to the cloud precipitation types, mass-power relations for snow and graupel, and melting modeling. We present and validate the simulation results for a cold front stratiform cloud and a deep convective process with observations from W-band cloud profiling radar (CPR) on the CloudSat satellite. The simulated bright band features, echo structure, and intensity showed good agreement with the CloudSat observations; the average relative error of radar reflectivity in the vertical profile was within 20 %. Our results quantify the uncertainty in the millimeter wave radar echo simulation that may be caused by the physical model parameters and provide a scientific basis for optimal forward modeling. They also provide suggestions for prior physical parameter constraints for the retrieval of the microphysical properties of clouds and precipitation.

## 1 Introduction

The development of clouds and precipitation is the result of interactions among dynamic, thermodynamic, and microphysical processes. The vertical structure of clouds is closely related to the characteristics of cloud radiation, as well as the physical process, mechanism, and efficiency of precipitation. Measurements of the three-dimensional (3D) structure and global distribution of cloud precipitation, as well as an understanding of the microphysical characteristics and transformation of

cloud precipitation, are the key factors affecting the accuracy of weather forecasting and climate models (Kollias et al., 2007; Li et al., 2013; Luo et al., 2008; Stephens et al., 2002).

Cloud radars are mainly operated spaceborne, airborne or ground based. Among them, spaceborne radar plays an important role in global cloud precipitation measurements owing to its strong penetration, high precision, and wide coverage. The most widely used spaceborne cloud radar is the millimeter wave cloud profiling radar (CPR) carried onboard the CloudSat satellite (Stephens et al., 2008; Tanelli et al., 2008). The CPR is a W-band, nadir-pointing radar system, with a minimum detectable signal of about -29 dBZ. The CPR footprint size is 1.4 km across-track and 2.5 km along-track, and the vertical resolution is approximately 500 m (Stephens et al., 2008). Since its launch, CloudSat CPR has obtained a large quantity of cloud vertical profile data, and has been widely used in cloud physics, weather, environment, climatology, and other fields (Dodson et al., 2018; Stephens et al., 2018; Battaglia et al., 2020). Spaceborne millimeter wave radar can not only detect the vertical structure of various cloud systems, but also measure the distribution of snow, light rain, and even moderate rain (Haynes et al., 2009). This provides an opportunity to advance the understanding of the way water cycles through the atmosphere, by jointly observing clouds and associated precipitation (Behrangi et al., 2013; Ellis et al., 2009; Hayden et al., 2018).

Recently, many countries have begun research on next-generation spaceborne cloud radar (Battaglia et al., 2020; Illingworth et al., 2015; Tanelli et al., 2018; Wu et al., 2018), such as the CPR on the EarthCARE satellite and dual-frequency cloud radar on the Aerosol/Clouds/Ecosystem (ACE) mission (Illingworth et al., 2015; Tanelli et al., 2018). Forward modeling and simulation play an important role in the design of the observation system and interpretation of cloud and precipitation observation data (Horie et al., 2012; Lamer et al., 2021; Leinonen et al., 2015; Marra et al., 2013; Sassen et al., 2007; Wang et al., 2019; Wu et al., 2011). QuickBeam is a user-friendly radar simulation package that converts modeled clouds to the equivalent radar reflectivities measured by a wide range of meteorological radar(Haynes et al., 2007). The Satellite Data Simulator Unit (SDSU) developed by Nagoya University, Japan, is a satellite multisensor simulator integrating radar, microwave radiometer, and visible/infrared imager. The Goddard Satellite Data Simulator Unit (G-SDSU) is a derivative version of the SDSU (Masunaga et al., 2010). In addition to the basic functions of the SDSU, it can be coupled with high-precision NASA atmospheric models, such as the Weather Research and Forecasting-Spectral Bin Microphysics (WRF-SBM) (Iguchi et al., 2012). The Global Precipitation Measurement (GPM) satellite simulator is also based on the G-SDSU, which converts the geophysical parameters simulated by the WRF-SBM into observable microwave brightness and equivalent reflectivity factor signals of the GPM (Matsui et al., 2013). The particle shape, composition, orientation, and mass relation all affect the scattering characteristics, and then influence the radar reflectivity simulation results. The radar reflectivity for the W-band is also sensitive to microphysical parameters like the particle size distribution (PSD) model and parameter, particle shape, orientation, and mass (Mason et al., 2019; Nowell et al., 2013; Sy et al., 2020; Wood et al., 2013; Wood et al., 2015). A sensitivity analysis is essential for estimating the effects of these uncertainties on simulated radar reflectivity, and guiding appropriate parameter setting in forward modeling.

China has also begun its own spaceborne millimeter wave radar project. The National Satellite Meteorological Center plans to launch a cloud-detecting satellite, whose main load will be the cloud profiling radar (Wu et al., 2018). For development of spaceborne cloud radar, simulation research on cloud and precipitation detection can provide important theoretical support for the design and performance analysis of the system.

In this study, we quantify the uncertainty of different physical model parameters for hydrometeors contributing to radar reflectivity uncertainty via a sensitivity analysis, and present radar reflectivity simulations with optimal parameter settings, based on forward modeling for spaceborne millimeter wave (94 GHz, W-band) radar. Sensitivity analyses of typical cloud parameters on the radar equivalent reflectivity factors were carried out. Parameters included the particle size distribution (PSD) parameters, PSD model, particle density parameters, shape, and orientation. Using appropriate physical parameter settings, we present and compare the simulation results of two typical cloud precipitation scenarios with measured CloudSat results. Based on a sensitivity analysis of typical cloud parameters, and a demonstration of cloud precipitation cases, we show the radar reflectivity uncertainty caused by the physical modeling of hydrometeors while emphasizing the importance of assuming more realistic scattering characteristics, as well as appropriate density relations and PSD parameters corresponding to different cloud precipitation types.

## 2 Modeling

### 2.1 Overview

The framework of forward modeling and simulation for spaceborne millimeter radar was composed of eight sub modules: cloud precipitation scene simulation with Weather Research and Forecasting (WRF) model (Skamarock et al., 2019), WRF output result verification, hydrometeor microphysical characteristics modeling, particle scattering and attenuation characteristics calculations, atmospheric radiation transmission calculation, output radar echo through coupling with platform and instrument parameters, sensitivity analysis, and comparisons and analyses of the result. Figure 1 shows the logic structure between each sub module. The key points of each sub module are described as follows.

1) From CloudSat historical data and typical weather processes we obtained the cloud precipitation scene cases. According to the occurrence area and time, the corresponding National Center of Environmental Prediction Final (NCEP FNL) reanalysis data were obtained as the initial field in the WRF model.

2) The WRF model was used to simulate the distribution of all types of hydrometeors in these cases. In this research, we use version 4.1.2 of the advanced research WRF model (Skamarock et al., 2019). The WRF simulation results were then validated by using the real satellite and ground observation data such as ground-based radar data.

3) Based on the hydrometeor mixing ratio of the WRF output and assuming certain microphysical parameters based on empirical information obtained from a large amount of observation data, the PSD of the hydrometeor particles were modeled.

4) The complex refractive index of different hydrometeors was calculated according to the particle phase and temperature. The scattering and attenuation characteristics of the hydrometeor particles were then calculated using the T-matrix method (Mishchenko and Travis, 1998). Meanwhile, the absorption coefficients of the atmospheric molecules, such as the water vapor and oxygen, were calculated based on the Liebe attenuation model (Liebe, 1981).

5) The radar reflectivity factor was then calculated based on the atmospheric radiation transmission process and the scattering and attenuation coefficients of hydrometeors.

6) Through coupling with the instrument and platform parameters, the radar echo signal was calculated using the radar equation.

7) During the simulation process, the sensitivity analysis of typical cloud physical parameters was performed to guide the
optimal microphysical modeling of the hydrometeors.

8) Finally, the simulation results were compared with observation data, such as CloudSat data, to validate the forward simulations.

## 2.2 Hydrometeor microphysical modeling

The radar reflectivity factor depends on the size, shape, orientation, density, size distributions, and dielectric constants for
the hydrometeor particles. The microphysical characteristics of each hydrometeor are substantially different, which affects the scattering properties and further the radar echo. The following introduces the microphysical modeling of the different hydrometeors.

The complex refractive index of each hydrometeor was first calculated, which depends on its phase, composition, density, and radar wavelength. For pure water and pure ice, such as raindrops, cloud water, and cloud ice, we calculated the
refractive index according to Ray (1972). Dry snow and graupel are a mixture of air and ice, while wet snow and graupel are a mixture of air, ice, and water. The densities of air, ice, and water are generally 0.001, 0.917, and 1 g/cm$^3$, respectively. The mixture has different densities according to the proportions of different components. Given the proportion of air, ice, and water (or riming fraction or melting fraction) in the hydrometeor, the refractive index of the mixture can be calculated using the Maxwell-Garnett mixing formula (Ryzhkov et al., 2011).

### 2.2.1 Cloud water

Cloud water droplets form from the condensation of super-saturated water vapor onto cloud condensation nuclei. They are usually spherical due to surface tension, with a typical size of ~10 $\mu m$ (Mason, 1971; Miles et al., 2000). As the size of cloud droplets is small relative to the wavelength, with an approximately spherical shape, their scattering characteristics can usually be calculated via Mie theory (Bohren and Huffman, 1983) or Rayleigh approximation (Zhang, 2017) based on the
sphere assumption. The PSD of cloud water can generally be modeled with a normalized Gamma distribution (Bringi and Chandrasekar, 2001; Chase et al., 2020):

$$N(D)dD = N_w \frac{6}{3.67^4} \frac{(3.67+\mu)^{\mu+4}}{\Gamma(\mu+4)} \left(\frac{D}{D_0}\right)^{\mu} \exp\left[-(3.67+\mu)\frac{D}{D_0}\right]dD \;, \tag{1}$$

$$N_w = \frac{W}{\pi\rho_w}\left(\frac{4(3.67+\mu)}{(4+\mu)D_0}\right)^4 \;, \tag{2}$$

where $N(D)$ is the particle size distribution, $D$ is the volume equivalent diameter, $N_w$ is the normalized intercept parameter, $D_0$ is the median volume diameter, $\rho_w$ is the density of water, i.e. 1 g/cm$^3$, $\mu$ is the shape parameter, and $\Gamma$ is the gamma function. Uniform bin sizes are set for hydrometeors, for example, $dD$ is 0.01 mm for cloud water.

Here, $W$ in Eq. (2) is the water content of the cloud water, which is calculated by converting the mixing ratio of the hydrometeor from the WRF output:

$$W = \frac{P}{R_{gas}T_V}*1000*q \;, \tag{3}$$

where $R_{gas}$ is the specific gas constant, $P$ is the air pressure in hPa, $T_V$ is the virtual temperature in $K$, $q$ is the mixing ratio of the hydrometeor based on the WRF output in kg/kg, and the units of $W$ are g/m$^3$. As $W$ is the output of the WRF model, the PSD of the gamma distribution was mainly determined by two parameters, i.e., $D_0$ and $\mu$. According to Miles et al. (2000) and Yin et al. (2011), we simulated the PSD with $D_0$ and $\mu$ ranging from 0.005–0.05 mm and 0–4, respectively.

**2.2.2 Rain**

Owing to the effects of surface tension, aerodynamic force, and hydrostatic gradient force, raindrops often take the shape of an oblate spheroid (horizontal axis ($a_0$) > vertical axis ($b_0$)), with an increase in the size of the raindrop. Here, we used the axis ratio model proposed by Brandes (2002):

$$\gamma_w = \frac{b_0}{a_0} = 0.9951+0.0251D-0.03644D^2+0.005303D^3-0.0002492D^4 \;, \tag{4}$$

where $D$ is the equivolume diameter. The scattering and attenuation characteristics of raindrops were calculated using the T-matrix method. Considering the influence of aerodynamics on the particle orientation, the canting angle of raindrops was assumed to follow a Gaussian distribution with a mean value of 0° and a standard deviation (SD) of 7° (Zhang, 2017).

The PSD of raindrops was still modeled as the Gamma distribution shown in Eqs. (1) and (2), where $W$ was calculated based on the rain mixing ratio from the WRF output. According to Bringi and Chandrasekar (2001), $D_0$ and $\mu$ were uniformly distributed in ranges of 0.5–2.5 mm and –1 to 4, respectively.

**2.2.3 Cloud ice**

Cloud ice is mainly composed of various non-spherical ice crystals; the size and shape of ice crystal particles are complex and diverse, depending on the cloud temperature, the degree of supersaturation in the environment where the particle forms

and grows, and whether the particles have experienced aggregation processes in the cloud (Heymsfield et al., 2013; Ryzhkov and Zrnic, 2019). The database in Liu (2008) can be used to examine the scattering characteristics of ice crystals with different shapes. Here, we used the T-matrix (Mishchenko and Travis, 1998) to calculate the scattering properties of ice crystals, which were assumed to be either spheroids or circular cylinders. The spheroids were treated as horizontally aligned oblate spheroids with an axial ratio of 0.6 (Hogan et al., 2012); the relation between the larger and smaller dimension of the cylinders was as follows (Fu, 1996):

$$
\begin{cases} L/h = 5.068L^{0.586} & L > 0.2\,mm \\ L/h = 2 & L \le 0.2\,mm \end{cases},
\tag{5}
$$

Distribution of orientations of ice particles depends on their falling behavior. According to Melnikov and Straka (2013), we assume that the ice crystal orientations follow a Gaussian distribution, with a mean canting angle of 0° and a SD between 2° and 20°.

The PSD of cloud ice is mainly represented as an exponential or Gamma distribution (Ryzhkov and Zrnic, 2019). Here, the normalized Gamma distribution was adopted according to the empirical fits derived in Heymsfield et al (2013). The relation between the number concentration, $N_W$, and $D_0$ is as follows:

$$
N_w = \frac{W}{\pi \rho_i} \left( \frac{4(3.67 + \mu)}{(4 + \mu)D_0} \right)^4,
\tag{6}
$$

where $\rho_i$ is 0.917 g/cm³ and $W$ is the water content of cloud ice from the WRF output.

According to Heymsfield et al. (2013), the total number concentration, $N_t$, is a function of the temperature, $T$:

$$
N_t = \begin{cases} 2.7 \times 10^4 & T \le -60\,^oC \\ 3.304 \times 10^3 \exp(-0.04607T) & T > -60\,^oC \end{cases},
\tag{7}
$$

The maximum diameter, $D_{max}$, is also dependent on $T$:

$$
D_{max} = \begin{cases} 11\exp(0.069T) & stratiform \\ 21\exp(0.070T) & convective \end{cases},
\tag{8}
$$

where $T$ is in ℃, $N_t$ is in m⁻³, and $D_{max}$ is in mm. Given $T$ and the water content of cloud ice, $W$, as well as the empirical value of $\mu$, we can calculate $D_0$ from Eqs. (1), (6)– (8) and the following formula:

$$
N_t = \int_0^{D_{max}} N(D)dD,
\tag{9}
$$

Owing to the monotonicity of the functions, $D_0$ can be solved numerically. For cloud ice, $\mu$ usually ranges from 0 to 2 (Tinel et al., 2005; Yin et al., 2011).

### 2.2.4 Snow

Snowflakes are usually formed by the aggregation and growth of ice crystals. Although the shapes of snowflakes are irregular, they can also be modeled as spheroids, with a constant axis ratio of 0.75 (Zhang, 2017). For large snow aggregates, an axis ratio of approximately 0.6 is regarded as a good model especially for explaining multifrequency radar observations (Matrosov, 2007; Moisseev et al., 2017). As snowflakes fall with their major axis mainly aligned in the horizontal direction, the mean canting angle of snow is assumed to be $0°$ and the SD of the canting angle is assumed to be $20°$ (Zhang, 2017). The width of the canting angle distribution grows with an increase in aggregation. Garrett et al. (2015) showed that the average SD of moderate-to-heavy snow, consisting of dry aggregates, is approximately 40º.

The PSD of snow is modeled as an exponential distribution; the distribution parameters are constrained by the mass-power function relationship (Kneifel et al., 2011; Lin et al., 2011; Matrosov et al. 2007; Tomita, 2008; Woods, 2008):

$$N(D)dD = N_0 \exp(-\Lambda D)dD ,\tag{10}$$

$$m(D) = aD^b, \quad \text{or} \quad \rho_s(D) = \frac{6}{\pi} aD^{b-3} ,\tag{11}$$

$$\Lambda = \left[ \frac{aN_0\Gamma(b+1)}{W} \right]^{\frac{1}{b+1}} ,\tag{12}$$

where $N_0$ is the intercept parameter (usually ranging $10^3$–$10^5$ mm$^{-1}$ m$^{-3}$), $D$ is the volume equivalent diameter, and $m(D)$ and $\rho_s(D)$ are the mass and density of the particle, respectively.

Constants $a$ and $b$ strongly depend on the snow habit and microphysical processes that determine snow growth and are usually determined experimentally. The exponent value of $b$ is generally a Gaussian distribution, with a mean of 2.1 (Brandes et al., 2007; Heymsfield et al., 2010; Szyrmer and Zawadzki, 2010; von Lerber et al., 2017). The prefactor $a$ can vary considerably, and the value of $a$ increases with the aggregate density or riming degree (Huang et al., 2019; Ryzhkov and Zrnic, 2019; Sy et al., 2020; Wood et al., 2015). Most of the mass and density relations in previous studies (Brandes et al., 2007; Sy et al., 2020; Szyrmer and Zawadzki, 2010; Tiira et al., 2016) showed that the prefactor $a$ varies between 0.005 and 0.014 cgs units (i.e., in g/cm$^b$), where $D$ and $m$ are in centimeters and grams; the mean value is approximately 0.009. In different studies, the statistical results of mass-size relations vary slightly (Brandes et al., 2007; Mason et al., 2018; Tiira et al., 2016; Wood et al., 2015), with the primary difference being the diameter expression for the maximum dimension diameter, $D_m$, median volume diameter, $D_0$, or volume equivalent diameter, $D$. In this study, the diameters in the mass and density relations were converted to the volume equivalent diameter $D$ according to the assumed axis ratio.

### 2.2.5 Graupel

Graupel is generated in convective clouds by the accretion of supercooled liquid droplets on ice particles or by the freezing of supercooled raindrops lofted in updrafts. The density of graupel varies substantially depending on their formation

mechanism, time of growth from the initial embryo, liquid water content, and ambient temperature. The density is generally between 0.2 and 0.9 g/cm$^3$, with the typical value of approximately 0.4 g/cm$^3$ from the statistical results in observation experiments (Heymsfield et al., 2018; Ryzhkov and Zrnic, 2019).

Generally, graupel particles have irregular shapes. Here the shape of graupel was modeled as a spheroid, where the axis ratio for dry graupel was set to a constant value of 0.8, and the axis ratio for melting graupel was modeled according to Ryzhkov et al. (2011) as:

$$
\begin{aligned}
\gamma_g &= 0.8 \quad f_w \leq 0.2 \\
\gamma_g &= 0.88 - 0.4 f_w \quad 0.2 < f_w < 0.8 \\
\gamma_g &= 2.8 - 4\gamma_w + 5(\gamma_w - 0.56) f_w \quad f_w \geq 0.8
\end{aligned}
\quad , \tag{13}
$$

where $\gamma_w$ is the axis ratio of raindrops, and $f_w$ is the mass water fraction. The SD of the canting angle, $\delta$, was parameterized as a function of $f_w$:

$$
\delta = 60^o (1 - c f_w), \tag{14}
$$

where $c$ is an adjustment coefficient, set usually as 0.8 (Jung et al., 2008).

The PSD of graupel is assumed to be an exponential distribution, as shown in Eqs. (10)–(12). In convective clouds, a large part of graupel likely develops via collisions between frozen drops and smaller droplets, and its bulk density decreases with increasing graupel size (Khain and Pinsky, 2018). Similar mass relations can be found for graupel, and its exponent $b$ is larger than that for snow. The exponent for low-density graupel is approximately 2.3 (Erfani and Mitchell, 2017; von Lerber et al., 2017) while that for lump graupel approaches 3.0 (Mace and Benson, 2017; Mason et al., 2019). The mean value of $b$ is approximately 2.6, and prefactor $a$ varies mainly between 0.02 and 0.06 g/cm$^b$ (Mason et al., 2018; Heymsfield et al., 2018), where the units for $m$ and $D$ are grams and centimeters.

### 2.2.6 Melting modeling

Neglecting aggregation, collision-coalescence, evaporation, and the small amount of water that may collect on the particle owing to vapor diffusion, we assume that the mass of snow was conserved during the evolution process from dry snow, to wet snow to liquid water:

$$
\rho_w D_w^3 = \rho_{ms} D_{ms}^3 = \rho_s D_s^3, \tag{15}
$$

where $\rho_w$, $\rho_m$ and $\rho_s$ are the densities of the liquid water, melting and dry particles, respectively; $D_w$, $D_{ms}$ and $D_s$ are the diameters of water, melting snow, and dry snow, respectively.

If the mass fraction of melt water in the particle of $f_w$ is known, the density of melting snow can be obtained as follows (Haynes et al., 2009):

$$
\rho_{ms} = \frac{\rho_s \rho_w}{f_w \rho_s + (1 - f_w) \rho_w}, \tag{16}
$$

The density of snowflakes follows the power-law relation in Eq. (11). The density parameter in Eq. (11) can be obtained

according to the density-diameter relationship, where the density is calculated from Eq. (16) with an assumed $f_w$ value. Dielectric constant of melting snow depends on snow density and water fraction $f_w$. Here, we use the model that water is considered as background and snow is treated as inclusions, and compute the dielectric constant based on Maxwell-Garnett formulas for the mixture of snow and water (Ryzhkov et al., 2011; Zhang, 2017).

According to Eqs. (11) and (15), the relation between the particle diameters can be obtained as follows:

$$
\quad D_w = \left(\frac{6}{\pi}a\right)^{\frac{1}{3}} D_{ms}^{\frac{b}{3}}, \tag{17}
$$

where the equivalent-mass melted diameter $D_{ms}$ corresponding to diameter $D_s$ of each dry snow particle is calculated from Eq. (15).

Due to melting, the uniform bin size set no longer applies, such that a new bin size must be calculated. The bin size for rain ($dD_w$) can be obtained by differentiating as follows

$$
\quad dD_w = \frac{b}{3}\left(\frac{6}{\pi}a\right)^{\frac{1}{3}} D_{ms}^{\frac{b-3}{3}} \cdot dD_{ms}, \tag{18}
$$

According to the mass conservation model, the total liquid water content of a distribution is conserved. The number concentration of raindrops ($N_w$) in each size is calculated as follows

$$
N_w\left(D_w\right) = N_{ms}(D_{ms})\frac{3}{b}\left(\frac{6}{\pi}a\right)^{-\frac{1}{3}} D_{ms}^{\frac{3-b}{3}}, \tag{19}
$$

where $N_{ms}(D_{ms})$ is the number concentration of melting particles.

The scattering characteristics of melting particles are still calculated by T-matrix. It is assumed that the shape of melted ice particles gradually changes with the increase of mass water fraction $f_w$, so as to finally obtain the shape of raindrops with the same mass. We can introduce the axis ratio ($\gamma_{ms}$) relationship and the relationship of SD of the canting angle ($\delta_{mr}$) for melting particles as (Ryzhkov and Zrnic, 2019):

$$
\begin{aligned}
\gamma_{ms} &= \gamma_s + f_w(\gamma_w - \gamma_s) \\
\delta_{ms} &= \delta_s + f_w(\delta_r - \delta_s)
\end{aligned}, \tag{20}
$$

where $\gamma_s$ is the axis ratio of dry snow, $\gamma_w$ is the axis ratio of raindrop of diameter which is produced as a result of snow melting, $\delta_r$ is the SD of the canting angle distribution of raindrops, whereas $\delta_s$ is the corresponding SD of the distribution of dry snow.

**2.3 Radar equation**

The signal power, $P_r$, received by the radar was calculated using the radar equation:

$$P_r = C\frac{P_t}{r_0^2}Z_e \exp\left[-2\int_0^{r_0} k(r)dr\right],$$ (21)

where $P_t$ is the transmitted power, $r_0$ is the range to the atmospheric target, $C$ is the radar constant related to the instruments, and $k$ is the attenuation coefficient. The radar equivalent reflectivity factor, $Z_e$, was calculated from the scattering characteristics and the assumed PSD of the various hydrometeors:

$$Z_e = \frac{\lambda^4}{\pi^5 |K_w|^2}\int_0^\infty N(D)\sigma_b(D)dD,$$ (22)

where $\sigma_b(D)$ is the backscattering cross section of the particle with a diameter $D$, $\lambda$ is the radar wavelength, and $K_w = (n_w^2 - 1)/(n_w^2 + 2)$, where $n_w$ is the complex refractive index of water for a given wavelength and temperature.

For spaceborne millimeter wave radar, the equivalent radar reflectivity factor (hereafter, radar reflectivity) observed by the radar is the attenuated radar reflectivity factor, $Z_{e0}$:

$$Z_{e0} = Z_e \exp\left[-2\int_0^{r_0} k(r)dr\right], \quad k = 10^{-3}\int Q_t(D)N(D)dD,$$ (23)

where the units of $k$ are 1/km, $Q_t$ (mm$^2$) is the extinction cross-section of the corresponding hydrometeor calculated by the T-matrix, the units of $N(D)$ are m$^{-3}$ mm$^{-1}$, and the unit of $dD$ is mm. During radar reflectivity calculation, a look-up table of backscattering and extinction cross-sections is established for reducing the calculation workload.

If there are many types of hydrometeors at the same height, the equivalent unattenuated radar reflectivity and attenuation coefficient of each hydrometeor is calculated based on the look-up table. Then, the total unattenuated radar
reflectivity at this height is obtained by adding all types of hydrometeors, and the two-way attenuation is obtained by integrating the total attenuation coefficient with path. The attenuated radar reflectivity is obtained by subtracting the attenuation from the unattenuated radar reflectivity. Considering the difference between the resolution of the simulation data and the observation resolution of the instrument, the convolution of the simulation echo and antenna pattern were also performed during the coupling process of the simulation data and instrument parameters. During this process, the antenna
pattern was set as a two-dimensional Gaussian distribution.

After coupling with the antenna pattern, the final radar reflectivity was obtained. Here, the unit of $Z_e$ is mm$^6$/m$^3$, and it is usually expressed in decibel form as $dBZ_e = 10 * \log_{10}(Z_e)$.

## 3 Sensitivity analysis

Due to complex microphysical processes in cloud precipitation, the PSDs of hydrometeors vary substantially. An accurate
PSD is difficult to measure, especially for aloft particles. The phase, size, and shape of particles also change with the dominating microphysical processes and external environment, which all affect the simulation results. For optimizing the parameter settings of the forward modeling and more accurately interpreting the radar reflectivity results, we performed a

series of sensitivity analyses of cloud parameters. Here, we mainly focused on the scattering effects; the attenuation effects will be discussed in a follow-up study.

## 3.1 PSD parameters

The Gamma distribution is determined by three parameters. As one of the parameters is obtained from the water content, $W$, of the hydrometeor in the WRF output, we mainly considered the effects of $D_0$ and $\mu$ on the radar reflectivity. Figure 2 shows the radar reflectivity change with variations in the gamma PSD parameters for cloud water and rain. Cloud water particles are small compared to the radar wavelength, which is in the linear growth stage in the Mie scattering region. With a five-fold increase in $D_0$ ($W$ remains constant), e.g., increasing from 10 to 50 $\mu$m, the reflectivity increases by approximately 20 dB. For rain particles, the impact of $D_0$ is not as significant as that of cloud water: a five-fold change in $D_0$ can lead to a reflectivity change within 5 dB. Owing to the Mie scattering effect on raindrops, the contribution from relatively small raindrops may be more than that from larger raindrops considering the influence of the number concentration. In the gamma PSD, the effect of $\mu$ is relatively small; the reflectivity change caused by $\mu$ is within 1.5 dB when using a constant $D_0$.

For cloud ice, $D_0$ is calculated from Eqs. (6)-(9) given $W$ and $T$; $\mu$ is the only parameter that needs to be assumed. Figure 3a and 3b show the reflectivity change with $W$ and $\mu$, where Fig. 3a was obtained when $T$ was –20 ℃ and Fig. 3b was obtained when $T$ was –60 ℃. As the PSD of cloud ice was constrained by the total number concentration, $D_0$ and $\mu$ are interrelated and $D_0$ increases with an increase in $\mu$, $W$, and $T$. Based on Fig. 3a and b, we observed that when $\mu$ varies from 0 to 2, the maximum reflectivity change is approximately 4 dB at –20 ℃ while that at –60 ℃ is approximately 5 dB. The reflectivity change was still affected by the $D_0$ variation. Based on Eqs. (6)-(9), $D_0$ varied from 0.1–0.5 mm at –60 ℃ and 0.2–0.8 mm at –20 ℃ when $W$ ranged from 0 to 0.5 g/m$^3$. Figure 3c and d show the reflectivity change caused by $D_0$ and $\mu$ under a conventional gamma PSD without constraints on the total number concentration. In the conventional gamma PSD, the $D_0$ and $\mu$ vary independently; the reflectivity can change by 13 dB when $D_0$ varies from 0.2 to 0.8 mm. The results showed that the effect of PSD parameter variation on the reflectivity can be reduced by approximately 60 % owing to constraints on the total number concentration for the PSD of cloud ice.

An exponential PSD with a power-law mass spectrum was used for snow and graupel. Figure 4 shows the effects of intercept parameter $N_0$ and the mass power-law parameters of prefactor $a$ and exponent $b$. With the mean mass-size relationships for snow and graupel, changing the $\log_{10}(N_0)$ from 3 to 5 could cause a reflectivity increase of approximately 7–8 dB, as shown in Fig. 4a and d.

The mass power-law parameters vary with snow/graupel type, shape, and porosity. In Fig. 4b and e, we see that with a constant $N_0$ and mean value of exponent $b$, the reflectivity change caused by variation in prefactor $a$ from 0.005 to 0.013 g/cm$^b$ for snow and 0.02 to 0.06 g/cm$^b$ for graupel ($W$ remains constant) can reach 7–10 dB. An increase in $a$ leads to an obvious increase in the corresponding particle scattering properties, and then causes the reflectivity change. Using an

average mass-power relation assumption, the variation in *a* as a result of the degree of aggregation and riming, and particle
shapes may result in the reflectivity uncertainty of approximately 45 % and 30 % for snow and graupel, respectively. For
analyzing the effect of the variation in *b*, a Gaussian distribution of *b* was modeled. According to results from observation
experiments reported in the literatures, the exponent *b* for snow varies from 1.4 to 2.8, and. we derived the mean value of *b*
to be close to 2.1 via averaging literature values of *b* from list of studies (Brandes et al., 2007; Heymsfield et al., 2010;
Huang et al., 2019; Sy et al., 2020; Szyrmer and Zawadzki, 2010; Tiira et al., 2016; Wood et al., 2013). For graupel, the
325 exponent *b* varies from 2.1 to 3, and a mean value of approximately 2.6 was derived from the studies in the literatures
(Heymsfield et al., 2018; Mason et al., 2018; Von Lerber et al., 2017). Based on the range and mean value of *b* for the
Gaussian distribution, we calculated the standard deviation (SD) to be 0.28 and 0.16 for snow and graupel, respectively. The
error bars in Fig. 4c and f represent the SD of the reflectivity change caused by variation in *b*, which was approximately 2 dB
for snow and 0.5 dB for graupel. The results showed that the sensitivity of reflectivity to prefactor *a* was substantially greater
than that of exponent *b*.

In all, the mass relationships that depend on particle habits and formation mechanisms, cause substantial uncertainties in
W-band radar reflectivity. Our results are consistent with the sensitivity analysis by Wood and L'Ecuyer (2021) who pointed
out that the W-band radar reflectivity uncertainty for snowfall was dominated by the particle model parameter (e.g., the
prefactors and exponents of the mass relationships). The mass relationship can cause the reflectivity uncertainty of several to
335 more than 10 dB. The results indicate that improved constraints on assumed particle mass models would improve forward-
modeled radar reflectivity and physical parameter retrieval.

## 3.2 PSD models

The PSDs of hydrometeors can usually be represented by different distributions, such as the Gamma distribution and
lognormal distribution, which are frequently used in cloud water PSDs. This section discusses the influence that the selection
of different PSD models has on radar reflectivity factor, taking cloud water as an example. Figure 5a shows two PSD models
of cloud water, in which the black solid line represents the Gamma distribution, and the red-dotted line represents the
lognormal distribution. The lognormal distribution uses the following formula (Miles et al., 2000):

$$N(D)dD = \frac{6W}{\pi\sqrt{2\pi}\rho_p\sigma D_m^3}\exp\left(-\frac{9}{2}\sigma^2\right)\exp\left[-\frac{\left(\ln D - \ln D_m\right)^2}{2\sigma^2}\right]\frac{dD}{D}, \tag{24}$$

where $D_m$ is the mass weighted diameter, $\sigma$ is the dispersion parameter.
The parameters in the PSD model in Fig. 5a are based on the parameter settings for cloud water in terrestrial stratiform
clouds (Mason, 1971; Miles et al., 2000; Niu and He, 1995), where $D_0$ is 20 μm, $\mu$ is 2 in the Gamma distribution, $D_m$ is 20
μm, $\sigma$ is 0.35 in the log-normal distribution, and $W$ in both PSD models are set to 1g/m³. The black solid line represents for
the Gamma distribution, and the red dotted line represents the log-normal model. Corresponding to the typical parameter
settings of the Gamma and log-normal distributions, the difference between the two PSDs was notable; the reflectivity

change caused by the different PSD models was approximately 4.5 dB. This result showed that the PSD model had a certain impact on echo simulation, and it was necessary to carefully select the PSD model and set the parameters according to the type of cloud and precipitation.

## 3.3 Particle shape and orientation

The scattering properties of particles are sensitive to the hydrometeor shape and orientation. Previous studies (Marra et al.,
2013; Masunaga et al., 2010; Seto et al., 2021; Wang et al., 2019) often assume that the hydrometeor particle is a sphere, but most particles are non-spherical. This section discusses the influence that cloud ice, snow, graupel, and rain particle shapes (cloud water is generally spherical) have on radar reflectivity.

Figure 6 compares the backscattering cross-section and corresponding radar reflectivity under different shapes of cloud ice, dry snow, and rain. Three shape types, i.e., sphere, spheroid, and cylinder, for cloud ice were considered, where the
shape parameter setting refers to section 2.2.3. The solid and dotted lines in Fig. 6a indicate that the SD of the canting angle ($\delta$) is 2° and 20°, respectively. The backscattering difference for cloud ice was evident between the sphere and non-sphere when the diameter was greater than 1 mm. The radar reflectivity factor in Fig. 6b was obtained with the constrained PSD parameter (section 2.2.3) of $T = -60$ ℃ and $\mu=1$, and the maximum diameter was calculated according to Eq. (8) that was within 0.4 mm. Figure 6b shows that the spherical and non-spherical assumption for cloud ice may result in an average
reflectivity difference by approximately 8 %. The reflectivity difference caused by $\delta$ was approximately 1 %. Figure 6c shows the backscattering cross-section of dry snow with a mass-diameter relation of $m=0.0075D^{2.05}$ (Matrosov et al., 2007; Moisseev et al., 2017), where the axis ratio of the spheroid was 0.6 and the SD of the canting angle was assumed to be 20° and 40°, respectively. When calculating the radar reflectivity factor, the corresponding exponential distribution parameter was $N_0 = 3 \times 10^3\,\text{m}^{-3}\,\text{mm}^{-1}$ and the reflectivity difference between the sphere and spheroid can reach approximately 1.6 dB.
In particular, the average reflectivity difference reached 14 % for a $\delta$ of 20° and 12 % for a $\delta$ of 40°. For raindrops, the backscattering difference became apparent after the equivalent diameter was 2 mm, as shown in Fig. 6e. The reflectivity in Fig. 6f was obtained with a Gamma PSD parameter of $D_0 = 1.25$ mm and $\mu=3$. The reflectivity difference caused by the particle shape was negligible. This is because particles less than 2 mm mostly contribute to the radar reflectivity for rain. The influence of shape on raindrops can be negligible.
The axis ratio and particle orientation change with variations in the density of snow and graupel. Figure 7 compares backscattering and corresponding radar reflectivity for graupel between spheres and spheroids at different densities and orientations. The SD of the canting angle in Fig. 7a was calculated according to Eq. (14). Here, $\delta$ was 54° at a density of 0.4 g/cm$^3$ while $\delta$ was 20° at a density of 0.8 g/cm$^3$. Based on Fig. 7a, the backscattering section difference increased with density, which may have been due to the stronger refractive index. Figure 7b shows the corresponding radar reflectivity for
particles in (a), where the PSD was assumed to be an exponential distribution with $N_0$ of $4 \times 10^3\,\text{m}^{-3}\,\text{mm}^{-1}$. The spherical assumption may cause an average overestimation of the reflectivity by approximately 6 % when the density is 0.8 g/cm$^3$ and

$\delta$ is 20º, whereas the reflectivity difference is negligible at $\delta$ of 54º and density of 0.4 g/cm$^3$. This result showed that, besides particle shape, the particle density and orientation should also be considered in the scattering simulation. Here we mainly discuss the backscattering difference between spheres and spheroids. In future research, we will consider more realistic variations in particle shapes to evaluate sensitivity of the scattering properties to hydrometeor shapes more comprehensively.

## 4 Simulation results for typical cases

Based on the sensitivity analysis of typical cloud physical parameters, we simulated the radar reflectivity of typical cloud scenes by assuming appropriate physical parameters for different hydrometeors and cloud precipitation types with the hydrometeor mixing ratio from the WRF as input. The simulation results were compared with CloudSat observation data.

Two typical weather cases of a cold front stratiform cloud and a deep convective process were shown, which were simulated with improved setting accounting for the particle shapes, melting modeling, and mass-power relations for snow and graupel. The cases were selected by combining historical CloudSat data and typical weather processes observed on the ground. For comparison, the results with conventional simulation were also shown.

### 4.1 Stratiform case

#### 4.1.1 WRF scenario simulation

From September 24 to 25, 2012, there was a large-scale low trough cold front cloud system in northwest China, which moved from the west to the east and entered Shanxi Province. The CloudSat satellite observed the stratiform cloud process from 40.67ºN, 118.22ºE to 41.56ºN, and 117.93ºE at 04:23 AM on September 25, 2012. Centered on the observation range of CloudSat, this stratiform cloud process was simulated by the WRF model. This experiment adopted a one-way scheme with a quadruple nested grid. From the inside to the outside, the horizontal resolution was 1, 3, 9, and 27 km. It is divided into 40 layers vertically and the top of the model was 50 hPa. More details about model setup can refer to Appendix A.

Figure 8a shows the simulation area for the two interior domains (d03 and d04), in which the black line is the trajectory of the CloudSat CPR. Figure 8b shows the 3-D distribution of the total hydrometeor output of the WRF corresponding to the innermost grid. The hydrometeors were cloud water, snow, cloud ice, and rain. The hydrometeors were mainly distributed below 10 km; the maximum total water content was at approximately 3 km, ~0.9 g/m$^3$.

Figure 8c–f compares the fraction of cloud cover and cloud top temperature simulated by the WRF with European Centre for Medium-Range weather Forecasts ReAnalysis 5 (ERA5) data (Hersbach et al., 2020) and Moderate Resolution Imaging Spectrometer (MODIS) observation data (Menzel et al., 2008). The level-2 cloud product of cloud top temperature from MODIS with spatial resolutions of 5 km was used. Considering the resolution of ERA5 data (0.25º) and the MODIS scanning track (2330 km), the outermost grid in the WRF simulation data was used for comparison. Figure 8c and d show

the fraction of cloud cover from the WRF model and ERA5 data, respectively. The WRF simulates the northeastern and southwestern zonal distribution of the cold front cloud system; the simulated cloud area and cloud coverage are consistent with the ERA5 data. Figure 8e and f compares the cloud top temperature from the WRF simulation and MODIS observations. Both exhibited low cloud top temperatures in the northeast and high cloud top temperatures in the south. The value, location, and distribution of cloud top temperatures simulated by the WRF were consistent with the satellite observations.

### 4.1.2 Experiment design

For comparison with CloudSat data, the two-dimensional (2-D) hydrometeor profile from the WRF model on the track matching CloudSat was selected as the input for the radar reflectivity simulation. The WRF data at 04:30 AM was selected. Owing to the uneven output height layer of the WRF, data for the WRF simulation results were interpolated in the vertical direction. The vertical grid of the interpolated data was 240 m, corresponding to the CloudSat CPR data.

Figure 9a–e shows the latitude-height cross-section of the hydrometeors in the stratiform case simulated by the WRF for cloud water, cloud ice, snow, rain, and the total hydrometeors. The vertical extent of snow is widely distributed, ranging from 3 to 10 km. Rain is mainly below 3 km, with water contents between 0.1 and 0.2 $g/m^3$. At approximately 0 ℃, the water content for cloud water, snow, and rain were large, which led to a high total water content, with a maximum of 0.57 $g/m^3$.

Besides the comparison with the CloudSat observation data, the simulation results with improved and conventional setting were compared as well. For the stratiform case, the PSD parameters were assumed based on the empirical values of land stratiform precipitation clouds (Mason, 1971; Niu and He, 1995; Yin et al., 2011), in which the $D_0$ of cloud water was set to 0.01 mm, the $D_0$ of cloud ice was 0.02 mm, and $\mu$ was set as a constant of 1. As snow in stratiform clouds were mainly unrimed particles in middle and low latitudes (Yin et al., 2017), a mass-power relation representative $m=0.0075D^{2.05}$ of unrimed snow (Moisseev et al., 2017) was used in the simulation, where $D$ was the volume equivalent diameter. During simulation with improved microphysical setting, a melting layer with a width of 1 km was assumed below 0 ℃ based on the statistical median of melting layer width in stratiform precipitation observed by radars (Liu et al., 2016; Wang et al., 2012), and the PSD parameters of the raindrops were calculated according to the melting model. For conventional setting, the melting model was not included, and the PSD parameters for raindrops were set as $D_0=1$ mm, $\mu=3$ based on the statistical average values of microphysical parameters of stratiform precipitation in eastern China (Chen et al., 2013; Wen et al., 2019).

### 4.1.3 Radar reflectivity simulation results

Figure 9f–h shows the simulated radar reflectivity with the total hydrometeors, where Fig. 9f shows the unattenuated reflectivity, Fig. 9g shows the two-way attenuation, and Fig. 9h shows attenuated reflectivity. The reflectivity above 8 km was mainly a result of weak cloud ice and dry snow, which did not exceed –5 dBZ. The radar reflectivity caused by snow increased with an increase in the water content, up to approximately 10 dBZ. Melting led to an increase in the refractive

index and density of snow, which resulted in a sharp increase in the radar reflectivity. The unattenuated radar reflectivity in the melting layer was equivalent to the reflectivity in the rain region. With attenuation, the radar reflectivity showed a rapid

signal decline below the melting layer, and the bright band became evident (Sassen et al., 2007).

For the 94 GHz radar, the Mie scattering effect was dominant. The raindrops with a diameter less than 1 mm are the dominant contributor to the radar reflectivity profile (Kollias and Albrecht, 2005). Although larger snowflakes melt and produce larger raindrops at depth in the melting layer, their contribution to the reflectivity was not significant, owing to a decrease in their number concentration. Therefore, the bright band was not obvious without attenuation; the reflectivity

increased markedly in the upper part of the melting layer but did not decrease considerably in the lower part. However, the bright band at the melting layer was highlighted with attenuation owing to strong attenuation caused by rain, melting snow, and exponential growth of the attenuation.

Figure 10 shows a radar reflectivity comparison between the simulation results and CloudSat CPR observation data. The cross-sections in Fig. 10a and b show simulation results, where Fig. 10a corresponds to the improved microphysical

parameter settings shown in Fig. 9h, and Fig. 10b corresponds to the conventional setting. Figure 10c shows the observation results from the CloudSat CPR. The lines in Fig. 10d show the average vertical profiles of the reflectivity factor in Fig. 10a–c. The echo structure and echo intensity of the simulation results with the improved setting showed good agreement with the CloudSat observations. The trends in the two profiles were basically identical; the relative error ($|Z_{sim} - Z_{obs}|/Z_{obs}$, where $Z_{sim}$ represents the simulated reflectivity and $Z_{obs}$ represents the observations, the units of $Z_{sim}$ and $Z_{obs}$ are converted to mm$^6$/m$^3$)

at each height was within 20 %. The location and intensity of the bright band from the improved simulation and CloudSat observation were highly consistent; the radar reflectivity peak for both were approximately 12 dBZ at 2.88 km with a bright band width of approximately 0.9 km. Without the melting model, the PSD parameters for raindrops were based on the assumed fixed value. In Fig.10b, the radar reflectivity below 0 ℃ was evidently stronger than the echo above 0 ℃; the width and location of the bright band were considerably different from the bright band in the simulation with the improved setting

and CloudSat observation. The relative error in the average profile below the melting layer reached 40 %. The radar reflectivity peak in the vertical profile from the conventional simulation was 13 dBZ at approximately 2.6 km with a bright band width of approximately 1.4 km. In summary, the melting model can accurately capture the stratiform cloud precipitation characteristics.

## 4.2 Convective case

### 4.2.1 WRF scenario simulation

This case was a severe convective weather process that occurred in the Lower Yangtze-Huaihe river on June 23, 2016, in which strong winds and heavy rainfall occurred in Yancheng and Lianyungang City, Jiangsu Province. The simulation area covered 32–36°N and 116–120.5°E. Triple nested grids were adopted, with horizontal resolutions of 22.5, 7.5, and 1.5 km. More details about WRF model setup for the convective case can refer to Appendix A.

For validating the model result, the ERA5 data, ground radar reflectivity and rain gauge data were used. Figure 11a-f compares the fraction of cloud cover, reflectivity, and rainfall from the WRF model with the observation data. Figure 11a shows the fraction of cloud cover from the WRF model at d02 domain. The cloud area and coverage are consistent with the ERA5 data shown at Fig. 11b. Figure 11c and d compare the reflectivity from the WRF simulation over the d03 domain at 04:00 UTC on June 23 and ground radar at Lianyungang city at 04:02 UTC on June 23, 2016. From radar observation, we

can see that the strong echo area is relatively scattered, generally trending from northwest to southeast, and the maximum reflectivity is about 55 dBZ. In the simulation, the strong radar echo is mainly distributed along the northwest-southeast; the radar echo structure and echo intensity are close to the radar observation. Figure 11e and f show the 6-hour accumulated rainfall from 00:00 to 06:00 on June 23 from the WRF model and rain gauge data, respectively. The rainfall covers most areas in the north of Jiangsu Province, and there are two heavy rainfall centers of more than 100mm. The rainfall area in the

simulation is similar to that from rain gauge data, and three heavy rainfall centers can be seen in the model result. The maximum rainfall from rain gauge data is approximately 120 mm and maximum from WRF is approximately 126 mm. The amount, scope, and distribution of rainfall from WRF simulation are generally consistent with the rain gauge data. The main difference is in the strong rainfall location and extreme value. Considering the model limitations, the comparison results show that the model captured the convective precipitation process.

**4.2.2 Experiment design**

        CloudSat observed this convective process at 04:30 AM on June 23, 2016, covering the cloud region from 32.43°N, 119.13°E to 36.11°N, and 118.10°E. For comparison with the CloudSat data, the vertical cross-section of the hydrometeor matching the CloudSat observation was selected for simulation. Figure 12a–f shows the latitude-height cross-section of the hydrometeor for the convective case simulated by the WRF for the total hydrometeors, cloud water, cloud ice, snow, graupel,

and rain. The ice water content of the convective case was large, and the vertical extent of cloud ice, snow, and graupel particles were widely distributed with high contents. Snow existed from 4 to 14 km, with a water content reaching approximately 1.5 g/m$^3$. Graupel particles mainly ranged from 4–8 km, with a maximum water content of 1.2 g/m$^3$. Rain was mainly distributed between 34 and 36ºN, and the water content of the rain near 34.5 and 34.8ºN reached 5 g/m$^3$.

        In the convective case, snow and graupel were abundant. Unlike stratiform clouds, a large percentage of heavily

aggregated and/or rimed snow commonly exist in convective clouds (Yin et al., 2017); therefore, rimed particles were assumed for convective clouds modeling. Considering the effect of riming, a varying mass-power relationship was assumed in the simulation with improved setting. As the prefactor $a$ in the mass-power relations increases with the riming degree (Mason et al., 2018; Moisseev et al., 2017; Ryzhkov and Zrnic, 2019), an adjustment factor $f$ was considered in the simulation process, i.e., $a=a_u f$, where $a_u$ is the density prefactor for unrimed snow. $f$ is obtained from $f=1/(1-FR)$ where FR is

the ratio of the rime mass to the snowflake mass. According to Moisseev et al (2017), $FR$ can be expressed as a function of the effective liquid water path (ELWP), $ELWP \approx 4\alpha_u / \pi \cdot FR / (1-FR)$, given that the rime mass is determined by the mass

of swept supercooled liquid droplets. Considering the connection between ELWP and liquid water path LWP (according to Moisseev et al (2017), ELWP is approximately half of LWP), we assumed that the adjust factor $f$ increased linearly with LWP, and the relation between $f$ and LWP was derived to be $f \approx 0.5\pi LWP/\alpha_u + 1$. The assumption ignores possible changes in particle mass linked to the presence of different crystal habits, and the exponent $b$ in the mass-size relation remains constant. Large uncertainty may occur in the cases where majority of precipitation occurs in the form of crystals. The exponent $b$ for snow was assumed to be the mean value of 2.1 based on the sensitivity analysis. Then, the corresponding scattering properties and PSD for snow and graupel were calculated according to the mass-power relations.

The effect of riming was not considered in the conventional simulation. In the simulation with the conventional microphysical setting, a mass-power relation of $m=0.0075D^{2.05}$ of unrimed snow (Moisseev et al., 2017) was used for simulation of snow particles, and a constant density of 0.4 g/cm$^3$ was assumed for graupel particles.

### 4.2.3 Radar reflectivity simulation results

Figure 12g–i show the simulated radar reflectivity with the total hydrometeors, where Fig. 12g shows the unattenuated reflectivity, Fig. 12h shows the two-way attenuation, and Fig. 12i shows the attenuated reflectivity. Figure 12i shows that the internal vertical structure of the deep convective cloud can be accurately detected, but millimeter wave radar has difficultly penetrating the rainfall layer due to strong attenuation. Figure 13 shows a radar reflectivity comparison between the simulation results and CloudSat CPR observations for the deep convective case, where Fig. 13a–c shows the cross-sections of the reflectivity from simulations with improved and conventional settings, as well as the CloudSat observations. The lines in Fig. 13d are the average profiles corresponding to Fig. 13a–c. Figure 13a and c show that the echo distribution and echo intensity of the simulation and CloudSat observation are in good agreement. The echo top heights were approximately 16 km and the maximum reflectivity factor was approximately 18 dBZ. The hydrometeors for the cloud water, graupel, and rainfall particles were mainly concentrated between 34.5ºN and 35.5ºN, which produced strong echoes at middle heights and strong attenuation at lower heights. Comparing the profiles with the improved simulation to those with the conventional simulation in Fig. 13d, the fixed density in the conventional simulation caused the echo at high altitudes to be stronger and the echo at low altitudes to be weaker.

To further illustrate the effect of snow and graupel, Fig. 14 shows the water content and reflectivity profiles for snow and graupel corresponding to the black line in Fig. 13a. Figure 14a shows the vertical profile of the water content of snow and graupel. Figure 14b shows the simulation results corresponding to the hydrometeor profile in Fig. 14a. Relative to the reflectivity results with the conventional simulation, snow and graupel in the improved simulation showed weak echo at high altitudes and strong echo at low altitudes. The trend in the profile for snow and graupel in Fig. 14b is the same as that in the average profiles shown in Fig. 13d. The vertical profile in the improved simulation showed good consistency with that of the CloudSat observation, with an average relative error of approximately 20 %. In contrast, the average relative error in the conventional simulation reached approximately 100 %. The simulation results demonstrated that the radar reflectivity is

highly sensitive to the prefactor of the mass-power relation of snow and graupel; the effect of riming on the prefactor should
be considered in the forward modeling simulations or microphysical parameter retrieval for convective clouds.

**5 Conclusions**

Active remote sensing with spaceborne millimeter wave radar is one of the most effective means of cloud and precipitation
measurements. Many countries are developing next generation spaceborne cloud precipitation radar. During the design and
demonstration stage of observation systems and in the interpretation of observation data, forward modeling simulations play
a crucial role. The physical characteristics of hydrometeor particles, such as the shape, density, composition, PSD model and
parameters, have an important impact on the simulation results. Based on establishing a simulation framework with eight sub
modules, we quantified the uncertainty of different physical model parameters for hydrometeors via a sensitivity analysis,
presenting radar reflectivity simulations with optimized parameter settings.

   The sensitivity of radar reflectivity to changes in $D_0$ in the Gamma distribution was approximately 5–10-fold greater
than that of $\mu$; the variation in $\mu$ can cause reflectivity changes of less than 10 %. The constraints on PSD modeling from
the empirical relationships in the observations using interconnected parameters, rather than independent variations, can
significantly reduce the impact of PSD variation. Owing to the constraint on the total number concentration for the PSD of
cloud ice, the effect of $D_0$ on the radar reflectivity can be reduced by approximately 60 %. The mass-diameter relationships
for snow and graupel differ substantially for different particle habit types. Using the exponential PSD with a power-law mass
spectrum for snow and graupel, we found that the effects of prefactor $a$ on radar reflectivity were significant. Variation in $a$
mainly may result in reflectivity uncertainty of approximately 45 % for snow and 30 % for graupel, mainly due to changes in
the particle scattering properties. Owing to complex physical characteristics resulting from various microphysical processes,
the shape and orientation of frozen and mixed phase particles are variable. The assumption of sphere and spheroid could lead
to an average reflectivity difference of approximately 4–14 %. In addition to the PSD parameter and particle shape and
orientation, this study emphasized the importance of the particle mass parameters and PSD modeling constraints
corresponding to different cloud precipitation types in the forward simulation and microphysical properties retrieval.

   Two typical cloud precipitation cases were presented. The simulation results were compared with the CloudSat
observations. During simulation, we considered the PSD parameter settings for typical cloud precipitation types, particle
shapes, melting models, and influence of snow and graupel density relations. For snow and graupel microphysical modeling,
unrimed snow particles was assumed in the stratiform clouds, and rimed snow with varying density-power relations was
considered to be in the convective clouds. The simulation results with the improved microphysical setting showed good
agreement with the CloudSat observations. The average relative errors in radar reflectivity profile between the simulation
and CloudSat data were within 20 %, which improved by 20–80 percent points compared with the conventional setting, i.e.,
not considering the melting model and riming effect for snow and graupel. The melting layer modeling for stratiform clouds

accurately reproduce the bright band structure after attenuation. The varying prefactor of mass relations of snow and graupel considering the riming effects for convective clouds rendered the simulated echo structure consistent with the observations.

The selection and modeling of cloud microphysical characteristics not only affects the forward simulation and numerical modeling, but also has a significant impact on physical parameter retrieval. This study contributes to a quantitative understanding of the uncertainties of forward simulations or radar retrievals due to variation in the microphysical properties

of hydrometeors. It also provides a scientific basis for the analysis of millimeter wave radar observation data, the improvement of parameter settings in forward modeling, and microphysical constraints in parameter retrievals. The sensitivity test and simulation results suggest that accurate estimation of at least two parameters in the size distributions of hydrometeor particles including particle density factor is beneficial using certain methods. In future studies, we will consider establishing a cloud database for further improving prior information constraints by collecting a large amount of typical

cloud precipitation microphysical observation data at different climatic regions.

## Appendix A: Model setup and verification

### a. Stratiform case

The simulation for this stratiform case was conducted with four nested grids (d01, d02, d03, and d04), and the inner domain was centered at 41.08°N, 117.61°E. The horizontal grid spacings are 27km, 9km, 3km, and 1km and the corresponding grids

are 120×120, 180×180, 300×300, and 300×300. The vertical resolution increases with height from approximately 50 m near the surface to 600 m near 50 hPa. Time steps of 180s and 6.67s were used for d01 domain and d04 domain, respectively. The 6-hourly NCEP FNL operational global analysis data on 1° × 1° grids were used to provide the initial and boundary conditions. In term of physical scheme, the model adopted CAM 5.1 5-class scheme, Grell-Freitas cumulus parameterization scheme, RRTM long and short-wave radiation scheme, YSU boundary layer scheme, Monin-Obukhov surface layer scheme

and thermal diffusion land surface scheme. The cumulus parameterization scheme was used for d01 and d02 domains only. The simulation starts at 12:00 UTC on 24 September and ends at 12:00 UTC on 25 September 2012.

Besides the cloud fraction and cloud top temperature shown in Fig. 8, the cloud water path (CWP) from MODIS was used for model result verification as well. Figure A1a is the cloud water path calculated from vertical integration of WRF output cloud water over d01 domain at 03:30 UTC, 25 September, and Fig. A1b is the cloud water path from MODIS Level

2 product at 03:35 UTC, 25 September 2012. The scanning width of MODIS is 2330 km, and the horizontal resolution for the product of CWP is 1 km. The CWP distribution of model result has similar pattern as MODIS observation and the value of CWP are close, but the peaks of the two are slightly offset. Due to the measurement techniques and model limitations, the model simulations may be biased from the observations. However, the distribution, structure, and value of CWP from model and MODIS observation generally agree well.

## b. Convective case

For the convective case, three nested grids (d01, d02 and d03) with horizontal grid spacings of 22.5km, 7.5km, and 1.5km and corresponding grid points of 70×70, 126×126, and 280×280 were used for the convective case simulation. The inner domain d03 is centered at 34.02°N, 118.20°E. A total of 39 vertical layers with stretch spacing from the surface to 50 hPa were used, with time steps of 90, 30, and 6 s for d01, d02 and d03, respectively. The initial and boundary conditions used the NCEP FNL analysis data as well. The model adopted NSSL 2-moment 4-ice scheme for microphysical process, Kain-Fritsch cumulus parameterization scheme, RRTMG long and short-wave radiation scheme, YSU boundary layer scheme and five-layer thermal diffusion land surface scheme. The Kain-Fritsch cumulus scheme was not used for d03 domain. The simulation starts at 12:00 UTC on 22 June and ends at 12:00 UTC on 23 June 2016.

*Data availability*. The NCEP FNL reanalysis data for driving WRF model simulation are available at https://rda.ucar.edu/datasets/ds083.2. The CloudSat data are available at https://www.cloudsat.cira.colostate.edu/data-products.

*Author contributions*. LL and ZJ carried out the sensitivity analysis, weather case simulation experiment and data analysis; HY and SJ carried out the WRF model simulation and provided the input. PM provided technique support and analysis methods. LL wrote the original article with feedback from all the co-authors.

*Competing interests*. The authors declare that they have no conflict of interest.

*Acknowledgements*. We acknowledge the data providers of the National Centers for the Environmental Prediction Final Operational Model Global Tropospheric Analyses. We would like to thank the scientists and those who contribute to CloudSat mission. This study was supported in part by the National Key Research and Development Program of China (2021YFC2802502), in part by the National Natural Science Foundation of China (41975027).

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

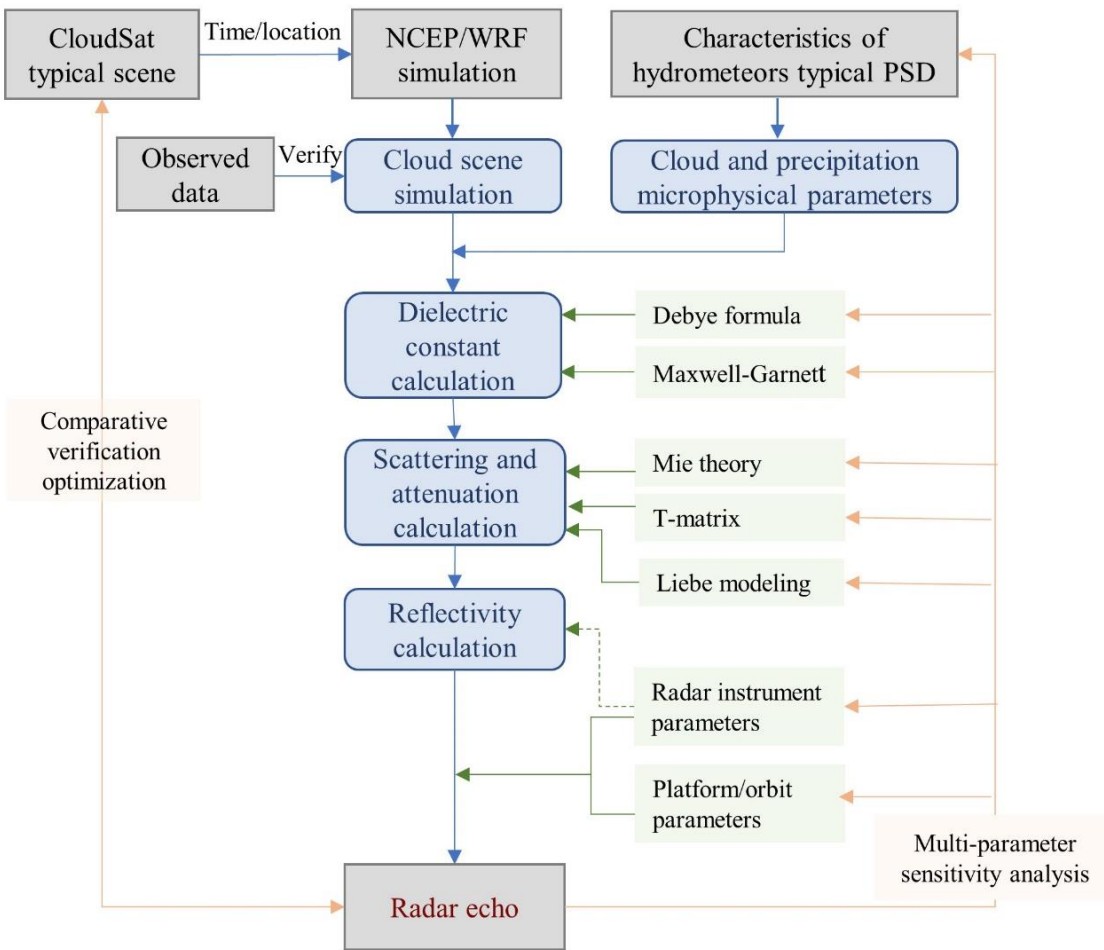

**Figure 1: Sub module structure and framework of the simulation model**

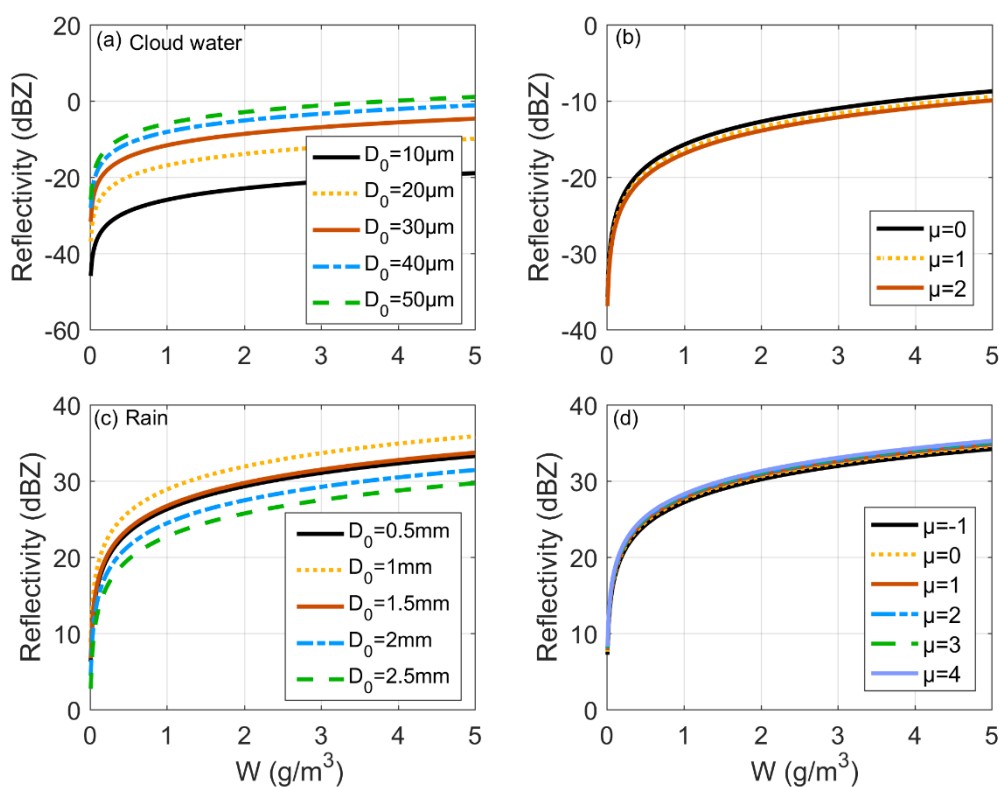

**Figure 2: Impact of the PSD parameters, i.e., $D_0$ and $\mu$, on radar reflectivity for cloud water and rain. Reflectivity variation in cloud water caused by (a) $D_0$ of 10, 20, 30, 40, and 50 $\mu m$ with a $\mu$ of 1 and (b) $\mu$ values of 0, 1, and 2 with $D_0$ of 20 $\mu m$. Reflectivity variation in rain caused by (c) $D_0$ of 0.5, 1, 1.5, 2, and 2.5 mm with a $\mu$ of 3 and (d) $\mu$ values of –1, 0, 1, 2, 3, and 4 with $D_0$ of 1.25 mm.**

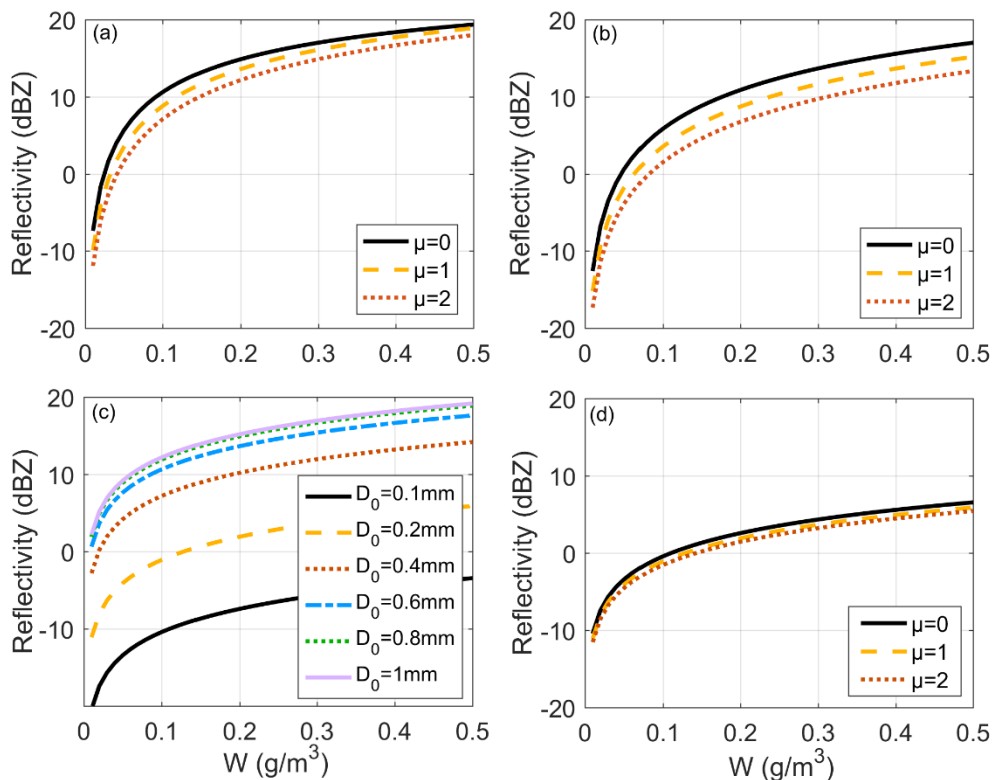

**Figure 3: Impact of PSD parameters on radar reflectivity for cloud ice. PSD parameters constrained by Eqs. (7)–(9); reflectivity variation obtained when $\mu$ was 0, 1, and 2 and (a) temperature $T$ was –20 ºC and (b) $T$ of –60 ºC. PSD parameters varied independently; reflectivity variation obtained by (c) $D_0$ of 0.1, 0.2, 0.4, 0.6, 0.8, and 1 mm with $\mu$ of 1 and (d) $\mu$ values of 0, 1, and 2 with $D_0$ of 0.2 mm.**

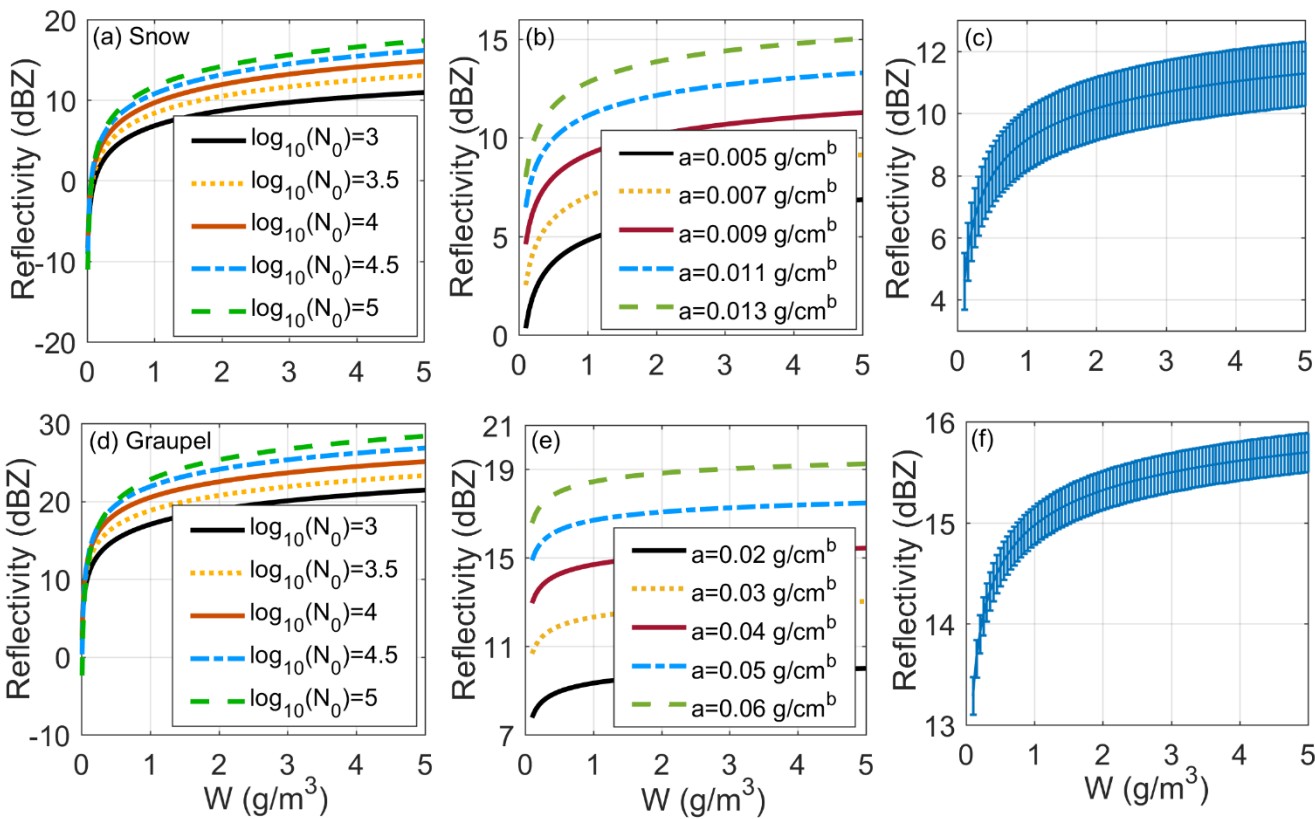

**Figure 4: Impact of PSD parameters on radar reflectivity for snow and graupel. Variation in reflectivity for snow at (a) $\log_{10}(N_0)$ values of 3, 3.5, 4, 4.5, and 5 with a mean mass-diameter relationship of $m = 0.009D^{2.1}$, where $D$ is in cm and $m$ is in g; (b) prefactor $a$ in mass-diameter relationship of 0.005, 0.007, 0.009, 0.011, and 0.013 $g/cm^b$, with exponent $b$ of 2.1 and $N_0$ assumed to be $3 \times 10^3$ $m^{-3}\,mm^{-1}$; (c) mean value $\pm$ standard deviation of $b$, where the mean is 2.1 and standard deviation (SD) is 0.28, with $a$ assumed to be 0.009. The vertical bars represent the SD of the reflectivity change caused by deviation from the mean value of $b$. Variation in reflectivity for graupel at (d) $\log_{10}(N_0)$ values of 3, 3.5, 4, 4.5, and 5 with a mean mass-diameter relationship of $m = 0.04D^{2.6}$, where $D$ is in cm and $m$ is in g; (e) prefactor $a$ in mass-diameter relationship of 0.02, 0.03, 0.04, 0.05, and 0.06 $g/cm^b$, with exponent $b$ of 2.6 and $N_0$ assumed to be $4 \times 10^3$ $m^{-3}\,mm^{-1}$; and (f) mean value $\pm$ standard deviation of $b$, where the mean is 2.6 and standard deviation is 0.16, with $a$ assumed to be 0.04. For $a$ and $b$ we took literature values from list of studies and calculated the mean, and the standard deviation of $b$ for snow and graupel are calculated according to the range and average of Gaussian distribution.**

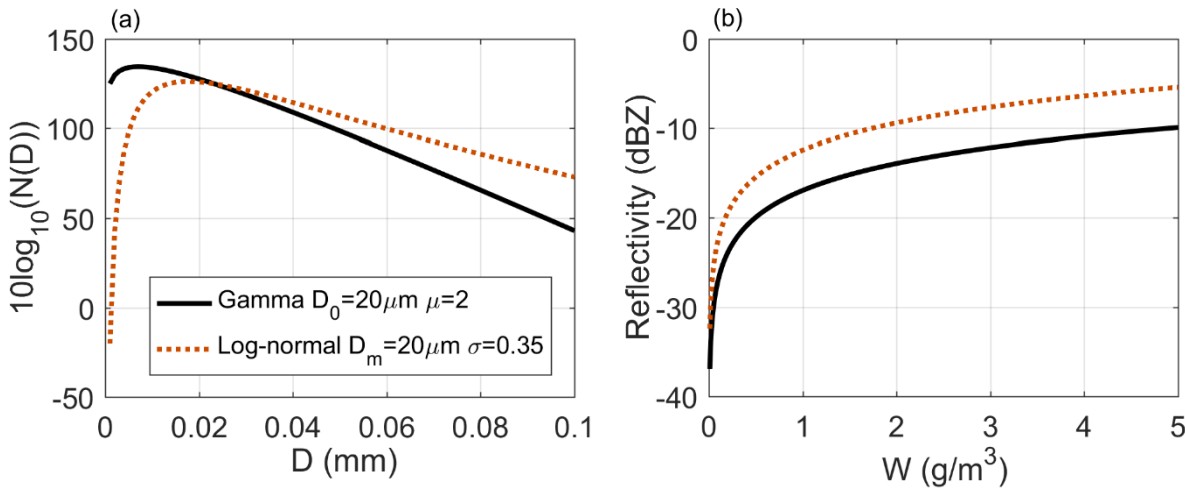

**Figure 5: Impact of PSD models on radar reflectivity for cloud water. (a) Black solid line is for the gamma distribution: $W = 1$ ($g/m^3$), $D_0 = 20\ \mu m$, and $\mu = 2$. Red-dotted line is for the log-normal distribution: $W = 1\ g/m^3$, $D_m = 20\ \mu m$, and $\sigma = 0.35$. (b) Variation in the radar reflectivity with $W$ and the PSD models, where the PSD models are from (a).**

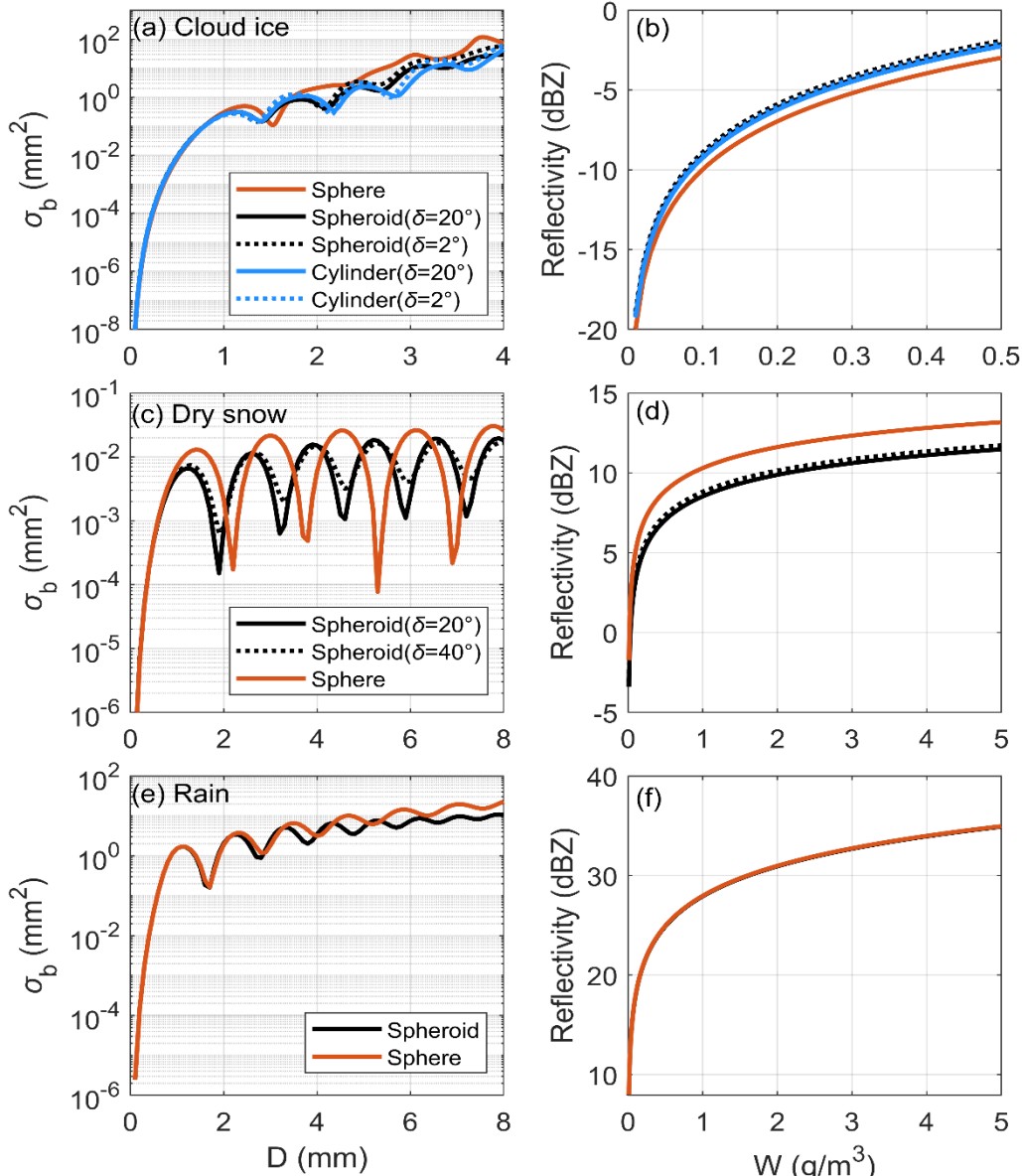

**Figure 6: Backscattering cross-section and corresponding radar reflectivity under different shapes for cloud ice, dry snow, and rain. (a) Comparison of the backscattering cross-sections of ice crystals as spheres, spheroids, or cylinders, where $\delta$ is the SD of the canting angle. (b) Radar reflectivity comparison for particles in (a), where the PSD was assumed as a Gamma distribution constrained by Eqs. (7)–(9), with $\mu=1$ and T = –60º C. (c) Comparison of the backscattering cross-sections for dry snow with spheres and spheroids. (d) Radar reflectivity comparison for particles in (c), where the PSD was assumed as an exponential distribution with $N_0 = 3 \times 10^3 \, \text{m}^{-3} \, \text{mm}^{-1}$. (e) Comparison of backscattering cross-sections for raindrops with spheres and spheroids. (f) Radar reflectivity comparison for particles in (e), where the PSD was assumed as a Gamma distribution with $D_0 = 1.25$ mm and $\mu=3$.**

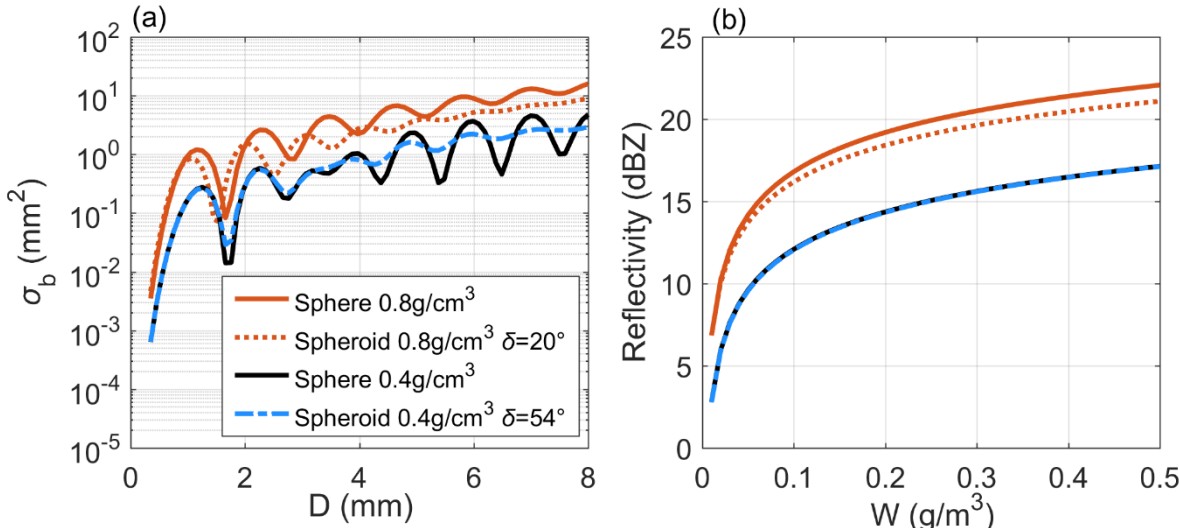

**Figure 7: Comparison of the backscattering cross-section and corresponding radar reflectivity for graupel between spheres and spheroids at different densities and orientations. (a) Backscattering cross-section at a density of 0.4 and 0.8 g/cm³ with $\delta$ (SD of canting angle) calculated from Eq. (14). After calculation, $\delta$ was 54º at a density of 0.4 g/cm³ while $\delta$ was 20º at a density of 0.8 g/cm³. (b) Radar reflectivity for particles in (a), where the PSD was assumed as an exponential distribution with $N_0$ of $4 \times 10^3$ m⁻³ mm⁻¹. Overestimation caused by the spherical assumption increased with an increase in density and decrease in $\delta$.**

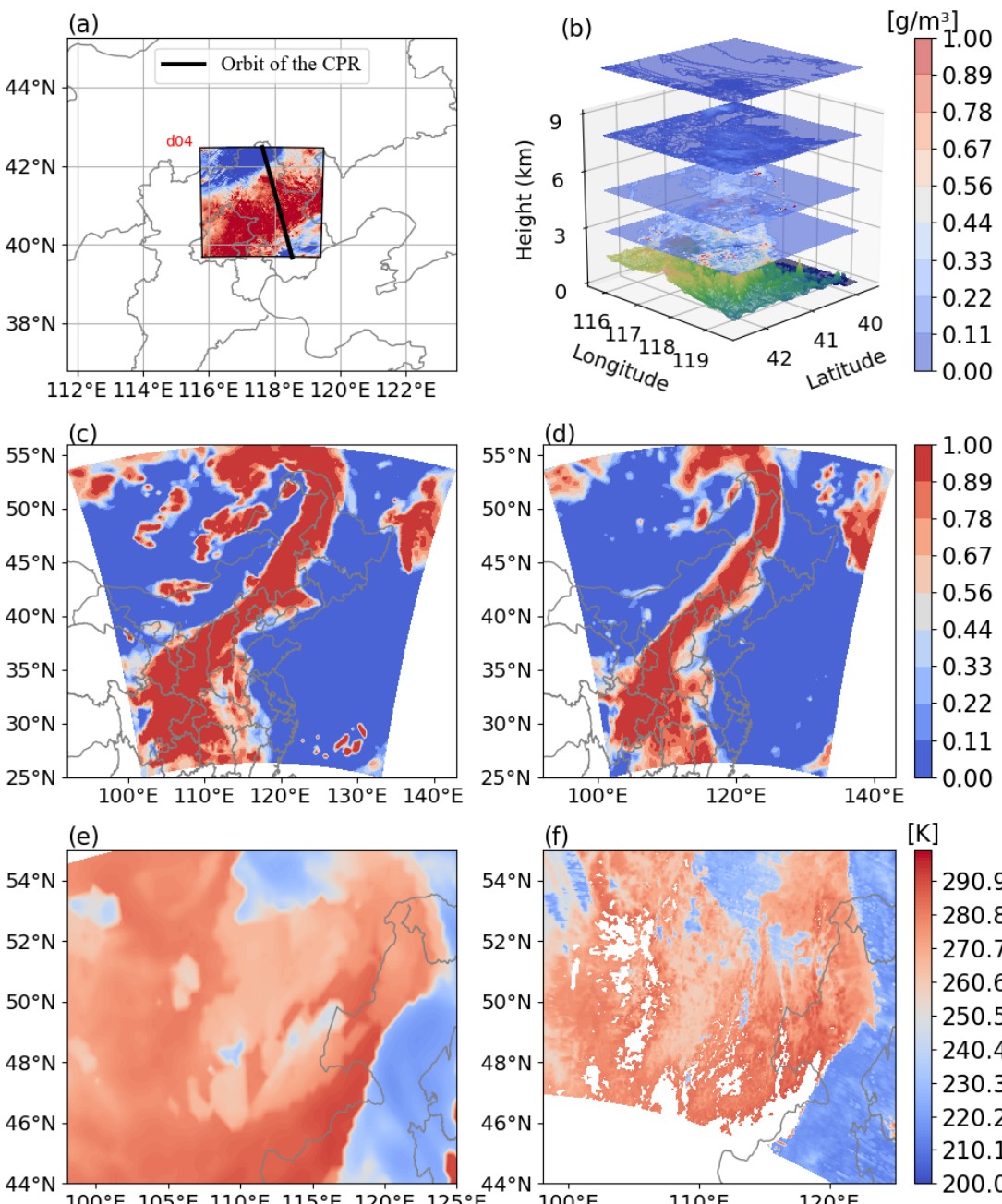

**Figure 8: Simulation area exhibition of the stratiform cloud case scenario and comparison between the WRF model results and observation data. (a) Exhibition of the internal two-layer simulation area, (b) 3-D distribution of the total hydrometeor output from the WRF, (c) fraction of cloud cover from the WRF model, (d) fraction of cloud cover from the ERA5 data, (e) WRF model-simulated cloud top temperature, and (f) MODIS-observed cloud top temperature.**

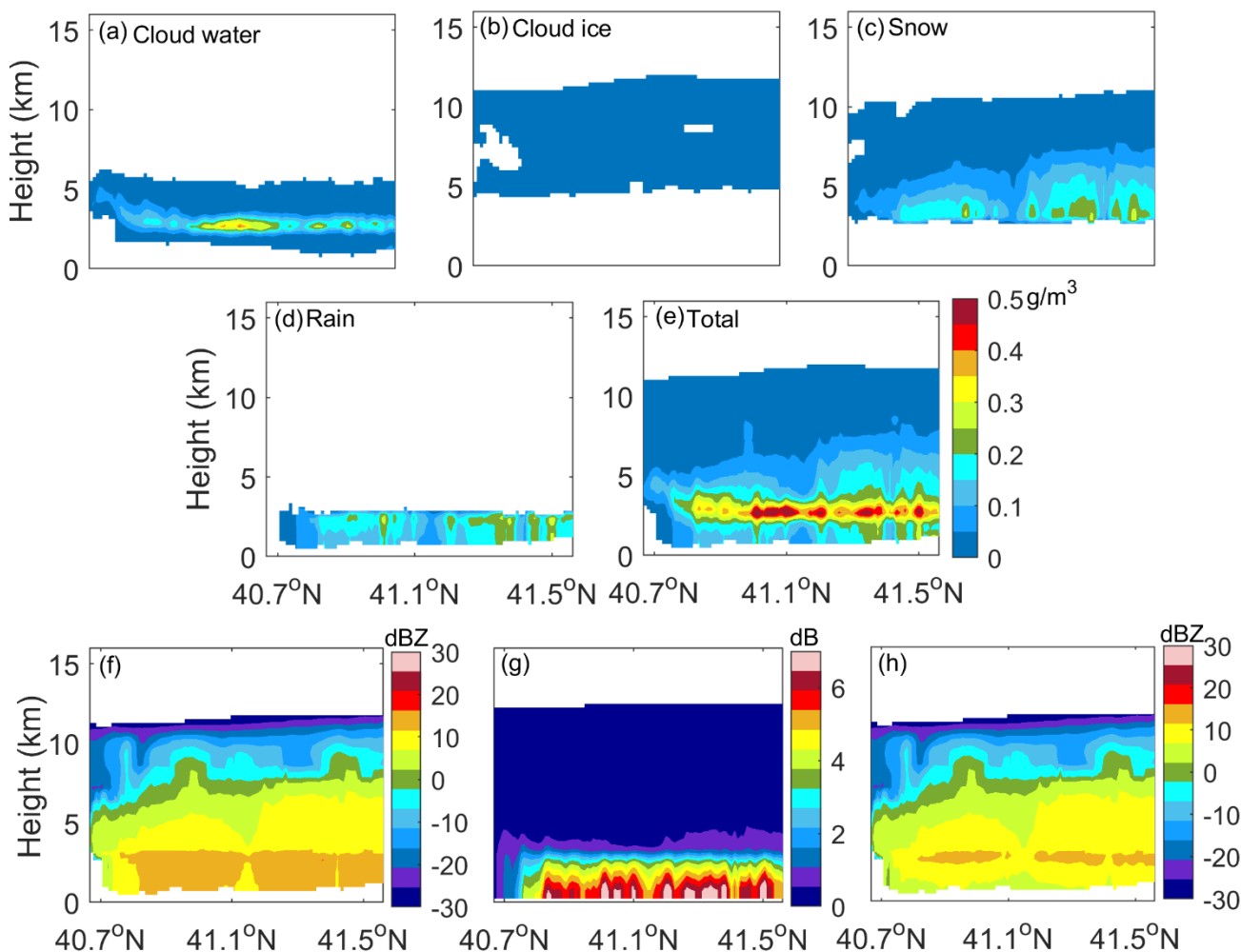

**Figure 9: Latitude-height cross-section of the hydrometeor for the stratiform case simulated by the WRF for: (a) cloud water, (b) cloud ice, (c) snow, (d) rain, and (e) total hydrometeors. (f) Simulated unattenuated radar reflectivity with the total hydrometeors, (g) two-way attenuation, and (h) attenuated radar reflectivity. Owing to the Mie scattering effect, the unattenuated radar reflectivity did not decrease markedly at the bottom of the melting layer, whereas the bright band at the melting layer was highlighted due to strong attenuation in the rain region.**

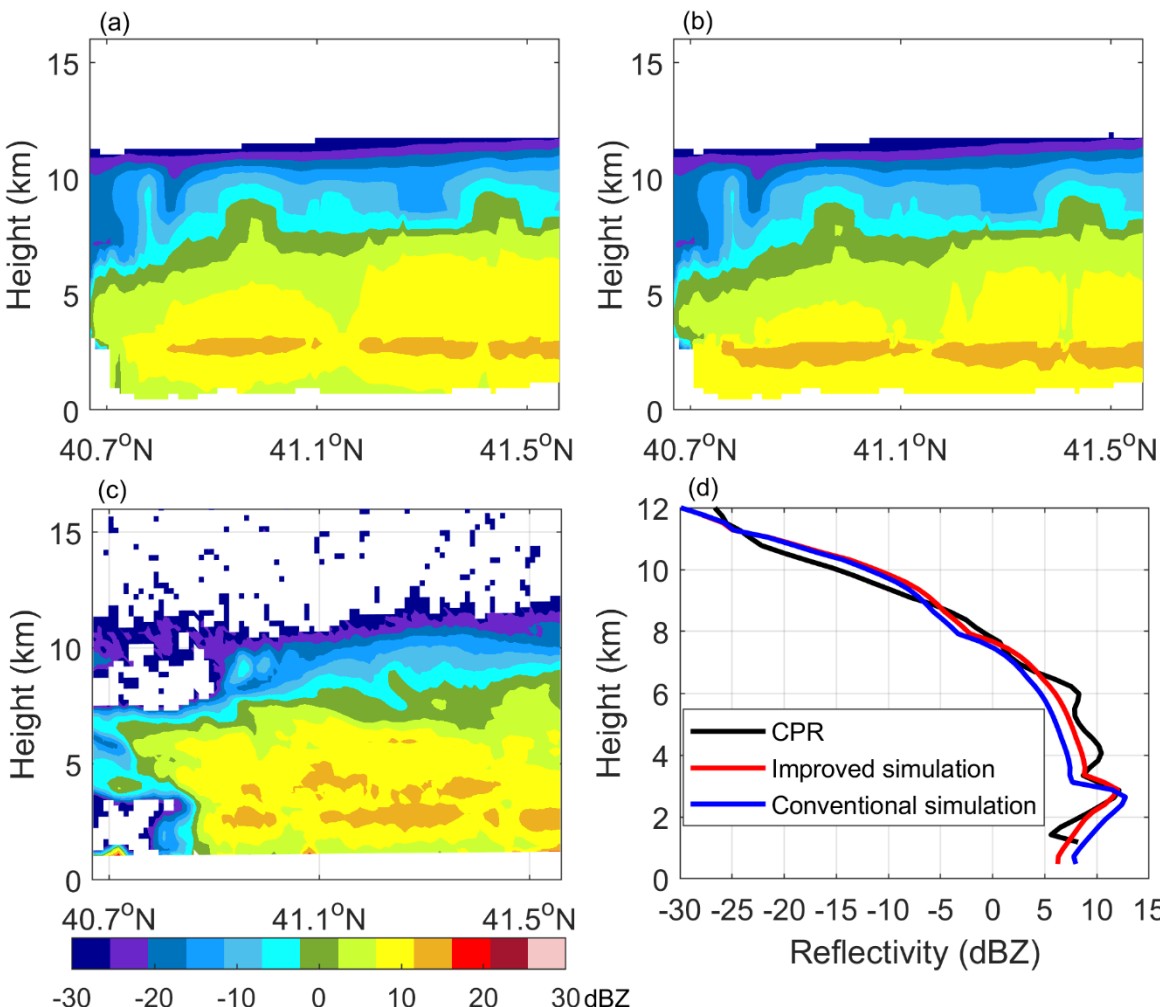

**Figure 10: Radar reflectivity comparison between the simulation results and CloudSat CPR observation data for the stratiform cloud precipitation case. (a) Cross-section of the simulation result with improved settings, (b) cross-section of the simulation result with the conventional settings, and (c) cross-section of the CloudSat CPR observation data. (d) Vertical profiles of the average reflectivity in (a)–(c), where the red line represents the simulation result with the improved settings, the blue line represents the**

890 **simulation results with the conventional settings, and the black line represents the results of the CPR observation. Owing to the melting modeling in the improved simulation, the echo structure and intensity were consistent with the CPR observation results.**

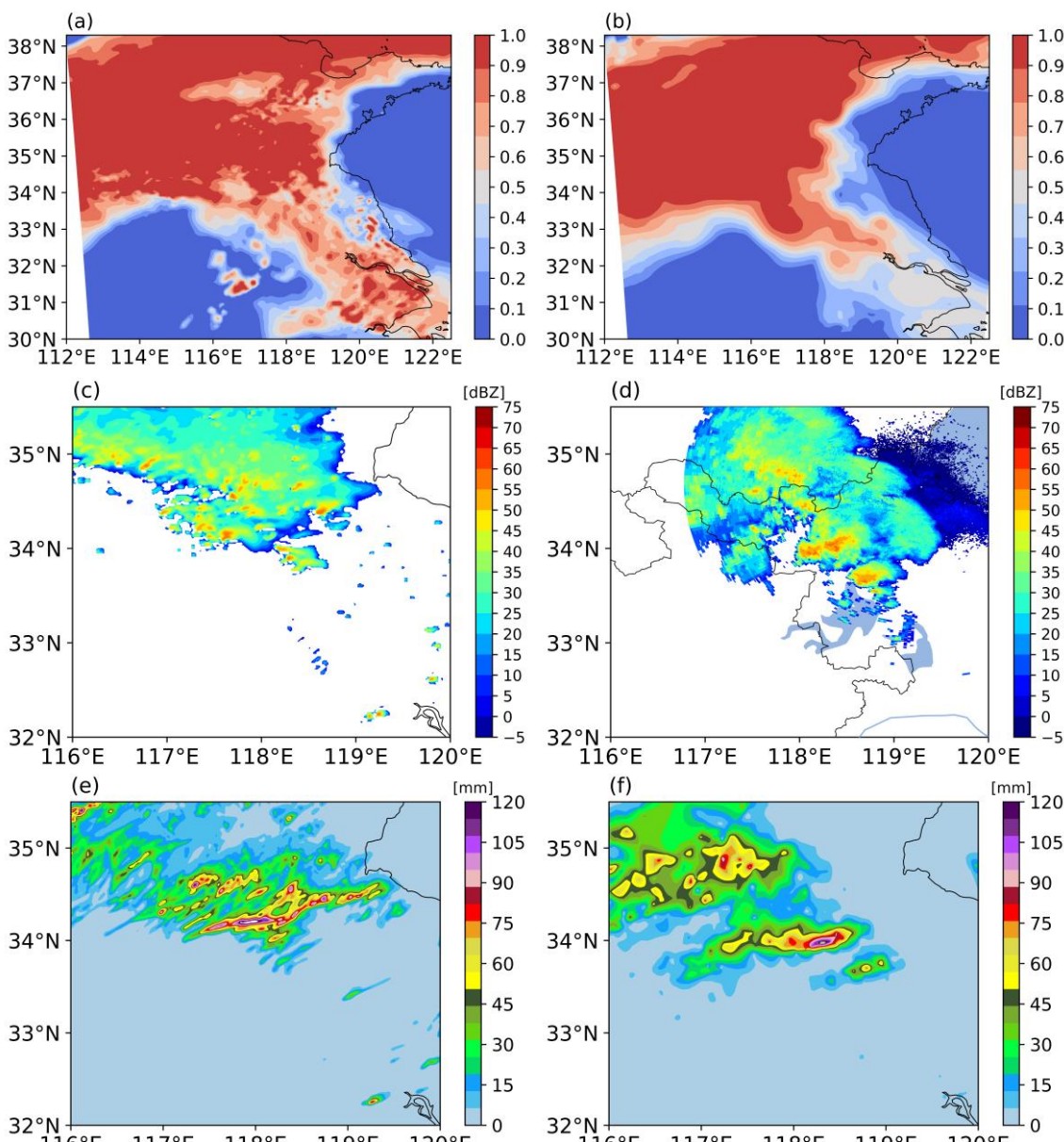

**Figure 11: Comparison between the WRF model results and observation data for the convective case. (a) Fraction of cloud cover from the WRF model, (b) fraction of cloud cover from the ERA5 data, (c) radar reflectivity from the WRF model at 04:00 UTC, (d) radar reflectivity observed by the Lianyungang radar at 04:02 UTC, 23 June, (e) WRF model-simulated 6h accumulated rainfall from 0:00 to 06:00 UTC, 23 June, (f) 6h accumulated rainfall from rain gauge data from 0:00 to 06:00 UTC, 23 June 2016.**

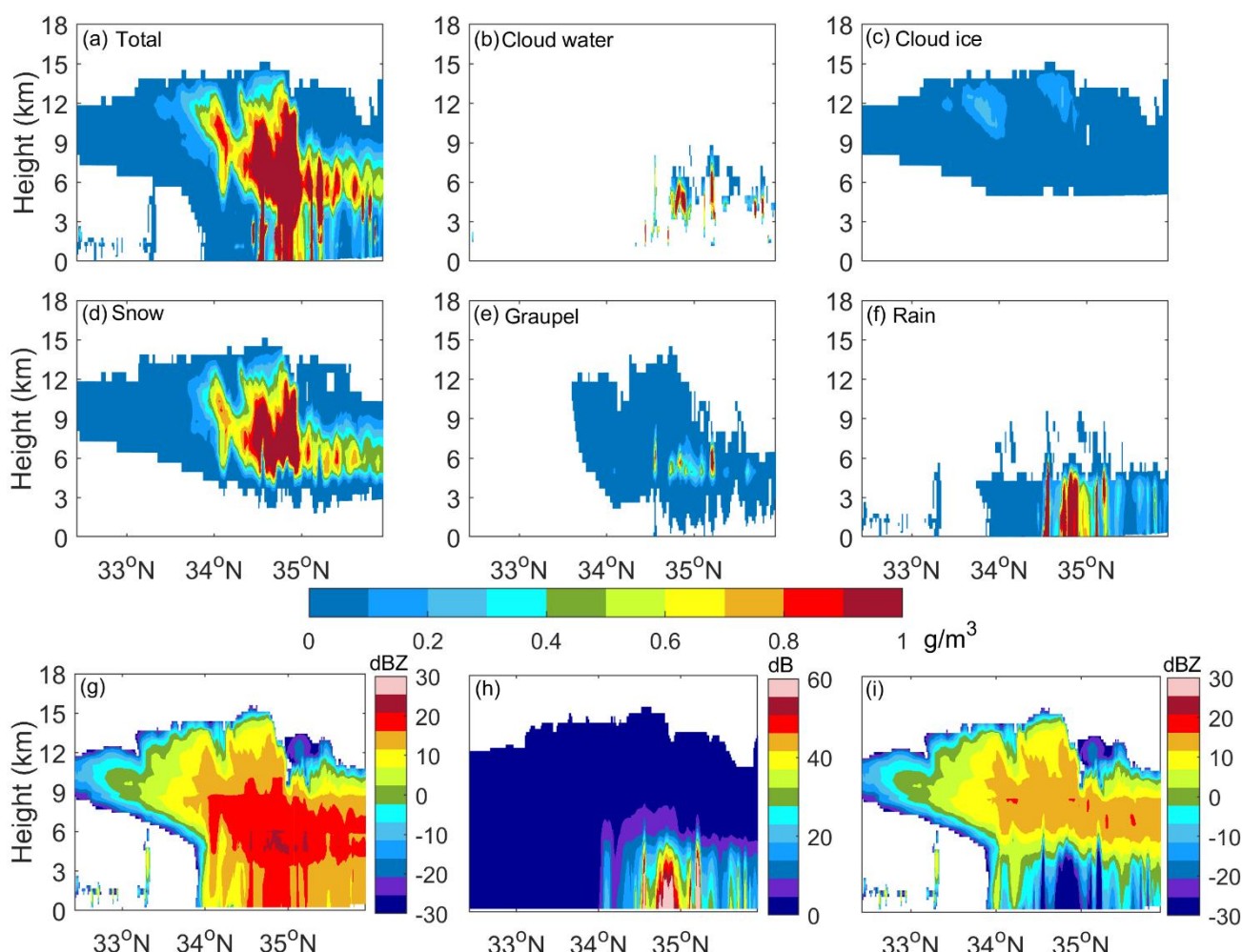

**Figure 12:** Latitude-height cross-section of the hydrometeor of the convective case simulated by the WRF for: (a) total hydrometeors, (b) cloud water, (c) cloud ice, (d) snow, (e) graupel, (f) rain. (g) Simulated unattenuated radar reflectivity with the total hydrometeors, (h) two-way attenuation, and (i) attenuated radar reflectivity.

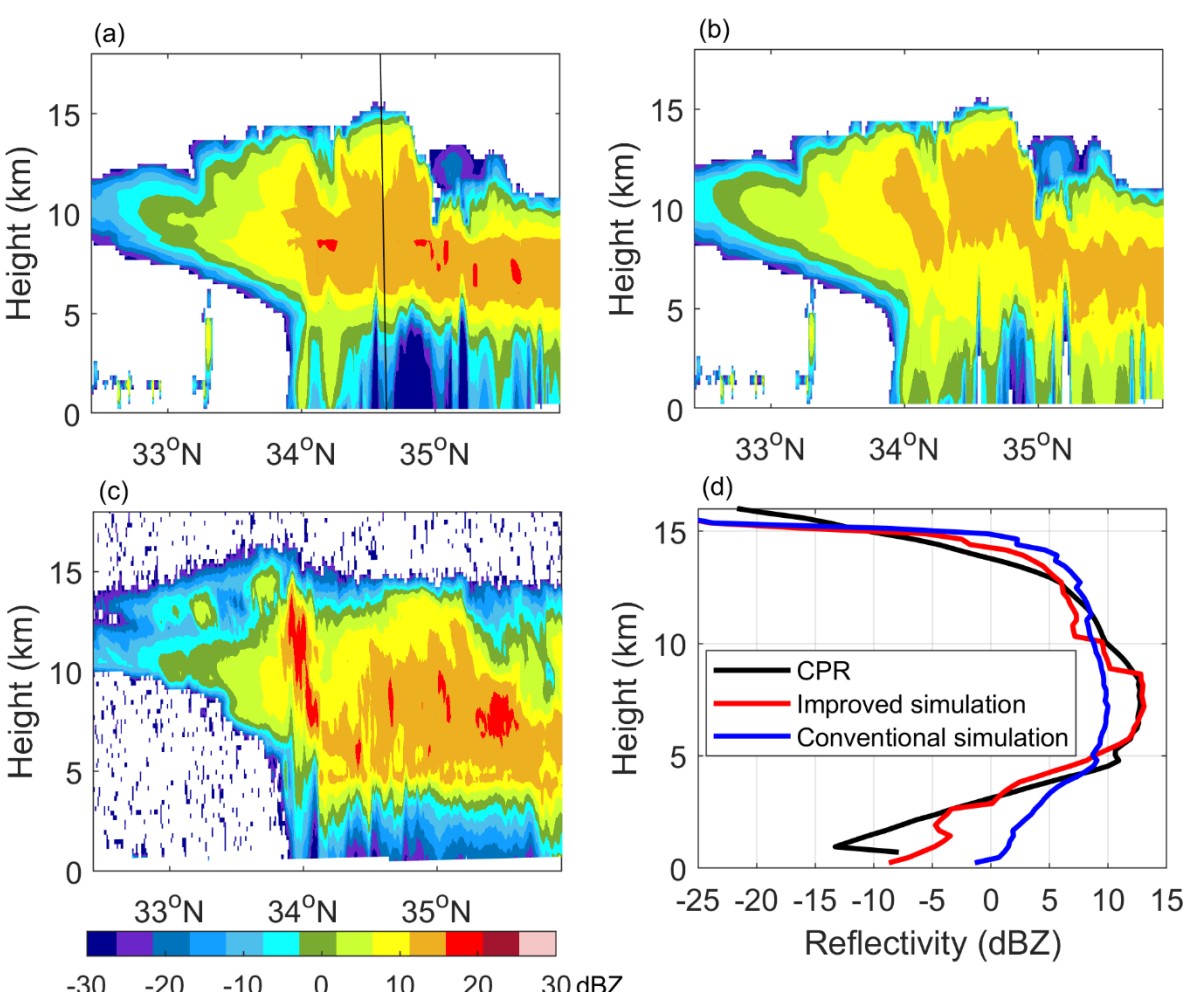

**Figure 13: Radar reflectivity comparison between the simulation results and CloudSat CPR observation data for the convective case. (a) Cross-section of the simulation result with the improved settings, (b) cross-section of the simulation result with the conventional settings using a fixed particle density, and (c) cross-section of the CloudSat CPR observation data. (d) Vertical profiles of the average reflectivity in (a)–(c), where the red line represents the simulation result with the improved settings, blue line represents the simulation results with the conventional settings, and black line represents the result of the CPR observation. The varying prefactor of density relations of snow and graupel due to the effect of riming was considered in the improved simulation. The echo structure and intensity between the improved simulation and CloudSat observation showed good agreement.**

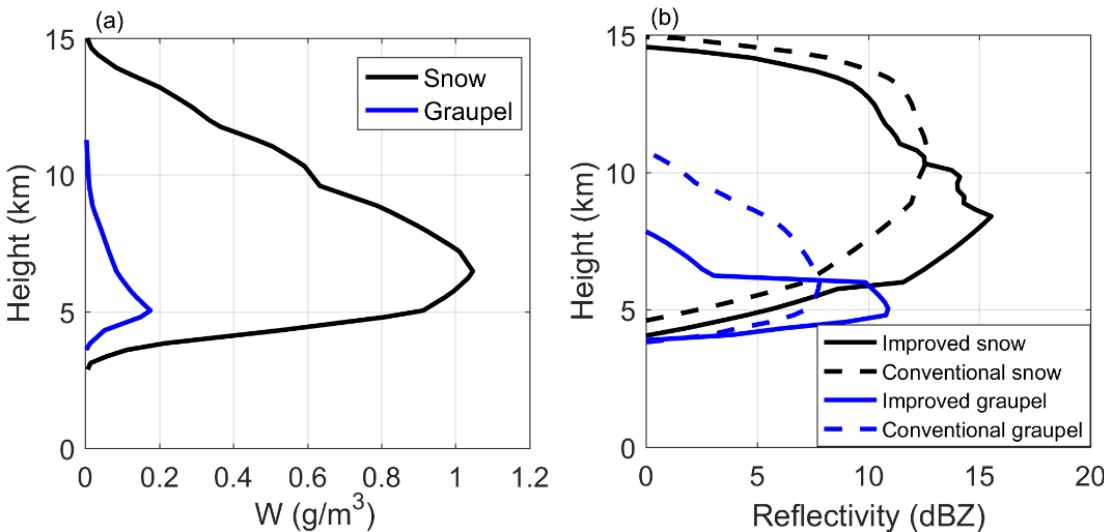

**Figure 14: (a) Vertical profiles of water content for snow and graupel along black line in Fig. 12a, where black line denotes snow and blue line denotes graupel. (b) Corresponding reflectivity profiles with the improved simulation and conventional simulation, where solid lines denote the simulation result with the improved settings and dashed lines denote the simulation result with the conventional settings.**

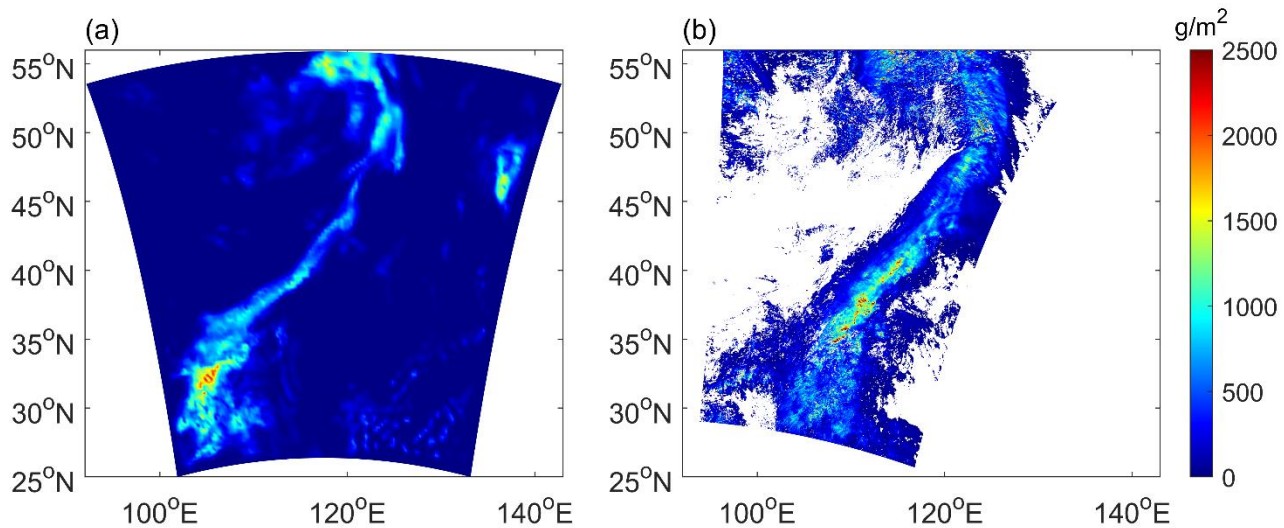

**Figure A1: Comparison of CWP between the WRF model result and MODIS data for the stratiform case. (a) CWP from the WRF simulation at 03:30 UTC, (b) CWP from MODIS observation at 03:35 UTC, 25 September 2012.**