# Peer review of "Simulation and sensitivity analysis for cloud and precipitation measurements via spaceborne millimeter wave radar"

_EGUsphere, 2022_

## Referee Comment (RC1)

**Review of "Simulation and sensitivity analysis for cloud and precipitation measurements via spaceborne millimeter wave radar" by Kou et al.**

**General comments**

The authors present a forward modeling framework to simulate spaceborne millimeter radar reflectivities and perform a sensitivity analysis to evaluate uncertainties in radar reflectivity caused by varying hydrometeor physical parameters in the forward model. Further, they apply the forward modeling framework to WRF model output for two case studies, a stratiform and a convective case, and compare to CloudSat measurements.

In general the study has some very interesting aspects and is suitable for AMT. The inclusion of a melting model is a rather novel approach and not incorporated in most radar simulators. However, I have major concerns regarding the presentation of the study's motivation and the structure of the paper in general. I therefore recommend major revision.

**Specific comments**

- The motivation of the study is not stated clearly to me. I think the aim of the authors is to present their forward modeling framework. If that is the case, the framework must either be described in more detail, or the code be made available. In the current state, the framework can't be reproduced from the descriptions in the manuscript (especially instrument specific aspects). Also it might help to give the framework a name, so that it can be referred to when used in the future.

- Further, the motivation of the sensitivity study and its relevance for the other parts of the paper is unclear to me. Quantifying uncertainties in radar reflectivity from varying PSD parameters, PSD models and particle shape and orientation has been done before in different studies as far as I'm aware. Maybe including a literature review in the introduction on this topic might be helpful to understand the importance of this step to the study?

- I recommend reworking the description of the forward modeling framework. While Fig. 1 shows the "sub module structure", the figure is not described that well in the text. I think reworking section 2.1 with a step by step description of Fig. 1 could solve the issue. I am not sure if "submodule" is the right term describing the framework. Maybe the authors mean working steps?

- The authors compare forward simulation results using "conventional" vs "optimized" settings. I am unsure what "conventional" and "optimized" refer to.

Does conventional mean "typically included in radar simulators"? Also "optimized" might not be the best term to use, because it sounds like an optimization algorithm was applied, which is not the case if I understood correctly. (If I am wrong, then the optimization needs to be described more clearly!) It should be stated more explicitly what exactly the terms "conventional" and "optimized" mean and what settings (PSD parameters etc.) were chosen for the case studies. I recommend including e.g. a table listing all settings. Only stating "the PSD parameters were assumed based on the typical empirical values of land stratiform precipitation clouds" (L370) and referencing three studies is not sufficient to me. I would prefer this information to be explicitly stated in the paper rather than having to look up the cited studies and guessing which values were used.

- When introducing models, software etc. the authors often omit citations. This is especially evident in paragraph L43-58 in the introduction, where citations for the discussed radar simulators QuickBeam, SDSU, G-SDSU as well as for WRF-SBM are missing. Further, citations for the described scattering models (Mie, T-matrix) should be included. I noticed some citations listed in the references don't appear in the manuscript. This should definitely be checked and corrected and might explain the missing citations.

- I find the description of the CloudSat and MODIS data that was used lacking. The CloudSat product that was used should be described in more detail and a short overview of CloudSat (resolution, sensitivity) should be included. That could for example be done by adding a new section either after the introduction or after the model overview. Or including 1-2 more sentences in the introduction.

- In section 2.2.1 N and D should be defined.

- What is the bin size of the hydrometeor model?

- For the mass-size parameters "mean" values are used in the study. It should be stated more clearly which literature values are averaged over. I recommend including a sentence like, "For a and b we took literature values from -list of studies- and calculated the mean". The units of a and b as stated in the text and should be included in the figure captions as well.

- Figures 9 and 11: To increase the readability of the figure, the hydrometeor types could be written next to the letters in the subfigures, similar to Fig. 6a,c,e.

**Technical corrections**

Language: I am not a native speaker, so take my comments with some caution.

Often past tense is used in the article, I would recommend switching to present tense. Sometimes incorrect pronouns are used (e.g. line 95: "...each hydrometeor…., which depends on **their** phase" → **its** phase).

- L34: restructure sentence?

- L35: typical → widely used

- L57: seriously → majorly (or omit)

- L62: I recommend starting a new paragraph beginning with "In this study.."

- L74: I think it should be a "," instead of "." This small typo resulted in me having a lot of trouble understanding the sentence.

- L93: then → further

- L115: omit "a" before D0

- L156: caused → formed

- L161: graupel → snow

- L196: which → and

- L174: prefactor a varies between

- L181: actual → in nature

- L192: omit "still"

- L236-238: unclear sentence

- L257-159: restructure sentence?

- L310: omit "mainly"

- L320: appeared → becomes significant?

- L323-234: unclear sentence

- L488: They → it?

- Figures 2, 4, 6: Optionally, the different y axis scales of the subplots could be noted to avoid confusion.

---

## Referee Comment (RC2)

This is a review of Kou et al., "Simulation and sensitivity analysis for cloud and precipitation measurements via spaceborne millimeter wave radar". The authors evaluate the sensitivity of a forward model for radar reflectivity to its microphysical input variables. The forward model includes cloud ice and water, melting mixed-phase precipitation, snow, graupel and rain. They then perform comparisons of reflectivities that are forward modeled for two WRF simulations (one stratiform and one convective event) against CloudSat observations of the same events. They find in particular that including radar attenuation in the forward model gives improved results over a forward model without attenuation.

Overall, this seems to be a concise and well-executed study. The conclusion that including attenuation in the forward model is necessary for reproducing W-band reflectivities in precipitation that includes melting and liquid phases is not surprising. Perhaps more surprising is that such good agreement was achieved in the comparisons of reflectivities between the WRF simulations and the CloudSat observations. It appears that the model fields were selected from the exact groundtrack and the exact time of the CloudSat overpass. It's unusual, I think, for model features, particularly precipitation, to be so well-located with the observations.

I think the study is a useful contribution to the precipitation retrieval literature. My overall comments relate to how the sensitivity perturbations were defined and how the WRF simulations were configured. Most significant is whether the assessment of uncertainties due to particle shape and orientation are sufficient. I'd like to see this addressed in revision. My specific comments are more extensive and touch mainly on unclear language and missing details. Because they are extensive, I am calling the necessary revisions "major".

I think that with revision, the paper can be acceptable for publication. For this current revision, the scientific significance is good, the scientific quality is fair but this is difficult to judge due to the presentation. The presentation quality is fair.

Overall comments:
#################

Perturbations to PSD parameters:
================================

No explanation of how the assumed perturbations in parameters were determined, or what sources were used to justify the assumptions. E. g., Line 284 "According to the range in b, the standard deviation (SD) as assumed to be 0.5 and 0.3 for snow and graupel".

Overall, the explanations of the sensitivity analysis falls short, particularly as related to PSD perturbations. For example, when "a" is increased, does the reflectivity increase because of the resulting change in the scattering properties of individual particles, or is it because the ice water content increased? When "a" was increased, was $N_w$ decreased so that the ice water content was unchanged? The same sort of concern applies to the evaluation of sensitivities for other PSD parameters.

I suggest also looking at Wood and L'Ecuyer, 2021, AMT. How do your sensitivity results compare to their conclusions about sources of uncertainty in retrieved snowfall?

WRF model simulations:
======================

Details of how the model simulations were performed are lacking. Sufficient details should be provided to reproduce the simulations. In particular, information about the microphysical parameterizations should be provided, but also other details such as nested domain sizes, positions, time steps, vertical gridding should be included. This information could be provided in an appendix.

Particle shape and orientation:
================================

The authors rely on a set of "soft" (mixture of ice and air) particle shapes to evaluated sensitivities to particle shape and orientation. The shapes are spheres and spheroids for snow and rain, and cylinders are additionally included for cloud ice. The T-matrix method is used to calculate scattering properties. It is very unlikely that soft ice spheres and spheroids provide adequate results for evaluating sensitivities to particle shape and orientation for snow or cloud ice at W-band. More realistic variations in particle shapes (e. g., Wood et al., 2015, JAMC) using the discrete dipole approximation for scattering properties can give backscatter cross-sections that vary by a couple of orders of magnitude at larger particle sizes. This seems inconsistent with the results in Figure 6. I request that the authors look at more realistic backscattering cross-sections, particularly for snow, and reexamine their conclusions. These backscattering properties are readily available, from either the Liu database or OpenSSP described in my specific comments beolow, for example.

Specific comments:
##################

L 36-37: I'm not sure why you would say that the CPR is the "most typical spaceborne radar". There are and have been several other spaceborne radars, none of which are cloud radars like the CPR, but rather precipitation radars.

L 41-42: I also don't understand here why you would say "comprehensive view" and "fully detecting clouds and associated precipitation". There are numerous limitations in terms of spatio-temporal sampling and in measurement capabilities that make the CPR observations incomplete.

L 43-45: How does initiating research demonstrate detection capability?

L 53-54: Note that "GPM" is the acronym for the project. The relevant instrument is the "Dual-frequency Precipitation Radar", "DPR".

L 56-57: The seriousness of the effects of particle shape and orientation depend very much on radar wavelength. The effects on Ka- and Ku-band radars like the DPR are much less than those on the W-band CPR.

L 57-58: It is not clear what is meant by "density of mixed particles" here. Does "density" mean the particle concentration, or the actual bulk density of individual particles? Does "mixed particles" mean "mixed-phase particles"? And how does this "density of mixed particles" impact PSD? Where are your citations for these statements?

L 65: What does "optimization physical parameter settings" mean?

L 73-246: There is a significant omission of citations to relevant reference material throughout this section. Please examine this section and add citations to appropriate references to support the assumptions you have made.

L 77-78: What makes the cases you selected "typical"? Were the cases really selected by going through the historical CloudSat data? Were there any other criteria? Why did you choose the particular cases presented in sections 4.1 and 4.2?

L 79-80: How were the WRF simulation results verified by observation data? The validation of the model results probably deserves a section of its own.

L 77-85: This is a very cursory description of the methodology for the simulations. It is missing many relevant details about the setup of the model. What microphysics parameterization was used?

L 97-98: Because contact freezing is essentially instantaneous, I think graupel are usually considered to be ice-air mixtures unless they fall below the freezing level and begin melting.

L 100-101: Is there a reasons to use Maxwell-Garnett rather than something like a three-component Bruggeman model (e.g., Haynes et al. 2009, JGR Atmosphere). Also, note that it is "Garnett" rather than "Garnet".

L 109: I think this formula is correct only if mu=0. See, e.g., Chase et al. (2020, Atmosphere), equations 7 and 9.

L 114: More correctly, R_gas is the specific gas constant. If you are using the R_gas for dry air and T is the air temperature, this formula is not correct.

L 127-129: Was there a reason for using the normalized gamma distribution rather than making the more common assumption of a negative exponential distribution?

L 131-133: Cloud ice particle habit also depends on the amount of supersaturation in the environment where the particle forms and grows. The more common term for "collision and merging" is "aggregation".

L 133-134: Is the Liu database relevant to this work? Was it used in some way? The next sentence states that T-matrix calculations were used, not the Liu database scattering properties.

L 136: I don't find this Hogan et al. citation in the bibliography. How was D defined for these ice particles? Is it an equivolume ice diameter?

L 137: The Fu, 1996 reference cited here is not in the bibliography. Please include it. Were these circular cylinders or hexagonal cylinders? Can you also provide a reference that describes the T-matrix method or code that was applied?

L 139: I'm not sure what point this sentence is making. Perhaps try to state it more clearly. To me, it seems the distribution of orientations is an inherent part of the "falling behavior".

L 142: I don't believe that cloud ice size distributions are considered similar to those of raindrops. Do you have references that suggest an exponential distribution is appropriate for cloud ice?

L 144: Same comment as I made above regarding L 109.

L 156: The term "aggregation" is more typically used, rather than "conglomeration".

L 157-159: Be cautious about using the terms "typically" or "normally", here and in other places in the paper. Is it reasonable to say that some value is

typical or normal when only one or two supporting citations are provided?

L 162-168:  How is D defined for the snowflakes?  The long axis of the assumed spheroid or the equivolume spherical ice diameter?

L 171:  The correct name for the second citation is "von Lerber et al."

L 177-178:  OK, this describes the "D" for the mass and density relations, but it still isn't clear what diameter was used.

L 188:  What is "mass water fraction"?  Most of the mass of a graupel particle is due to water (in the form of ice) so the mass fraction of that water will almost always be near 1.0 since the mass of air in the graupel particle is very small.

L 200-203:  Are you also ignoring aggregation and collision-coalescence?

L 209-211:  But the exponent "b" changes as the particle melts and the shape of the particle melts, does it not?  In the end, when the particle is fully melted, and nearly spherical, the value of "b" should be near 3.  Can you justify using b=2.1 over the full range of particle melting?

L221:  Should the left hand side of equation 19 be "$N_w(D_w)$"?

L 235:  Usually the term "extinction cross-section" is used.

L 239-241:  And how were the attenuation and the two-way path integrated attenuation addressed when combining different types of hydrometeors?

L 247-333:  This is a general comment for the sensitivity section.  Wood and L'Ecuyer (2021, AMT) looked at W-band retrieval uncertainty sources.  How do your results compare with theirs?

L 258:  It's probably more correct to say that the particles are small "compared to the radar wavelength".

L 259-261:  Check grammar/sentence structure.

L 259-264:  Were these sensitivities calculated by perturbing $D_0$ while simultaneously keeping W constant?  Or did W increase as $D_0$ was increased?

L 265:  Please check this equation reference.  I think it is not correct.

L 278:  This isn't the correct equation to convert $N_0$ to $dB(N_0)$.  "dB" indicates "decibel" (i.e., "deci" "Bel", or one-tenth of a Bel).  $dB(N_0)$ should be $10*\log10(N_0)$.

L 280-282:  This states "may result in an uncertainty of approximately 45% and 30% for snow and graupel", but it doesn't say what property of the snow and graupel this uncertainty applies to.   Please clarify.

L 296-299:  Please recheck your values for $D_0$.  20 mm and 30 mm seems extremely large for liquid cloud droplets.  Either there is a typographic error here, or an error in the calculation of $D_0$, I think.

L 297-299:  I don't think there is much gained by including the results from the Gamma($D_0=30$) case.  Clearly, if two PSDs for liquid water droplets are nearly the same, the simulated reflectivities will be nearly the same.  The significant point here is that, given the same water content, different assumptions about the shape of the PSD can have a strong effect on the simulated reflectivity.

L 300: It's probably more correct to say the "reflectivity change" was 4.5 dB.

L 314-334: I think it's questionable whether these different shapes of soft (mixtures of ice and air) particle shapes give a good representation of the sensitivity of reflectivity to particle shape and orientation. Methods such as the discrete dipole approximation are accepted as giving much more realistic values for backscattering by ice and snow particles. I think it would be appropriate to look at other sources of DDA backscattering properties (e.g. the Liu database mentioned earlier, or OpenSSP) to see if your results are consistent with DDA results.

L 345-346: What was the vertical grid spacing? Was the spacing uniform or stretched (with layers getting generally thicker with height)? What data were used for initial and boundary conditions? What time-stepping was used? What microphysical parameterizations were used?

L 347: It's probably more correct to say "interior domains" rather than "internal layers".

L 359-360: I'm not sure it's accurate to say that WRF "accurately simulated the cloud system" based only on comparisons of cloud fraction and cloud top temperature.

L 362-363: I'm not sure that choosing to use the WRF results along the CloudSat track is an effective way to do comparisons between models and satellite observations. One of the frequent errors in models is features like clouds and precipitation may not be located precisely in the location of interest at a particular time. As an example, modeled fronts and their associated precipitation may propagate more slowly or more rapidly than the observed preciptation. Perhaps a better approach would be to statistically compare the properties of the modeled versus the observed clouds and precipitation, using model results from the area under *and near* the CloudSat ground track.

L 367-369: Regarding "Snow is widely distributed...", can you provide more details? Are you talking about the horizontal extent, the vertical extent, or something else. Maybe something like "The vertical extent of snow is widely distributed...". Same comment with respect to the next statement, which is about rain. "Rich" is not a clear description. Do you just mean to say the the total water contents for cloud water, snow and rain were large?

L 372: The Yin et al. (2017) work cited here does not appear in the bibliography.

L 375-376: See my comment regarding the Figure 9 caption (L 724). It's probably more clear to refer to these as "unattenuated reflectivities", "attenuation" (is this one-way or two-way?), and "attenuated reflectivities".

L 379-381: Suggest using "unattenuated reflectivity" and "attenuated reflectivity". Also rather than the "end of the melting region", use "below the melting layer" or "below the melting level." Also suggest using "with attenuation" and "without attenuation" rather than "after attenuation" and "before attenuation". "Before" and "after" can have misleading implications when talking about a radar beam propagating downward through the atmosphere. Finally, there is a well-known reference to this behavior in W-band radar observations from space. See Sassen et al. (2007, Geophysical Research Letters).

L 382-383: Did you demonstrate this, or did you mean to cite existing work? How large does this diameter need to be, and how is this relevant to the

bright band discussion?  If I look at figures 6 and 7, the backscatter cross-sections for the larger particles do not appear to be stable.

L 396:  Usually just "bright band".

L 399:  Rather than using different names for this feature ("brightness band", "bright band",  "strong echo band"), please choose one name and use it consistently.  Also, when you say the reflectivity was stronger, what are you comparing to?

L 400-401:  How did you calculate this relative error?  Relative errors shouldn't be calculated using "dB" values (i.e. (dB_test - dB_true)/dB_true. The values should be converted back to linear units (e.g., mm^6 m^-3), then the relative error calculated.

L 407-408:  See my earlier comment concerning the modeling of the stratiform case.  Additional details about the model configuration would be interesting to see.  What was used for convective parameterization?

L 416:  See earlier comment concerning "rich" and "widely distributed".  Also, I would suggest that when discussing results that involve vertical profiles of data, don't use the terms "high" and "low" to describe data values.  Instead, use "large" and "small".

L 419-420:  Snow and graupel are not mixed-phase particles unless they are melting.  What is the meaning of "components of snow and graupel were complex"?

L 421:  See my earlier comment about the missing Yin et al. (2017) reference.

L 425-428:  Does this mean that the assumed rime mass fraction, and therefore the adjustment factor "f"  was uniform with height for each simulated profile (since liquid water path is a column variable)?  Also, are you saying that you treated the rime mass fraction as liquid water for the purpose of refractive index calculations?

L 429:  But you adjusted the PSD so that the water contents in the simulated profile matched the water contents output by the model, yes?

L 451-452:  See my earlier comment about computing relative error with reflectivities.

L 463:  You mean the sensitivity of reflectivity to $D_0$?  When describing sensitivities, try to express them as "the sensitivity of x to changes in y" so that the meaning is clear.

L 463-467:  You mean the sensitivity of reflectivity to $D_0$?  When describing sensitivities, try to express them as "the sensitivity of x to changes in Y" so that the meaning is clear.  Yes, by imposing the empirical constraints on the PSD, the PSD itself has few degrees of freedom compared to a PSD with independent variations in parameters, so the PSD is less variable.  Is this an unexpected result?  Finally see my earlier comment about computing relative changes in reflectivity - be sure these percentages are calculated correctly.

L 467-468:  How does the particle density affect the PSD?  In general the particle density (as defined by the coefficient "a" and exponent "b" of the mass power law) are considered independent of the PSD paramemters.

L 469-472:  But is this sensitivity due to the increase in "a" changing the scattering properties of particles, or is it because the increase in "a" increases the water content for the population of the particles?  These two effects need to be separated, otherwise the influence of the change in "a" is

overestimated.  Also, see my earlier comment about computing fractional sensitivities for reflectivity.

L 480-481:  Relative errors in what?  Also, see my earlier comment about computing fractional errors in reflectivity.

L 724:  It's not clear what is meant by "before", "during", and "after" attenuation.  I'm guessing that panel (f) shows the unattenuated reflectivities, panel (g) shows the attenuation (is this one-way or two-way?), and panel (h) shows attenuated reflectivities.  Is that correct?

---

## Author Comment (AC1)

**Response to comments by reviewers 1#:**

Thanks very much for your careful reviewing. We will benefit impressively from your suggestions about writing and technique details.

After carefully reading the revised opinions, we have made targeted revisions after discussion. Specific revisions against each point are explained as follows. All the changed contents are highlighted in blue in the revised manuscript.

**Specific comments:**

**● The motivation of the study is not stated clearly to me. I think the aim of the authors is to present their forward modeling framework. If that is the case, the framework must either be described in more detail, or the code be made available. In the current state, the framework can't be reproduced from the descriptions in the manuscript (especially instrument specific aspects). Also it might help to give the framework a name, so that it can be referred to when used in the future.**

Response: Thanks very much for the valuable suggestion. Sorry for the unclear description. The focus is to present the W-band radar reflectivity uncertainty caused by cloud precipitation microphysical parameters and guide appropriate parameter settings in the forward modeling. We have rewritten the research objective.

"The radar reflectivity for the W-band is also sensitive to microphysical parameters like the particle size distribution (PSD) model and parameter, particle shape, orientation, and mass (Mason et al., 2019; Sy et al., 2020; Wood et al., 2013; Wood et al., 2015). A sensitivity analysis is essential for estimating the effects of these uncertainties on simulated radar reflectivity, and guiding appropriate parameter setting in forward modeling.

China has also begun its own spaceborne millimeter wave radar project. The National Satellite Meteorological Center plans to launch a cloud-detecting satellite, whose main load will be the cloud profiling radar (Wu et al., 2018). For development of spaceborne cloud radar, simulation research on cloud and precipitation detection can provide important theoretical support for the design and performance analysis of the system.

In this study, we quantify the uncertainty of different physical model parameters for hydrometeors contributing to radar reflectivity uncertainty via a sensitivity analysis, and present radar reflectivity simulations with optimal parameter settings, based on forward modeling for spaceborne millimeter wave (94 GHz, W-band) radar. Parameters included the particle size distribution (PSD) parameters, PSD model, particle density parameters, shape, and orientation. Using appropriate physical parameter settings, we present and compare the simulation results of two typical cloud precipitation scenarios with measured CloudSat results. Based on a sensitivity analysis

of typical cloud parameters, and a demonstration of cloud precipitation cases, we show the radar reflectivity uncertainty caused by the physical modeling of hydrometeors while emphasizing the importance of assuming more realistic scattering characteristics, as well as appropriate density relations and PSD parameters corresponding to different cloud precipitation types."

**● Further, the motivation of the sensitivity study and its relevance for the other parts of the paper is unclear to me. Quantifying uncertainties in radar reflectivity from varying PSD parameters, PSD models and particle shape and orientation has been done before in different studies as far as I'm aware. Maybe including a literature review in the introduction on this topic might be helpful to understand the importance of this step to the study?**

Response: Thanks very much for the valuable suggestion. We have added more details about the importance of sensitivity analysis in the introduction.

"However, the particle shape, composition, orientation, and mass relation all affect the scattering characteristics. The radar reflectivity for the W-band is also sensitive to microphysical parameters like the particle size distribution (PSD) model and parameter, particle shape, orientation, and mass (Mason et al., 2019; Sy et al., 2020; Wood et al., 2013; Wood et al., 2015). A sensitivity analysis is essential for estimating the effects of these uncertainties on simulated radar reflectivity, and guiding appropriate parameter setting in forward modeling."

**● I recommend reworking the description of the forward modeling framework. While Fig. 1 shows the "sub module structure", the figure is not described that well in the text. I think reworking section 2.1 with a step by step description of Fig. 1 could solve the issue. I am not sure if "submodule" is the right term describing the framework. Maybe the authors mean working steps?**

Response: Thanks very much for the suggestion. Sub module here refers to the function module (similar to working step) in the forward framework. We have added more details and rewritten the section 2.1 with a step by step description of Fig. 1.

"The key points of each sub module are described as follows.

1) From CloudSat historical data and typical weather processes we obtained the cloud precipitation scene cases. According to the occurrence area and time, the corresponding National Center of Environmental Prediction Final (NCEP FNL) reanalysis data were obtained as the initial field in the WRF model.

2) The WRF model was used to simulate the distribution of all types of hydrometeors in these cases. In this research, we use version 4.1.2 of the advanced research WRF model (Skamarock et al., 2019). The WRF simulation results were then validated by using the real observation data.

3) Based on the hydrometeor mixing ratio of the WRF output and assuming certain microphysical parameters based on empirical information obtained from a large amount of observation data, the PSD of the hydrometeor particles were modeled.

4) The complex reflective index of different hydrometeors was calculated according to the particle phase and temperature. The scattering and attenuation characteristics of the hydrometeor particles were then calculated using the T-matrix method (Mishchenko and Travis, 1998). Meanwhile, the absorption coefficients of the atmospheric molecules, such as the water vapor and oxygen, were calculated based on the Liebe attenuation model.

5) The radar reflectivity factor was then calculated based on the atmospheric radiation transmission process and the scattering and attenuation coefficients of hydrometeors.

6) Through coupling with the instrument and platform parameters, the radar echo signal was calculated using the radar equation.

7) During the simulation process, the sensitivity analysis of typical cloud physical parameters was performed to guide the optimal microphysical modeling of the hydrometeors.

8) Finally, the simulation results were compared with observation data, such as CloudSat data, to validate the forward simulations."

● **The authors compare forward simulation results using "conventional" vs "optimized" settings. I am unsure what "conventional" and "optimized" refer to. Does conventional mean "typically included in radar simulators"? Also "optimized" might not be the best term to use, because it sounds like an optimization algorithm was applied, which is not the case if I understood correctly. (If I am wrong, then the optimization needs to be described more clearly!) It should be stated more explicitly what exactly the terms "conventional" and "optimized" mean and what settings (PSD parameters etc.) were chosen for the case studies. I recommend including e.g. a table listing all settings. Only stating "the PSD parameters were assumed based on the typical empirical values of land stratiform precipitation clouds" (L370) and referencing three studies is not sufficient to me. I would prefer this information to be explicitly stated in the paper rather than having to look up the cited studies and guessing which values were used.**

Response: Sorry for the unclear description. Thanks very much for your valuable suggestion. The word of "optimized" is ambiguous. The "optimized" has been modified to "improved". Also, we have added detailed PSD parameter information (the values of used PSD parameters) at L370 in the revised manuscript.

"For the stratiform case, the PSD parameters were assumed based on the empirical values of land stratiform precipitation clouds (Mason, 1971; Niu and He, 1995; Yin et al., 2011), in which the $D_0$ of cloud water was set to 0.01 mm, the $D_0$ of cloud ice was 0.02 mm, and $\mu$ was set as a constant of 1. As snow in stratiform clouds were mainly unrimed particles (Yin et al., 2017), a mass-power relation representative $m=0.0075D^{2.05}$ of unrimed snow (Moisseev et al., 2017) was used in the

simulation, where *D* was the volume equivalent diameter. In addition, a melting layer model with a width of 1 km was assumed below 0 ℃ and the PSD parameters of the raindrops were calculated according to the melting model.

The main difference between the conventional and improved setting is that the conventional setting does not consider the melting model, and the PSD parameters for rain were set as $D_0$=1 mm, $\mu$=3."

A table listing all settings was not included, because most of parameters are the same. The main difference between the conventional and improved setting was the application of the melting model (the PSD parameters of rain were calculated from melting model in the improved setting, and the PSD parameters of rain were assumed according to the experience values in the conventional setting), and the varying mass relations for snow and graupel in the convective cloud.

● **When introducing models, software etc. the authors often omit citations. This is especially evident in paragraph L43-58 in the introduction, where citations for the discussed radar simulators QuickBeam, SDSU, G-SDSU as well as for WRF-SBM are missing. Further, citations for the described scattering models (Mie, T-matrix) should be included. I noticed some citations listed in the references don't appear in the manuscript. This should definitely be checked and corrected and might explain the missing citations.**

Response: I am very sorry for the carelessness. Thanks very much for pointing out. The citations in paragraph L43-58 and scattering models have been added now in the revised manuscript. Also, we have checked the references and citations one by one throughout this manuscript.

"In the design of the observation system and interpretation of cloud and precipitation observation data, forward modeling and simulation play a highly important role (Horie et al., 2012; Lamer et al., 2021; Leinonen et al., 2015; Marra et al., 2013; Sassen et al., 2007; Wang et al., 2019; Wu et al., 2013). QuickBeam is a user-friendly radar simulation package that compares modeled clouds to observations from CloudSat, but it cannot simulate mixed phase particles in the melting state (Haynes et al., 2007). The Satellite Data Simulator Unit (SDSU) developed by Nagoya University, Japan, is a satellite multisensor simulator integrating radar, microwave radiometer, and visible/infrared imager. Goddard Satellite Data Simulator Unite (G-SDSU) is a derivative version of the SDSU (Masunage et al., 2010). In addition to the basic functions of the SDSU, it can be coupled with high-precision National Aeronautics and Space Administration (NASA) atmospheric models, such as the Weather Research and Forecasting-Spectral Bin Microphysics (WRF-SBM). The Global Precipitation Measurement (GPM) satellite simulation is also based on the G-SDSU, which converts the geophysical parameters simulated by the WRF-SBM into observable microwave brightness and equivalent reflectivity factor signals of the GPM (Matsui et al., 2013).

The WRF model was used to simulate the distribution of all types of hydrometeors in these cases. In this research, we use version 4.1.2 of the advanced research WRF model (Skamarock et al., 2019). The scattering and attenuation characteristics of the hydrometeor particles were then calculated using the T-matrix method (Mishchenko and Travis, 1998)."

● **I find the description of the CloudSat and MODIS data that was used lacking. The CloudSat product that was used should be described in more detail and a short overview of CloudSat (resolution, sensitivity) should be included. That could for example be done by adding a new section either after the introduction or after the model overview. Or including 1-2 more sentences in the introduction.**

Response: Thanks very much for your suggestion. We have added the description of CloudSat data in the introduction, and the information of MODIS data in section 4.1.1.

"The CPR is a W-band, nadir-pointing radar system, with a minimum detectable signal of approximately -29 dBZ. The CPR footprint size is 1.4 km across-track and 2.5 km along-track, and the vertical resolution is approximately 500 m (Stephens et al., 2008).

The level-2 cloud product of cloud top temperature from MODIS with spatial resolutions of 5 km was used in the comparison."

● In section 2.2.1 N and D should be defined.

Response: The definition of N and D has been added now.

"where $N(D)$ is the particle size distribution, $D$ is the volume equivalent diameter, $N_w$ is the normalized intercept parameter, $D_0$ is the median volume diameter, $\rho_w$ is the density of water, i.e. 1 g/cm$^3$, $\mu$ is the shape parameter, and $\Gamma$ is the gamma function."

● **What is the bin size of the hydrometeor model?**

Response: The bin size for different hydrometeor is different, for example, the bin size for rain is 0.1 mm, and that for cloud water is 0.01 mm.

● **For the mass-size parameters "mean" values are used in the study. It should be stated more clearly which literature values are averaged over. I recommend including a sentence like, "For a and b we took literature values from -list of studies- and calculated the mean". The units of a and b as stated in the text and should be included in the figure captions as well.**

Response: Thanks very much for the valuable suggestion. The relevant information has been added in Figure 4.

[Figure]

Figure 4: Impact of PSD parameters on radar reflectivity for snow and graupel. Variation in reflectivity for snow at (a) $dBN_0$ values of 3, 3.5, 4, 4.5, and 5 with a mean mass-diameter relationship of $m = 0.009D^{2.1}$, where $D$ is in cm and $m$ is in g; (b) prefactor $a$ in mass-diameter relationship of 0.005, 0.007, 0.009, 0.011, and 0.013 $g/cm^b$, with exponent $b$ of 2.1 and $N_0$ assumed to be $3 \times 10^3$ $m^{-3}$ $mm^{-1}$; (c) mean value $\pm$ standard deviation of $b$, where the mean is 2.1 and standard deviation (SD) is 0.28, with $a$ assumed to be 0.009. The vertical bars represent the SD of the reflectivity change caused by deviation from the mean value of $b$. Variation in reflectivity for graupel at (d) $dBN_0$ values of 3, 3.5, 4, 4.5, and 5 with a mean mass-diameter relationship of $m = 0.04D^{2.6}$, where $D$ is in cm and $m$ is in g; (e) prefactor $a$ in mass-diameter relationship of 0.02, 0.03, 0.04, 0.05, and 0.06 $g/cm^b$, with exponent $b$ of 2.6 and $N_0$ assumed to be $4 \times 10^3$ $m^{-3}$ $mm^{-1}$; and (f) mean value $\pm$ standard deviation of $b$, where the mean is 2.6 and standard deviation is 0.16, with $a$ assumed to be 0.04. The value range in $a$ and $b$ and the mean value are obtained from literatures, and the standard deviation of $b$ are calculated according to the range and average of Gaussian distribution.

● **Figures 9 and 11: To increase the readability of the figure, the hydrometeor types could be written next to the letters in the subfigures, similar to Fig. 6a,c,e.**

Response: Thanks very much for the suggestion. The hydrometeor types have been added in the subfigures of Figure 9 and 11.

[Figure]

Figure 9: Latitude-height cross-section of the hydrometeor for the stratiform case simulated by the WRF for: (a) cloud water, (b) cloud ice, (c) snow, (d) rain, and (e) total hydrometeors. (f) Simulated unattenuated radar reflectivity with the total hydrometeors, (g) two-way attenuation, and (h) attenuated radar reflectivity. Owing to the Mie scattering effect, the unattenuated radar reflectivity did not decrease markedly at the bottom of the melting layer, whereas the bright band at the melting layer was highlighted due to strong attenuation in the rain region.

[Figure]

Figure 11: Latitude-height cross-section of the hydrometeor of the convective case simulated by the WRF for: (a) total hydrometeors, (b) cloud water, (c) cloud ice, (d) snow, (e) graupel, (f) rain. (g) Simulated unattenuated radar reflectivity with the total hydrometeors, (h) two-way attenuation, and (i) attenuated radar reflectivity.

**Technical corrections**

● **L34: restructure sentence?**

Response: Thanks for your suggestion. The sentence has been restructured. "The cloud radar platform mainly includes spaceborne, airborne, and ground-based radars."

● **L35: typical → widely used**

● **L57: seriously → majorly (or omit)**

● **L62: I recommend starting a new paragraph beginning with "In this study.."**

● **L74: I think it should be a "," instead of "." This small typo resulted in me having a lot of trouble understanding the sentence.**

● **L93: then → further**

● **L115: omit "a" before D0**

● **L156: caused → formed**

● **L161: graupel → snow**

● **L196: which → and**

● **L174: prefactor a varies between**

● **L181: actual → in nature**

● **L192: omit "still"**

● **L310: omit "mainly"**

Response: Sorry for the poor writing. Thanks for your suggestions. The corresponding expressions and sentences have been modified in the revised manuscript.

● **L236-238: unclear sentence**

Response: Thanks for your suggestions. The sentence has been rewritten. "During radar reflectivity calculation, a look-up table of backscattering and extinction cross-sections is established for reducing the calculation workload."

● **L257-259: restructure sentence?**

Response: Thanks for your suggestion. The sentence has been rewritten. "Figure 2 shows the radar reflectivity change with variations in the gamma PSD parameters for cloud water and rain. Cloud water particles are small compared to the radar wavelength, which is in the linear growth stage in the Mie scattering region."

● **L320: appeared → becomes significant?**

● **L488: They → it?**

Response: Thanks for the suggestion. "appeared" has been modified to "becomes significant". "They" has been modified to "it".

● **Figures 2, 4, 6: Optionally, the different y axis scales of the subplots could be noted to avoid confusion.**

Response: Thanks for the suggestion. The $y$ axis in Figure 2 and 4 are for radar reflectivity, and the $y$ axis scales are different for different hydrometeors. The $y$ axes of the subplots in Figure 6 are for backscattering cross-section and corresponding radar reflectivity, and the $y$ axis scales have been noted.

Thanks so much for helping us with the English. To edit the text further, we have paid another commercial editing service to polish our manuscript for the language. We would like to thank the reviewer for his/her significant effort to suggest changes for our manuscript.
* * *
Special thanks to the reviewer for the good comments and his/her patience.

---

## Author Comment (AC2)

**Response to comments by reviewers 2#:**

We really appreciate you for your carefulness and conscientiousness. We will benefit impressively from your suggestions about writing and technique details.

    After carefully reading the revised opinions, we have made all the changes suggested by the reviewer and addressed all the comments in the notes below. All the changed contents are highlighted in blue in the revised manuscript.

**Overall comments:**

**Perturbations to PSD parameters:**

1) **No explanation of how the assumed perturbations in parameters were determined, or what sources were used to justify the assumptions. E. g., Line 284 "According to the range in b, the standard deviation (SD) as assumed to be 0.5 and 0.3 for snow and graupel".**

Response: Thanks very much for the valuable comment. The relevant information has been added, and the SD of $b$ has been recalculated.

    "According to results from observation experiments reported in the literatures, the exponent $b$ for snow varies from 1.4 to 2.8, and most of the mass relations have the mean value of $b$ close to 2.1(Brandes et al., 2007; Heymsfield et al., 2010; Huang et al., 2019; Sy et al., 2020; Szyrmer and Zawadzki, 2010; Tiira et al., 2016; Wood et al., 2013). For graupel, the exponent $b$ varies from 2.1 to 3 (Heymsfield et al., 2018; Mason et al., 2018; Von Lerber et al., 2017), with a mean value of approximately 2.6. Based on the range and mean value of $b$ for the Gaussian distribution, we calculated the standard deviation (SD) to be 0.28 and 0.16 for snow and graupel, respectively."

2) **Overall, the explanations of the sensitivity analysis falls short, particularly as related to PSD perturbations. For example, when "a" is increased, does the reflectivity increase because of the resulting change in the scattering properties of individual particles, or is it because the ice water content increased? When "a" was increased, was N_w decreased so that the ice water content was unchanged? The same sort of concern applies to the evaluation of sensitivities for other PSD parameters.**

Response: Thanks very much for your valuable suggestion. when "a" is increased, the reflectivity increases mainly due to the change in the scattering properties. The PSD was slightly changed to keep the water content unchanged. The part of the sensitivity analysis for PSD perturbations have been rewritten.

    "An exponential PSD with a power-law mass spectrum was used for snow and graupel. Figure 4 shows the effects of intercept parameter $N_0$ and the mass power-law parameters of prefactor $a$ and exponent $b$. With the mean mass-size relationships for snow and graupel, changing the $dBN_0$ ($dBN_0 = \log_{10}(N_0)$) from 3 to 5 could cause a reflectivity increase of approximately 7–8 dB, as shown in Fig. 4a and d. The mass power-law parameters vary with snow/graupel type, shape, and porosity. In

Fig. 4b and e, we see that with a constant $N_0$ and mean value of exponent $b$, the reflectivity change caused by variation in prefactor $a$ from 0.005 to 0.013 g/cm$^b$ for snow and 0.02 to 0.06 g/cm$^b$ for graupel ($W$ remains constant) can reach 7–10 dB. An increase in $a$ lead to an increase in the corresponding particle scattering properties. The intercept parameter $N_0$ will slightly decrease with the increase in $a$ implicitly representing the effects of aggregation at warmer temperatures (Woods et al., 2008). Among them, the change to particle scattering properties caused by the perturbation of $a$ play a dominant role in the reflectivity change. Using an average mass-power relation assumption, the variation in $a$ as a result of the degree of aggregation and riming, and particle shapes may result in the reflectivity uncertainty of approximately 45 % and 30 % for snow and graupel, respectively. For analyzing the effect of the variation in $b$, a Gaussian distribution of $b$ was modeled. According to results from observation experiments reported in the literatures, the exponent $b$ for snow varies from 1.4 to 2.8, and most of the mass relations have the mean value of $b$ close to 2.1(Brandes et al., 2007; Heymsfield et al., 2010; Huang et al., 2019; Sy et al., 2020; Szyrmer and Zawadzki, 2010; Tiira et al., 2016; Wood et al., 2013). For graupel, the exponent $b$ varies from 2.1 to 3 (Heymsfield et al., 2018; Mason et al., 2018; Von Lerber et al., 2017), with a mean value of approximately 2.6. Based on the range and mean value of $b$ for the Gaussian distribution, we calculated the standard deviation (SD) to be 0.28 and 0.16 for snow and graupel, respectively. The error bars in Fig. 4c and f represent the SD of the reflectivity change caused by variation in $b$, which was approximately 2 dB for snow and 0.5 dB for graupel. The results showed that the sensitivity of reflectivity to prefactor $a$ was substantially greater than that of exponent $b$. In all, the mass relationships that depend on particle habits and formation mechanisms, cause substantial uncertainties in W-band radar reflectivity. Our results are consistent with the sensitivity analysis by Wood and L'Ecuyer (2021) who pointed out that the W-band radar reflectivity uncertainty for snowfall was dominated by the particle model parameter (e.g., the prefactors and exponents of the mass relationships). The mass relationship can cause the reflectivity uncertainty of several to more than 10 dB. The results indicate that improved constraints on assumed particle mass models would improve forward-modeled radar reflectivity and physical parameter retrieval."

**3) I suggest also looking at Wood and L'Ecuyer, 2021, AMT. How do your sensitivity results compare to their conclusions about sources of uncertainty in retrieved snowfall?**

Response: Thank very much for the valuable suggestion. We have carefully read the literature (Wood and L'Ecuyer, 2021), and compared our results with those in Wood and L'Ecuyer (2021). Wood and L'Ecuyer (2021) showed that the contributions to uncertainties in W-band radar reflectivity from the particle model parameters (e.g., the coefficients and exponents of the mass relationships) was most substantial, which may cause 5 to 15 dB reflectivity uncertainty. Our study shows that he reflectivity change caused by variation in prefactor $a$ from 0.005 to 0.013 g/cm$^b$ for snow and 0.02 to 0.06 g/cm$^b$ for graupel ($W$ remains constant) can reach 7–10 dB. The reflectivity change caused by variation in $b$ was approximately 0.5-2 dB. The mass relationships cause substantial uncertainties in W-band radar reflectivity. Our results are consistent with those in Wood

and L'Ecuyer (2021). The comparison with Wood and L'Ecuyer (2021) has been added in the revised manuscript.

"Our results are consistent with the sensitivity analysis by Wood and L'Ecuyer (2021) who pointed out that the W-band radar reflectivity uncertainty for snowfall was dominated by the particle model parameter (e.g., the prefactors and exponents of the mass relationships). This relationship can cause the reflectivity uncertainty of several to more than 10 dB. The results indicate that improved constraints on assumed particle mass models would improve forward-modeled radar reflectivity and physical parameter retrieval."

**WRF model simulations:**

1) **Details of how the model simulations were performed are lacking. Sufficient details should be provided to reproduce the simulations. In particular, information about the microphysical parameterizations should be provided, but also other details such as nested domain sizes, positions, time steps, vertical gridding should be included. This information could be provided in an appendix.**

Response: Thank you very much for the valuable suggestion. Details of WRF model simulations have been added in the Appendix A.

"**Appendix A: Model setup and verification**

**a. Stratiform case**

[revised manuscript text omitted]

**Particle shape and orientation:**

1) **The authors rely on a set of "soft" (mixture of ice and air) particle shapes to evaluated sensitivities to particle shape and orientation. The shapes are spheres and spheroids for snow and rain, and cylinders are additionally included for cloud ice. The T-matrix method is used to calculate scattering properties. It is very unlikely that soft ice spheres and spheroids provide adequate results for evaluating sensitivities to particle shape and orientation for snow or cloud ice at W-band. More realistic variations in particle shapes (e. g., Wood et al., 2015, JAMC) using the discrete dipole approximation for scattering properties can give backscatter cross-sections that vary by a couple of orders of magnitude at larger particle sizes. This seems inconsistent with the results in Figure 6. I request that the authors look at more realistic backscattering cross-sections, particularly for snow, and reexamine their conclusions. These backscattering properties are readily available, from either the Liu database or OpenSSP described in my specific comments below, for example.**

Response: Thanks very much for your valuable suggestion. We have recalculated the scattering properties for cloud ice and dry snow using T matrix and DDA, respectively. Figure 6 has been updated. "

[Figure]

Figure 6: Backscattering cross-section and corresponding radar reflectivity under different shapes for cloud ice, dry

snow, and rain. (a) Comparison of the backscattering cross-sections of ice crystals as spheres, spheroids, or cylinders, where $\delta$ is the SD of the canting angle. (b) Radar reflectivity comparison for particles in (a), where the PSD was assumed as a Gamma distribution constrained by Eqs. (7)–(9), with $\mu$=1 and T = –60º C. (c) Comparison of the backscattering cross-sections for dry snow with spheres and spheroids. (d) Radar reflectivity comparison for particles in (c), where the PSD was assumed as an exponential distribution with $N_0 = 3 \times 10^3$ m$^{-3}$ mm$^{-1}$. (e) Comparison of backscattering cross-sections for raindrops with spheres and spheroids. (f) Radar reflectivity comparison for particles in (e), where the PSD was assumed as a Gamma distribution with $D_0 = 1.25$ mm and $\mu$=3.

The solid and dotted lines in Fig. 6a indicate that the SD of the canting angle ($\delta$) is 2º and 20º, respectively. The backscattering difference for cloud ice was evident between the sphere and non-sphere when the diameter was greater than 1 mm. The radar reflectivity factor in Fig. 6b was obtained with the constrained PSD parameter (section 2.2.3) of $T = –60$ ºC and $\mu$=1, and the maximum diameter was calculated according to Eq. (8) that was within 0.4 mm. Figure 6b shows that the spherical and non-spherical assumption for cloud ice may result in an average reflectivity difference by approximately 8 %. Figure 6c shows the backscattering cross-section of dry snow with a mass-diameter relation of $m$=0.0075$D^{2.05}$ (Matrosov et al., 2007; Moisseev et al., 2017), where the axis ratio of the spheroid was 0.6 and the SD of the canting angle was assumed to be 20º and 40º, respectively. When calculating the radar reflectivity factor, the corresponding exponential distribution parameter was $N_0 = 3 \times 10^3$ m$^{-3}$ mm$^{-1}$ and the reflectivity difference between the sphere and spheroid can reach approximately 1.6 dB."

The results in Matrosov (2007) and Wood et al (2015) showed that the reflectivity difference for dry snow between the sphere and spheroid assumption can reach approximately 2 dB. The magnitude of backscattering cross-section and reflectivity difference between spherical and non-spherical in Fig. 6 are basically consistent with those in Matrosov (2007) and Wood et al (2015). The slight difference is mainly due to the different setting of the SD of the canting angle.

In our study, the scattering characteristics of cloud ice (composed of pure ice) and dry snow (composed of ice and air) are calculated separately, as shown in Fig.6. For comparison, Figure R1 shows the backscattering cross-sections for ice and dry snow with spheres (Rayleigh spheres and Mie spheres) and spheroids (axis ratio of 0.6). The result is consistent with that in Wood et al (2015), but more particle shapes were included in Wood et al (2015). We mainly considered the difference between sphere and spheroid with different orientations in this study. In future research, we will consider the influence of more particle shapes on radar reflectivity.

[Figure]

Figure R1: Backscattering cross-sections for solid ice and dry snow (a mass-diameter relation of $m=0.0075D^{2.05}$ was used for dry now), where blue line represents Rayleigh sphere for solid ice, red line represents Mie sphere for solid ice, orange line represents spheroid with axis ratio of 0.6 for solid ice, purple line represents Mie sphere for dry snow, and green line represents spheroid with axis ratio of 0.6 (SD of canting orientation of 20°) for dry snow.

The backscattering cross-sections from T-matrix and DDA were compared as well (the Liu database were not used, because we cannot access the download link). Figure R2 shows the backscattering cross-sections for dry snow with T-matrix and DDA method. The result shows that the scattering properties from T-matrix is similar to those from DDA if the particles are with similar shape assumption.

[Figure]

Figure R2: Comparison of backscatter cross-sections for dry snow with T-matrix and DDA method, where blue line represents sphere, orange line represents spheroid with DDA algorithm, yellow line represents hexagonal prism with DDA algorithm, purple line represents spheroid with T-matrix algorithm, and green line represents circular cylinder with T-matrix algorithm.

**Specific comments:**

**L 36-37:  I'm not sure why you would say that the CPR is the "most typical spaceborne radar".  There are and have been several other spaceborne radars, none of which are cloud radars like the CPR, but rather precipitation radars.**

Response: Sorry for the inaccurate expression. The sentence has been rewritten.

"The most widely used spaceborne cloud radar is the millimeter wave cloud profiling radar (CPR) carried onboard the CloudSat satellite (Stephens et al., 2008; Tanelli et al., 2008)."

**L 41-42:  I also don't understand here why you would say "comprehensive view" and "fully detecting clouds and associated precipitation". There are numerous limitations in terms of spatio-temporal sampling and in measurement capabilities that make the CPR observations incomplete.**

Response: Sorry for the unclear description. The sentence has been rewritten.

"This provides an opportunity to advance the understanding of the way water cycles through the atmosphere, by jointly observing clouds and associated precipitation (Behrangi et al., 2013; Ellis et al., 2009; Hayden et al., 2018)."

**L 43-45:  How does initiating research demonstrate detection capability?**

Response: Sorry for the inaccurate statement. The sentence has been rewritten.

"Recently, many countries have begun research on next-generation spaceborne cloud radar (Battaglia et al., 2020; Illingworth et al., 2015; Tanelli et al., 2018; Wu et al., 2018), such as the CPR on the EarthCARE satellite and dual-frequency cloud radar on the Aerosol/Clouds/Ecosystem (ACE) mission (Illingworth et al., 2015; Tanelli et al., 2018)."

**L 53-54:  Note that "GPM" is the acronym for the project.  The relevant instrument is the "Dual-frequency Precipitation Radar", "DPR".**

Response: Yes. Thanks for pointing out. The "GPM satellite simulation" has been modified to "Global Precipitation Measurement (GPM) satellite simulator".

**L 56-57:  The seriousness of the effects of particle shape and orientation depend very much on radar wavelength.  The effects on Ka- and Ku-band radars like the DPR are much less than those on the W-band CPR.**

Response: Yes, the effects on Ka- and Ku-band radars like the DPR are much less than those on the W-band CPR. Sorry for the inaccurate expression. Thanks for the comment. The sentence has been rewritten. "The radar reflectivity for the W-band is also sensitive to microphysical parameters like the particle size distribution (PSD) model and parameter, particle shape, orientation, and mass".

**L 57-58:  It is not clear what is meant by "density of mixed particles" here. Does "density" mean the particle concentration, or the actual bulk density of individual particles? Does "mixed particles" mean "mixed-phase particles"? And how does this "density of mixed particles" impact PSD?  Where are your citations for these statements?**

Response: Sorry for the unclear description. The density here refers to the density-diameter relationship of snow/graupel. The relevant sentences have been rewritten.

"However, the particle shape, composition, orientation, and mass relation all affect the scattering characteristics. The radar reflectivity for the W-band is also sensitive to microphysical parameters like the particle size distribution (PSD) model and parameter, particle shape, orientation, and mass (Mason et al., 2019; Sy et al., 2020; Wood et al., 2013; Wood et al., 2015). A sensitivity analysis is essential for estimating the effects of these uncertainties on simulated radar reflectivity, and guiding appropriate parameter setting in forward modeling."

**L 65:   What does "optimization physical parameter settings" mean?**

Response: Sorry for the unclear description. The "optimization" has been modified to "appropriate".

**L 73-246:   There is a significant omission of citations to relevant reference material throughout this section.   Please examine this section and add citations to appropriate references to support the assumptions you have made.**

Response: I am very sorry for the carelessness. Thanks very much for pointing out. The citations in section 2 have been added now in the revised manuscript. Also, we have checked the references and citations one by one throughout this manuscript.

**L 77-78:   What makes the cases you selected "typical"?   Were the cases really selected by going through the historical CloudSat data?   Were there any other criteria?   Why did you choose the particular cases presented in sections 4.1 and 4.2?**

Response: The cases were selected by combining historical CloudSat data and typical weather processes observed on the ground. For example, the stratiform case in section 4.1 was a large-scale low trough cold front cloud system in northwest China. The weather process covered a large area and lasted for a long time. Many Chinese scholars have simulated and studied the weather process (Liu et al., 2015; Sun et al., 2015). The convective case in section 4.2 was a large-scale severe convective weather process that occurred in the Lower Yangtze-Huaihe river on June 23, 2016. This deep convective process caused strong wind, hail and rainstorm in Jiangsu province (Kuang et al., 2018).

Liu, T., Sun, J., Zhou, Y.Q., Peng, C., and Yan, F.: A simulation study on through cold front cloud structure (in Chinese), Meteorological Monthly, 41: 1-13, 2015, doi:10.7519/j.issn.1000-0526.2015.10.006.

Sun, J., Yang, W.X., Zhou, Y.Q.: Numerical simulations of cloud structure and seed ability of a precipitation stratiform in Hebei (in Chinese), Plateau Meteorology, 34: 1699-1710, 2015, doi:10.7522/j.issn.1000-0534.2014.00086.

Kuang, X., Yin, Y., Chen J.H., and Xiao, H.: Simulation analysis of strong convective weather processes in Huanghuai River based on WRF model and CloudSat satellite data (in Chinese). Journal of the Meteorological Sciences, 38:331-341, 2018, doi:10.3969/2017jms.0035.

**L 79-80:   How were the WRF simulation results verified by observation data? The validation**

**of the model results probably deserves a section of its own.**

Response: Thanks very much for your valuable suggestion. The model validation was provided in Section 4.1.1. More validation results have been added in Appendix A. Detailed information are in the response to "overall comments: WRF model simulations".

**L 77-85: This is a very cursory description of the methodology for the simulations. It is missing many relevant details about the setup of the model. What microphysics parameterization was used?**

Response: Thanks very much for the suggestion. Detailed description of the model setup has been added in Appendix A.

"The simulation for this stratiform case was conducted with four nested grids (d01, d02, d03, and d04), and the inner domain was centered at 41.08°N, 117.61°E. The horizontal grid spacings are 27km, 9km, 3km, and 1km and the corresponding grids are 120×120, 180×180, 300×300, and 300×300. The vertical resolution increases with height from approximately 50 m near the surface to 600 m near 50 hPa. Time steps of 180s and 6.67s were used for d01 domain and d04 domain, respectively. The 6-hourly NCEP FNL operational global analysis data on 1° × 1° grids were used to provide the initial and boundary conditions. In term of physical scheme, the model adopted CAM 5.1 5-class scheme, Grell-Freitas cumulus parameterization scheme, RRTM long and short-wave radiation scheme, YSU boundary layer scheme, Monin-Obukhov surface layer scheme and thermal diffusion land surface scheme. The cumulus parameterization scheme was used for d01 and d02 domains only."

**L 97-98: Because contact freezing is essentially instantaneous, I think graupel are usually considered to be ice-air mixtures unless they fall below the freezing level and begin melting.**

Response: Sorry for the incorrect statement. Thank you for pointing out. The sentence has been rewritten. "Dry snow and graupel are a mixture of air and ice, while wet snow and graupel are a mixture of air, ice, and water."

**L 100-101: Is there a reason to use Maxwell-Garnett rather than something like a three-component Bruggeman model (e.g., Haynes et al. 2009, JGR Atmosphere). Also, note that it is "Garnett" rather than "Garnet".**

Response: Bruggeman model can also be used. In this study, we use Maxwell-Garnett model.

Thanks very much for pointing out. "Garnet" has been revised to "Garnett".

**L 109: I think this formula is correct only if mu=0. See, e.g., Chase et al. (2020, Atmosphere), equations 7 and 9.**

Response: Yes, this formula is correct only if μ=0. Thank you very much for pointing out. This formula has been revised to

$$N_w = \frac{W}{\pi \rho_w} \left( \frac{4(3.67 + \mu)}{(4 + \mu) D_0} \right)^4 \tag{2}$$

**L 114: More correctly, R_gas is the specific gas constant. If you are using the R_gas for dry air and T is the air temperature, this formula is not correct.**

Response: Sorry for the incorrect expression. Thank you for pointing out. The formula has been revised.

$$W = \frac{P}{R_g T_V} * 1000 * q ,$$

(3)

where $R_g$ is the specific gas constant, $P$ is the air pressure in hPa, $T_V$ is the virtual temperature in $K$, $q$ is the mixing ratio of the hydrometeor based on the WRF output in kg/kg, and the units of $W$ are g/m$^3$.

**L 127-129: Was there a reason for using the normalized gamma distribution rather than making the more common assumption of a negative exponential distribution?**

Response: In this study, we used the normalized gamma distribution for raindrop and cloud water. For raindrop and cloud water, the three-parameter gamma distribution is more general form of DSD compared with the exponential distribution with two variables (Ryzhkov and Zrnic, 2019; Zhang, 2017).

**L 131-133: Cloud ice particle habit also depends on the amount of supersaturation in the environment where the particle forms and grows. The more common term for "collision and merging" is "aggregation".**

Response: Thanks very much for your suggestion. The sentence has been rewritten. "Cloud ice is mainly composed of various non-spherical ice crystals; the size and shape of ice crystal particles are complex and diverse, depending on the cloud temperature, the degree of supersaturation in the environment where the particle forms and grows, and whether the particles have experienced aggregation processes in the cloud (Heymsfield et al., 2013; Ryzhkov and Zrnic, 2019)."

**L 133-134: Is the Liu database relevant to this work? Was it used in some way? The next sentence states that T-matrix calculations were used, not the Liu database scattering properties.**

Response: The database in Liu (2008) included the scattering characteristics of ice crystals, but it was not used in our study (we cannot access the download link). Here, we used the T-matrix to calculate the scattering properties of hydrometeor particles.

**L 136: I don't find this Hogan et al. citation in the bibliography. How was D defined for these ice particles? Is it an equivolume ice diameter?**

Response: I am very sorry for the careless. Hogan et al. citation has been added now in the bibliography. Here $D$ refers to the larger dimension of cylinder. The $D$ in this formula has been revised to $L$ to be distinguished from the equivalent volume diameter in other formulas.

$$\begin{cases} L/h = 5.068 L^{0.586} & L > 0.2\,mm \\ L/h = 2 & L \leq 0.2\,mm \end{cases}$$

(5)

Hogan, R.T., Tian, L., Brown, P.R.A., Westbrook, C.D., Heymsfield, A.J., and Eastment, J.D.: Radar scattering from ice aggregates using the horizontally aligned oblate spheroid approximation, J. Appl. Meteorol. Climatol., 51, 655-671, doi: 10.1175/JAMC-D-11-074.1, 2012.

**L 137:   The Fu, 1996 reference cited here is not in the bibliography.   Please include it. Were these circular cylinders or hexagonal cylinders?    Can you also provide a reference that describes the T-matrix method or code that was applied?**

Response: I am very sorry for the careless. Thanks very much for pointing out. Fu (1996) has been added in the references. These were circular cylinders. Mishchenko and Travis (1998) has been added to describe the T-matrix method.

Fu, Q.: An accurate parameterization of the solar radiative properties of cirrus clouds for climate models, J. Climate, 9, 2058-2082, doi: 10.1175/1520-0442(1996)009<2058:APPOTS>2.0.CO;2, 1996.

Mishchenko, M.I., and Travis, L.D..: Capabilities and limitations of a current FORTRAN implementation of the T-matrix method for randomly oriented, rotationally symmetric scatterers, J. Quant. Spectrosc. Radiat. Transfer, 60, 309-324, doi: 10.1016/S0022-4073(98)00008-9, 1998.

**L 139:    I'm not sure what point this sentence is making.    Perhaps try to state it more clearly. To me, it seems the distribution of orientations is an inherent part of the "falling behavior".**

Response: Sorry for the unclear statement. This sentence has been rewritten. "Distribution of orientations of ice particles depends on their falling behavior. According to Melnikov and Straka (2013), we assume that the ice crystal orientations follow a Gaussian distribution, with a mean canting angle of 0° and a SD between 2° and 20°."

**L 142:   I don't believe that cloud ice size distributions are considered similar to those of raindrops.   Do you have references that suggest an exponential distribution is appropriate for cloud ice?**

Response: In this study, the normalized gamma distribution was adopted for cloud ice, which was according to the empirical fits derived in Heymsfield et al (2013).

Heymsfield, A. J., Schmitt, C., and Bansemer, A.: Ice cloud particle size distributions and pressure-dependent terminal velocities from in situ observations at temperatures from 0º to -86ºC, J. Atmos. Sci., 70, 4123-4154, doi: 10.1175/JAS-D-12-0124.1, 2013.

**L 144:   Same comment as I made above regarding L 109.**

Response: Thanks very much for pointing out. The formula has been revised.

**L 156:   The term "aggregation" is more typically used, rather than "conglomeration".**

Response: Thanks very much for your suggestion. This sentence has been rewritten. "Snowflakes are usually formed by the aggregation and growth of ice crystals."

**L 157-159:   Be cautious about using the terms "typically" or "normally", here and in other**

**places in the paper. Is it reasonable to say that some value is typical or normal when only one or two supporting citations are provided?**

Response: Thanks for your suggestion. The terms "typically" and "normally" have been removed from the corresponding sentences.

**L 162-168: How is D defined for the snowflakes? The long axis of the assumed spheroid or the equivolume spherical ice diameter?**

Response: $D$ is the volume equivalent diameter. This has been added in the revised manuscript.

**L 171: The correct name for the second citation is "von Lerber et al."**

Response: Thanks very much for pointing out. "von et al" has been revised to "von Lerber et al".

**L 177-178: OK, this describes the "D" for the mass and density relations, but it still isn't clear what diameter was used.**

Response: Sorry for unclear description. Thank you for pointing out. The sentence has been rewritten. "In this study, the diameters in the mass and density relations were converted to the volume equivalent diameter $D$ according to the assumed axis ratio."

**L 188: What is "mass water fraction"? Most of the mass of a graupel particle is due to water (in the form of ice) so the mass fraction of that water will almost always be near 1.0 since the mass of air in the graupel particle is very small.**

Response: Sorry for the inaccurate statement. The mass water fraction is for wet graupel. The sentence has been rewritten to make it clear. "Here the shape of graupel was modeled as a spheroid, where the axis ratio for dry graupel was set to a constant value of 0.8, and the axis ratio for melting graupel was modeled according to Ryzhkov et al. (2011)."

**L 200-203: Are you also ignoring aggregation and collision-coalescence?**

Response: Yes, the aggregation and collision-coalescence were ignored as well. "Neglecting aggregation, collision-coalescence, evaporation, and the small amount of water that may collect on the particle owing to vapor diffusion, we assumed that the mass of snow was conserved during the evolution process from dry snow, to wet snow to liquid water."

**L 209-211: But the exponent "b" changes as the particle melts and the shape of the particle melts, does it not? In the end, when the particle is fully melted, and nearly spherical, the value of "b" should be near 3. Can you justify using b=2.1 over the full range of particle melting?**

Response: Yes, the value of "b" varies with the melting degree. Sorry for the unclear description. With an assumed $f_w$ value, the density of wet snow can be calculated from Eq. (16). Then, the density parameter (prefactor "a" or exponent "b") can be obtained according to the density-diameter relationship. The sentence has been rewritten. "The density parameter in Eq. (11) can be obtained according to the density-diameter relationship, where the density is calculated from Eq. (16) with an assumed $f_w$ value."

**L221:  Should the left hand side of equation 19 be "N_w(D_w)"?**

Response: Yes. Thanks for pointing out. The equation 19 has been rewritten.

$$N_w\left(D_w\right)=N_{ms}\left(D_{ms}\right)\frac{3}{b}\left(\frac{6}{\pi}a\right)^{-\frac{1}{3}}D_{ms}^{\frac{3-b}{3}},\tag{19}$$

**L 235:  Usually the term "extinction cross-section" is used.**

Response: Thanks for your suggestion. The term "extinction section" has been revised to "extinction cross-section".

**L 239-241:  And how were the attenuation and the two-way path integrated attenuation addressed when combining different types of hydrometeors?**

Response: The relevant information has been added now in the revised manuscript. "If there are many types of hydrometeors at the same height, the equivalent unattenuated radar reflectivity and attenuation coefficient of each hydrometeor is calculated based on the look-up table. Then, the total unattenuated radar reflectivity at this height is obtained by adding all types of hydrometeors, and the two-way attenuation is obtained by integrating the total attenuation coefficient with path. The attenuated radar reflectivity is obtained by subtracting the attenuation from the unattenuated radar reflectivity."

**L 247-333:  This is a general comment for the sensitivity section.  Wood and L'Ecuyer (2021, AMT) looked at W-band retrieval uncertainty sources.  How do your results compare with theirs?**

Response: Thanks very much for your valuable suggestion. Wood and L'Ecuyer (2021) showed that the contributions to uncertainties in W-band radar reflectivity from the particle model parameters (e.g., the coefficients and exponents of the mass relationships) was most substantial, which may cause 5 to 15 dB reflectivity uncertainty. Our study shows that he reflectivity change caused by variation in prefactor $a$ from 0.005 to 0.013 g/cm$^b$ for snow and 0.02 to 0.06 g/cm$^b$ for graupel ($W$ remains constant) can reach 7–10 dB. The reflectivity change caused by variation in $b$ was approximately 0.5-2 dB. The mass relationships cause substantial uncertainties in W-band radar reflectivity. Our results are consistent with those in Wood and L'Ecuyer (2021). The comparison with Wood and L'Ecuyer (2021) has been added in the revised manuscript.

"In all, the mass relationships that depend on particle habits and formation mechanisms, cause substantial uncertainties in W-band radar reflectivity. Our results are consistent with the sensitivity analysis by Wood and L'Ecuyer (2021) who pointed out that the W-band radar reflectivity uncertainty for snowfall was dominated by the particle model parameter (e.g., the prefactors and exponents of the mass relationships). This relationship can cause the reflectivity uncertainty of several to more than 10 dB. The results indicate that improved constraints on assumed particle mass models would improve forward-modeled radar reflectivity and physical parameter retrieval."

**L 258:  It's probably more correct to say that the particles are small "compared to the radar**

**wavelength".**

Response: Thanks for your suggestion. The sentence has been rewritten. "Cloud water particles are small compared to the radar wavelength, which is in the linear growth stage in the Mie scattering region."

**L 259-261:  Check grammar/sentence structure.**

Response: Sorry for poor writing. The sentence has been rewritten. "With a five-fold increase in $D_0$ ($W$ remains constant), e.g., increasing from 10 to 50 $\mu m$, the reflectivity increases by approximately 20 dB."

**L 259-264:  Were these sensitivities calculated by perturbing D_0 while simultaneously keeping W constant?  Or did W increase as D_0 was increased?**

Response: These sensitivities were calculated by perturbing $D_0$ while simultaneously keeping W constant.

**L 265:  Please check this equation reference.  I think it is not correct.**

Response: This equation does not cite references. The references are for the value of $\mu$.

$$N_t = \int_0^{D_{max}} N(D)dD \tag{9}$$

Owing to the monotonicity of the functions, $D_0$ can be solved numerically. For cloud ice, $\mu$ usually ranges from 0 to 2 (Tinel et al., 2005; Yin et al., 2011).

**L 278:  This isn't the correct equation to convert N_0 to dB(N_0).  "dB" indicates "decibel" (i.e., "deci" "Bel", or one-tenth of a Bel).  dB(N_0) should be 10*log10(N_0).**

Response: Sorry for the unclear exhibition. Here, we use $dBN_0 = \log_{10}(N_0)$ to convert $N_0$ of $10^3$ to 3 just for the convenience of writing and image display. We don't convert $N_0$ to decibel.

**L 280-282:  This states "may result in an uncertainty of approximately 45% and 30% for snow and graupel", but it doesn't say what property of the snow and graupel this uncertainty applies to.  Please clarify.**

Response: Thanks very much for your valuable comment. We have added the description of the cause of reflectivity uncertainty contributed by "a" variation. "The mass power-law parameters vary with snow/graupel type, shape, and porosity. An increase in $a$ lead to an increase in the corresponding particle scattering properties. The intercept parameter $N_0$ will slightly decrease with the increase in $a$ implicitly representing the effects of aggregation at warmer temperatures (Woods et al., 2008). Among them, the change to particle scattering properties caused by the perturbation of $a$ play a dominant role in the reflectivity change. Using an average mass-power relation assumption, the variation in $a$ as a result of the degree of aggregation and riming, and particle shapes may result in the reflectivity uncertainty of approximately 45 % and 30 % for snow and graupel, respectively."

**L 296-299:  Please recheck your values for D_0.  20 mm and 30 mm seems extremely large for liquid cloud droplets.  Either there is a typographic error here, or an error in the**

**calculation of D_0, I think.**

Response: Sorry for the careless. This is a typographic error. Thanks very much for pointing out. 20 mm and 30 mm has been revised to 20 μm and 30 μm.

**L 297-299:   I don't think there is much gained by including the results from the Gamma(D_0=30) case.   Clearly, if two PSDs for liquid water droplets are nearly the same, the simulated reflectivities will be nearly the same.   The significant point here is that, given the same water content, different assumptions about the shape of the PSD can have a strong effect on the simulated reflectivity.**

Response: Thanks very much for your valuable suggestion. This gamma case ($D_0$=30 μm) really doesn't make much sense. The case of gamma $D_0$=30 μm has been removed.

[Figure]

Figure 5: Impact of PSD models on radar reflectivity for cloud water. (a) Black solid line is for the gamma distribution: $W = 1$ ($g/m^3$), $D_0 = 20 \ \mu m$, and $\mu = 2$. Red-dotted line is for the log-normal distribution: $W = 1 \ g/m^3$, $D_m = 20 \ \mu m$, and $\sigma = 0.35$. (b) Variation in the radar reflectivity with $W$ and the PSD models, where the PSD models are from (a).

**L 300:   It's probably more correct to say the "reflectivity change" was 4.5dB.**

Response: Thanks very much for your suggestion. The sentence has been rewritten. "The reflectivity change caused by the different PSD models was approximately 4.5 dB."

**L 314-334:   I think it's questionable whether these different shapes of soft (mixtures of ice and air) particle shapes give a good representation of the sensitivity of reflectivity to particle shape and orientation.   Methods such as the discrete dipole approximation are accepted as giving much more realistic values for backscattering by ice and snow particles.   I think it would be appropriate to look at other sources of DDA backscattering properties (e.g. the Liu database mentioned earlier, or OpenSSP) to see if your results are consistent with DDA results.**

Response: Thanks very much for your valuable suggestion. We have recalculated the scattering properties for cloud ice and dry snow using T matrix and DDA algorithm, respectively. Figure 6 has been updated. Detailed information can be seen in the response to the "overall comments: particle shape and orientation".

[Figure]

Figure 6: Backscattering cross-section and corresponding radar reflectivity under different shapes for cloud ice, dry snow, and rain. (a) Comparison of the backscattering cross-sections of ice crystals as spheres, spheroids, or cylinders, where $\delta$ is the SD of the canting angle. (b) Radar reflectivity comparison for particles in (a), where the PSD was assumed as a Gamma distribution constrained by Eqs. (7)–(9), with $\mu=1$ and T = –60º C. (c) Comparison of the backscattering cross-sections for dry snow with spheres and spheroids. (d) Radar reflectivity comparison for particles in (c), where the PSD was assumed as an exponential distribution with $N_0 = 3 \times 10^3 \, m^{-3} \, mm^{-1}$. (e) Comparison of backscattering cross-sections for raindrops with spheres and spheroids. (f) Radar reflectivity comparison for particles in (e), where the PSD was assumed as a Gamma distribution with $D_0 = 1.25$ mm and $\mu=3$.

The results in Matrosov (2007) and Wood et al (2015) showed that the reflectivity difference for dry snow between the sphere and spheroid assumption can reach approximately 2 dB. The magnitude of backscattering cross-section and reflectivity difference between spherical and non-spherical in Fig. 6 are basically consistent with those in Matrosov (2007) and Wood et al (2015). The slight difference is mainly due to the different setting of the SD of the canting angle.

**L 345-346: What was the vertical grid spacing? Was the spacing uniform or stretched (with layers getting generally thicker with height)? What data were used for initial and boundary conditions? What time-stepping was used? What microphysical parameterizations were used?**

Response: Thanks for the comment. The detailed description of model setup has been added in the Appendix A.

"The simulation for this stratiform case was conducted with four nested grids (d01, d02, d03, and d04), and the inner domain was centered at 41.08°N, 117.61°E. The horizontal grid spacings are 27km, 9km, 3km, and 1km and the corresponding grids are 120×120, 180×180, 300×300, and 300×300. The vertical resolution increases with height from approximately 50 m near the surface to 600 m near 50 hPa. Time steps of 180s and 6.67s were used for d01 domain and d04 domain, respectively. The 6-hourly NCEP FNL operational global analysis data on 1° × 1° grids were used to provide the initial and boundary conditions. In term of physical scheme, the model adopted CAM 5.1 5-class scheme, Grell-Freitas cumulus parameterization scheme, RRTM long and short-wave radiation scheme, YSU boundary layer scheme, Monin-Obukhov surface layer scheme and thermal diffusion land surface scheme. The cumulus parameterization scheme was used for d01 and d02 domains only. "

**L 347: It's probably more correct to say "interior domains" rather than "internal layers".**

Response: Thanks very much for your suggestion. The term of "internal layers" has been revised to "interior domains".

**L 359-360: I'm not sure it's accurate to say that WRF "accurately simulated the cloud system" based only on comparisons of cloud fraction and cloud top temperature.**

Response: Besides cloud fraction and cloud top temperature, we have added comparisons of cloud water path (CWP) in Appendix A in the revised manuscript.

Besides the cloud fraction and cloud top temperature shown in Fig. 8, the cloud water path (CWP) from MODIS was used for model result verification as well. Figure A1a is the cloud water path calculated from vertical integration of WRF output cloud water over d01 domain at 03:30 UTC, 25 September, and Fig. A1b is the cloud water path from MODIS Level 2 product at 03:35 UTC, 25 September 2012. The scanning width of MODIS is 2330 km, and the horizontal resolution for the product of CWP is 1 km. The CWP distribution of model result has similar pattern as MODIS observation and the value of CWP are close, but the peaks of the two are slightly offset. Due to the measurement techniques and model limitations, the model simulations may be biased from the observations. However, the distribution, structure, and value of CWP from model and MODIS observation generally agree well.

[Figure]

Figure A1: Comparison of CWP between the WRF model result and MODIS data for the stratiform case. (a) CWP from the WRF model at 03:30 UTC, (b) CWP from MODIS observation at 03:35 UTC, 25 September 2012.

**L 362-363:   I'm not sure that choosing to use the WRF results along the CloudSat track is an effective way to do comparisons between models and satellite observations.   One of the frequent errors in models is features like clouds and precipitation may not be located precisely in the location of interest at a particular time.   As an example, modeled fronts and their associated precipitation may propagate more slowly or more rapidly than the observed preciptation.   Perhaps a better approach would be to statistically compare the properties of the modeled versus the observed clouds and precipitation, using model results from the area under *and near* the CloudSat ground track.**

Response: Thanks very much for your valuable comment. Yes, the model result may not be located precisely in the location of interest at a particular time. The area used for validation was much larger than that of CloudSat trajectory, which included the area under and near the CloudSat ground track. Besides, we have added the statistically comparison for the model validation of the convective case.

In Fig.8, the CloudSat track was along the black line in the d04 domain, which was just in the innermost domain. The model validation is for d02 or d01 domain. For the convective case, CloudSat observed this convective process at 04:30 AM on June 23, 2016. In the model validation of convective case, we compared the reflectivity from the WRF simulation over the d03 domain at 04:00 UTC on June 23 and ground radar at Lianyungang city at 04:02 UTC on June 23, 2016. The 6-hour accumulated rainfall data were also compared with those from rain gauge.

"For validating the model result, the ERA5 data, ground radar reflectivity and rain gauge data were used. Figure A2a-f compares the fraction of cloud cover, reflectivity, and rainfall from the WRF model with the observation data. Figure A2a shows the fraction of cloud cover from the WRF model at d02 domain. The cloud area and coverage are consistent with the ERA5 data shown at Fig. A2b. Figure A2c and d compare the reflectivity from the WRF simulation over the d03 domain at 04:00 UTC on June 23 and ground radar at Lianyungang city at 04:02 UTC on June 23, 2016. From radar observation, we can see that the strong echo area is relatively scattered, generally trending from northwest to southeast, and the maximum reflectivity is about 55 dBZ. In the simulation, the strong radar echo is mainly distributed along the northwest-southeast; the radar echo structure and echo intensity are close to the radar observation. Figure A2e and f show the 6-hour accumulated

rainfall from 00:00 to 06:00 on June 23 from the WRF model and rain gauge data, respectively. The rainfall covers most areas in the north of Jiangsu Province, and there are two heavy rainfall centers of more than 100mm. The rainfall area in the simulation is similar to that from rain gauge data, and three heavy rainfall centers can be seen in the model result. The maximum rainfall from rain gauge data is approximately 120 mm and maximum from WRF is approximately 126 mm. The amount, scope, and distribution of rainfall from WRF simulation are generally consistent with the rain gauge data. The main difference is in the strong rainfall location and extreme value. Considering the model limitations, the comparison results show that the model captured the convective precipitation process.

[Figure]

Figure A2: Comparison between the WRF model result and observation data for the convective case. (a) Fraction of cloud cover from the WRF model, (b) fraction of cloud cover from the ERA5 data, (c) radar reflectivity from the WRF model at 04:00 UTC, (d) radar reflectivity observed by the Lianyungang radar at 04:02 UTC, 23 June, (e) WRF model-simulated 6h accumulated rainfall from 0:00 to 06:00 UTC, 23 June, (f) 6h accumulated rainfall from rain gauge data from 0:00 to 06:00 UTC, 23 June 2016."

**L 367-369: Regarding "Snow is widely distributed...", can you provide more details? Are you talking about the horizontal extent, the vertical extent, or something else. Maybe**

something like **"The vertical extent of snow is widely distributed...". Same comment with respect to the next statement, which is about rain. "Rich" is not a clear description. Do you just mean to say the total water contents for cloud water, snow and rain were large?**

Response: Sorry for the unclear statement. Thanks very much for your suggestion. The relevant sentences have been rewritten. "The vertical extent of snow is widely distributed, ranging from 3 to 10 km. Rain is mainly below 3 km, with water contents between 0.1 and 0.2 g/m$^3$. At approximately 0 ºC, the water content for cloud water, snow, and rain were large, which led to a high total water content, with a maximum of 0.57 g/m$^3$."

**L 372: The Yin et al. (2017) work cited here does not appear in the bibliography.**

Response: I am very sorry for the careless. Thanks very much for pointing out. The Yin et al. (2017) has been added in the bibliography.

Yin, M.T., Liu, G.S., Honeyager, R., and Turk, F.J.: Observed differences of triple-frequency radar signatures between snowflakes in stratiform and convective clouds, J. Quant. Spectrosc. Radiat. Transfer, 193, 13-20, doi: 10.1016/j.jqsrt.2017.02.017, 2017.

**L 375-376: See my comment regarding the Figure 9 caption (L 724). It's probably more clear to refer to these as "unattenuated reflectivities", "attenuation" (is this one-way or two-way?), and "attenuated reflectivities".**

Response: Thanks very much for the suggestion. The terms of "reflectivity before attenuation", "attenuation", "reflectivity after attenuation" have been revised to "unattenuated reflectivities", "two-way attenuation", "attenuated reflectivities".

**L 379-381: Suggest using "unattenuated reflectivity" and "attenuated reflectivity". Also rather than the "end of the melting region", use "below the melting layer" or "below the melting level." Also suggest using "with attenuation" and "without attenuation" rather than "after attenuation" and "before attenuation". "Before" and "after" can have misleading implications when talking about a radar beam propagating downward through the atmosphere. Finally, there is a well-known reference to this behavior in W-band radar observations from space. See Sassen et al. (2007, Geophysical Research Letters).**

Response: Sorry for the unclear expression. Thanks very much for your valuable suggestion. The relevant terms and sentences have been rewritten. "The unattenuated radar reflectivity in the melting layer was equivalent to the reflectivity in the rain region. With attenuation, the radar reflectivity showed a rapid signal decline below the melting layer, and the bright band became evident (Sassen et al., 2007)."

**L 382-383: Did you demonstrate this, or did you mean to cite existing work? How large does this diameter need to be, and how is this relevant to the bright band discussion? If I look at figures 6 and 7, the backscatter cross-sections for the larger particles do not appear to be stable.**

Response: Sorry for the unclear description. The sentence has been rewritten. "For the 94 GHz radar,

the Mie scattering effect was dominant. The raindrops with a diameter less than 1 mm are the dominant contributor to the radar reflectivity profile (Kollias and Albrecht, 2005). Although larger snowflakes melt and produce larger raindrops at depth in the melting layer, their contribution to the reflectivity was not significant, owing to a decrease in their number concentration."

**L 396:   Usually just "bright band".**

Response: Thanks for your suggestion. The term of "brightness band" has been revised to "bright band".

**L 399:    Rather than using different names for this feature ("brightness band", "bright band", "strong echo band"), please choose one name and use it consistently.    Also, when you say the reflectivity was stronger, what are you comparing to?**

Response: Thanks very much for your valuable suggestion. The terms of "brightness band", "strong echo band" all have been revised to "bright band". Besides, the sentence about stronger reflectivity has been rewritten. "In Fig.10b, the radar reflectivity below 0 ℃ was evidently stronger than the echo above 0 ℃; the width and location of the bright band were considerably different from the bright band in the simulation with the improved setting and CloudSat observation."

**L 400-401:    How did you calculate this relative error?    Relative errors shouldn't be calculated using "dB" values (i.e. (dB_test - dB_true)/dB_true. The values should be converted back to linear units (e.g., mm^6 m^-3), then the relative error calculated.**

Response: Thanks very much for your valuable suggestion. The description of the calculation of relative error has been added. "The trends in the two profiles were basically identical; the relative error ($|Z_{sim} - Z_{obs}|/Z_{obs}$, where $Z_{sim}$ represents the simulated reflectivity and $Z_{obs}$ represents the observations, the units of $Z_{sim}$ and $Z_{obs}$ are converted to mm$^6$/m$^3$) at each height was mostly within 15 %."

**L 407-408:    See my earlier comment concerning the modeling of the stratiform case. Additional details about the model configuration would be interesting to see.    What was used for convective parameterization?**

Response: Thanks for your suggestion. Detailed description of model setup has been added in the Appendix A.

   "For the convective case, three nested grids (d01, d02 and d03) with horizontal grid spacings of 22.5km, 7.5km, and 1.5km and corresponding grid points of 70×70, 126×126, and 280×280 were used for the convective case simulation. The inner domain d03 is centered at 34.02°N, 118.20°E. A total of 39 vertical layers with stretch spacing from the surface to 50 hPa were used, with time steps of 90, 30, and 6 s for d01, d02 and d03, respectively. The initial and boundary conditions used the NCEP FNL analysis data as well. The model adopted NSSL 2-moment 4-ice scheme for microphysical process, Kain-Fritsch cumulus parameterization scheme, RRTMG long and short-wave radiation scheme, YSU boundary layer scheme and five-layer thermal diffusion land surface

scheme. The Kain-Fritsch cumulus scheme was not used for d03 domain. The simulation starts at 12:00 UTC on 22 June and ends at 12:00 UTC on 23 June 2016."

**L 416: See earlier comment concerning "rich" and "widely distributed". Also, I would suggest that when discussing results that involve vertical profiles of data, don't use the terms "high" and "low" to describe data values. Instead, use "large" and "small".**

Response: We are sorry for the poor writing. Thanks so much for helping us with the English. These terms and sentences have been corrected.

**L 421: See my earlier comment about the missing Yin et al. (2017) reference.**

Response: Thanks for pointing out. The reference of Yin et al. (2017) has been added, as can be seen in "response to comment L372".

**L 425-428: Does this mean that the assumed rime mass fraction, and therefore the adjustment factor "f" was uniform with height for each simulated profile (since liquid water path is a column variable)? Also, are you saying that you treated the rime mass fraction as liquid water for the purpose of refractive index calculations?**

Response: The adjustment factor "f" increased as the height decreased. Sorry for the inaccurate statement. The sentence has been rewritten. "According to Mason et al (2018) and Moisseev et al (2017), we assumed that the rime mass fraction increased linearly with liquid water path. The ELWP was defined as the vertical integration of liquid water content (LWC), i.e., $ELWP = \int_{h}^{\infty} LWCdz$, where $h$ is the height."

For the refractive index, it was calculated according to the volume fraction of water ($f_w$). With the rime mass fraction, the density-diameter relation can be obtained, and then the $f_w$ can be obtained according to the relationship between density and $f_w$ in Eq. (16).

**L 429: But you adjusted the PSD so that the water contents in the simulated profile matched the water contents output by the model, yes?**

Response: Yes, we adjusted the PSD so that the water contents in the simulated profile matched the water contents output by the model. The sentence about the number concentration of convective cloud precipitation has been removed.

**L 451-452: See my earlier comment about computing relative error with reflectivities.**

Response: Thanks very much for the comment. The calculation of relative error has been checked.

**L 463: You mean the sensitivity of reflectivity to D_0? When describing sensitivities, try to express them as "the sensitivity of x to changes in y" so that the meaning is clear.**

Response: Thanks very much for your valuable suggestion. Yes, we mean the sensitivity of reflectivity to $D_0$. The sentence has been rewritten. "The sensitivity of radar reflectivity to changes in $D_0$ in the Gamma distribution was approximately 5–10-fold greater than that of $\mu$; the variation in $\mu$ can cause reflectivity changes of less than 10 %."

**L 463-467:  You mean the sensitivity of reflectivity to D_0?  When describing sensitivities, try to express them as "the sensitivity of x to changes in Y" so that the meaning is clear.  Yes, by imposing the empirical constraints on the PSD, the PSD itself has few degrees of freedom compared to a PSD with independent variations in parameters, so the PSD is less variable. Is this an unexpected result?  Finally see my earlier comment about computing relative changes in reflectivity - be sure these percentages are calculated correctly.**

Response: Thanks very much for your suggestion. Yes, by imposing the empirical constraints on the PSD, the PSD itself has few degrees of freedom compared to a PSD with independent variations in parameters, so the PSD is less variable. This is not an unexpected result. This gives the quantitative value of the reduction of reflectivity uncertainty through sensitivity analysis and demonstrates the importance of the constraint on PSD modeling.

Thanks for reminding. The calculation of relative changes in reflectivity has been checked.

**L 467-468:  How does the particle density affect the PSD?  In general the particle density (as defined by the coefficient "a" and exponent "b" of the mass power law) are considered independent of the PSD paramemters.**

Response: Sorry for the unclear description. The sentence has been rewritten. "The mass-diameter relationships for snow and graupel differ substantially for different particle habit types, which not only affects the particle scattering properties, but also affects the shape of PSD. Using the exponential PSD with a power-law mass spectrum for snow and graupel, we found that the effects of prefactor $a$ on radar reflectivity were significantly. Variation in $a$ mainly via changes in the particle scattering properties may result in reflectivity uncertainty of approximately 45 % for snow and 30 % for graupel."

**L 469-472:  But is this sensitivity due to the increase in "a" changing the scattering properties of particles, or is it because the increase in "a" increases the water content for the population of the particles?  These two effects need to be separated, otherwise the influence of the change in "a" is overestimated.  Also, see my earlier comment about computing fractional sensitivities for reflectivity.**

Response: The water content ($W$) remains constant when analyzing the sensitivity of reflectivity to particle density parameters (prefactor "a" and exponent "b"). "Variation in $a$ mainly via changes in the particle scattering properties may result in reflectivity uncertainty of approximately 45 % for snow and 30 % for graupel."

The calculation of fractional sensitivities for reflectivity has been checked.

**L 480-481:  Relative errors in what?  Also, see my earlier comment about computing fractional errors in reflectivity.**

Response: These are relative errors in radar reflectivity profile between the simulation and CloudSat data ($|Z_s - Z_c| / Z_c$, where $Z_s$ is simulated reflectivity and $Z_c$ is CloudSat observed reflectivity in mm⁶/m³). "The average relative errors in radar reflectivity profile between the simulation and

CloudSat data were within 20 %, which improved by 20–80 % compared with the conventional setting." The calculation of fractional errors in reflectivity has been checked.

**L 724: It's not clear what is meant by "before", "during", and "after" attenuation. I'm guessing that panel (f) shows the unattenuated reflectivities, panel (g) shows the attenuation (is this one-way or two-way?), and panel (h) shows attenuated reflectivities. Is that correct?**
Response: Yes, you are right. Thanks for your comment. The terms have been corrected. "(f) Simulated unattenuated radar reflectivity with the total hydrometeors, (g) two-way attenuation, and (h) attenuated radar reflectivity."

For further improving the fluency of reading the manuscript, we have paid another commercial editing service to polish our manuscript for the language. We would like to thank the reviewer for his/her significant effort to suggest changes for our manuscript.
* * *
Special thanks to the reviewer for the valuable comment and his/her patience.

---

## Referee Report (RR1)

**2nd round of reviews of "Simulation and sensitivity analysis for cloud and precipitation measurements via spaceborne millimeter wave radar" by Kou et al.**

**General comments**

I want to thank the authors for their hard work in revising and improving the manuscript. In general, the revised manuscript shows vast improvements, especially in terms of readability, motivation and in clearing up previously ambiguous passages. I stand by considering the study as suitable for AMT. However there are still a few points that should be changed before publishing. Also, one explanation regarding the calculation of mass size parameters depending on rime fraction is puzzling to me. I therefore recommend another round of major revisions.

**Specific comments**

I still find the difference between "conventional" and "optimized" settings to be unclear in the text. Writing "the main difference…" gives the impression that there are other differences that are not mentioned / explained. If that is the case, please write down the other differences. If not, then omit the "main".

It is still not clear to me what mass-size parameter literature values are averaged over. I still recommend including a sentence like, "For a and b we took literature values from -list of studies- and calculated the mean" and please go into detail, if you have left out any values that are listed in the literature (for example values for hail).

I don't understand the description of the variable density (variable mass size prefactor a) depending on rime mass fraction. As far as I'm aware the ELWP is the liquid water path along the trajectory of the rimed particle assuming a riming efficiency of 1. It is therefore not equal to the LWP and the equation in L510 (which is the definition of LWP not ELWP as far as I know) is incorrect. In Moissev et al. 2017, ELWP is about two times lower than LWP. If LWP=ELWP is assumed in the study, this must be discussed in more detail. I also don't understand how FR was calculated from ELWP, the formula seems to be missing. It is written that a linear increase of FR with ELWP is assumed and Moissev et al. 2017 is referenced. However, in Moissev Eq. 8 ELWP is proportional to FR/(1-FR) which is not linear. Also the scatter plot in Fig 9 of Moissev et al. 2017 does not show a clear linear behavior between FR and LWP. I would therefore like to ask the authors to clear up these issues and discuss the calculation of the adjustment factor f in more detail. It is fine in my opinion to assume a linear relation between f and LWP, if this is what was done (I am not sure). But this decision must be discussed and possible errors resulting from that approach must be mentioned. In addition, the formula that was used to calculate f should be included (including numerical parameters).

**Technical corrections (& minor comments)**
**L18** "optimal physical modeling" is misleading
**L34** Not the best sentence, maybe: "Cloud radars are mainly operated spaceborne, airborne or ground-based."
**L39** "with a minimum detectable signal of about -30 dBZ"
**L42** environment → environmental (?)
**L50** highly important → important
**L60**
**L128** was → were (or are)
**L140** include references, where did you get the knowledge that 10 microns is typical?
**L141** include Mie and Rayleigh citations

**Eq 8** form looks a bit weird, maybe inlcude spacing or dot between number and exp?

**L200** I don't think mature is the right term. Also what do you mean by that?

**L215** & **L219** it's misleading that "prefactor a can vary considerably" and "relations vary slightly" are written

**L229** why is 0.4 the typical value? Explanation on how that was derived is missing

**L231** graupels → graupel particles

**L240** unclear sentence

**L258** uniform bin size set → I still think that the bin size should be mentioned somewhere earlier

**L282** by  T-matrix

**L284** I don't understand. Is the look-up table for backscattering cross-sections of individual particles?

**L301** change with dominating microphysical processes (?)

**L311** do you mean Rayleigh regime?

**L335** leads to

**L342** "most of the mass relations have the mean value of b close to…" → misleading / unclear; maybe better write: When averaging literature values of the exponent b from – list of studies – we derive b = 2.1 for particles classified as snow.

**L346** see **L342** but for graupel

**L358** represented as different models → represented by different distributions

**L369** represents  the

**L395** became apparent (?)

**L413** Maybe better "We then selected two typical weather cases: …." And maybe include a bit more explanation on why these were chose as reviewer#2 has suggested.

**L420** reference to appendix?

**L425** the maximum total water content was at approximately 3 km with ~0.9 g/m³

**L429** please include the resolution of ERA5 and MODIS (maybe in brackets?) here

**L447**  empirical

**L449** maybe include geographic region. In high latitudes (arctic) riming is observed also in stratiform clouds

**L467** with attenuation → in the attenuated signal ?

**L478** what does mostly mean? Either write xx% was within yy% or omit mostly and write the (higher) uncertainty that holds for all data

**L503** include "typically" or "commonly" otherwise statement is incorrect

**L546** nine sub modules → eight sub modules?

**L557** significantly greater than … what?

**L570** is it % or rather percent points?

---

## Referee Report (RR2)

```
===========
1.  Summary
===========
```

This is a review of the revised version of Kou et al., "Simulation and
sensitivity analysis for cloud and precipitation measurements via spaceborne
millimeter wave radar".  The authors evaluate the sensitivity of a forward
model for radar reflectivity to its microphysical input variables.  The
forward model includes cloud ice and water, melting mixed-phase precipitation,
snow, graupel and rain.  They then perform comparisons of reflectivities that
are forward-modeled for two WRF simulations (one stratiform and one convective
event) against CloudSat observations of the same events.

Although the authors have provided sufficient responses to most of my original
concerns,  there are still two substantial issues that have not been addressed
adequately.

Issue 1 could be addressed by deferring the particle shape and orientation
part of this study to a future, more complete study.  Issue 2 could be
addressed by following the revisions I've suggested below.

I believe that addressing these issues and that by addressing the remaining
comments on this revision of the paper, the paper will be suitable for
publication.

```
===============
2.  Main issues
===============
```

```
Issue 1:  Particle shape sensitivities
++++++++++++++++++++++++++++++++++++++
```

The authors responded to my original comment by performing DDA simulations of
the scattering properties of their chosen particle shapes (sphere, spheroid,
cylinder).  The point of my comment wasn't that DDA needed to be applied to
these shapes.  Instead the point was that more realistic shape variations are
needed and that DDA is the method usually used to calculate scattering
properties for more realistic shapes.  The use of realistic shapes and DDA
(or perhaps Raleigh-Gans) to calculate scattering properties is the current
standard for evaluating the shape sensitivity for millimeter-wavelength radar
reflectivity in snow.  I think that the authors cannot claim to be assessing
shape sensitivity accurately when using only spheres, spheroids and cylinders.

```
Issue 2:  "Conventional" versus "improved"
++++++++++++++++++++++++++++++++++++++++++
```

After going through this version of the paper thoroughly, I still find it
difficult to discern what are the conventional and improved assumptions for
the two test cases.  This needs to be stated more clearly.  Part of the
problem is that there is no clear layout of the experimental design (this
would usually be included in a methods or objectives section, but section 2.1
is the closest we have to this).

I would suggest:

a) Add a paragraph just after the first paragraph in section 4.  The new
paragraph should describe the authors' intentions to test the forward model
simulations using both conventional and improved parameter settings and
briefly describe in general terms the objective of using the conventional and

improved settings.

b) Add a section just before section 4.1.2 that contains an outline of what is
being tested for the stratiform case.  Describe what parameters are changed
between the conventional and improved radar simulations for this case and the
scientific justification for those parameter changes.  Then proceed to
describe the radar reflectivity simulation results.

c) Do the same thing in section 4.2 for the convective case.  Also, structure
section 4.2 similar to the way section 4.1 is structured:  A subsection for the
WRF simulation description, a subsection for the experiment design (describing
the conventional and improved parameter settings), and a subsection for the
results.

```
==============================
3.  Responses to prior comments
==============================
```

These notes provide my assessment of the authors' responses to my original
comments (egusphere-2022-886-author_response-version1.pdf)

Prior comments, overall
+++++++++++++++++++++++

1.  Thanks for providing these additional details.  They are sufficient for
explaining the perturbations in b.

2.  This revision addresses my original comment, thanks.  There are some
additional comments that apply to these revisions, please see the specific
comments section that follows.

3.  Thanks, this additional text resolves my comment.

Prior comments, WRF model simulations
+++++++++++++++++++++++++++++++++++++

1.  Thanks for this response and the details provided in the new Appendix A.
This addresses my concern, but please also see the specific comments section
that follows.

Prior comments, particle shape and orientation
++++++++++++++++++++++++++++++++++++++++++++++

1.  This doesn't really address the point of my original comment.  The meaning
of the original comment is that using soft spheres, spheroids and cylinders
doesn't give a realistic representation of how scattering properties for snow
particles vary with shape at 94 GHz.  This is true regardless of whether the
spherical/spheroidal particles' scattering properties are calculated using DDA
or T-matrix theory.

See for example, Figure 12 and the related discussion in Wood et al. (2015).

In order to accurately assess sensitivities of radar reflectivity to particle
shape variations, more realistic partical shape variations must be used.  And
in order to evaluate the scattering properties of more realistic particle
shapes, a technique such as DDA must be used.

The authors comment:

"We mainly considered the difference between sphere and spheroid with
with different orientations in this study.  In future research, we will
consider the influence of more particle shapes on radar reflectivity."

I think this is not sufficient to support the authors claims of evaluating
particle shape and orientation effects in this study.

===============================================
4.  Specific comments from review of version 2
===============================================

Note that the ATC document and the version2 paper are not consistent in their
revisions.  For example, L21 of the ATC uses the phrase "brightness band"
while the corresponding line in the version 2 paper (also L21) uses the term
"bright band".  L39 in the ATC gives CloudSat minimum detectable signal of -30
dBZ, while L38 of the version2 paper gives -29 dBZ.

Comments and the line numbers used here refer to the revised version2 paper.

L11:  Should be "improve" rather than "improving".

L22-23:  Relative error in the vertical profile of what variable?

L51-53:  To be correct, QuickBeam doesn't compare modeled clouds to
observations, it is a radar simulator package.  It is up to the users to make
the comparisons.  Also, QuickBeam is capable of simulating radar
reflectivities for radars other than CloudSat.  Finally, to say that QuickBeam
does not simulate mixed-phase melting particles is entirely incorrect.  See
section 4 of the Haynes et al. paper you have referenced.

L57:  No citation for WRF-SBM.

L60-61:  I am not sure what a "cloud data simulator" is, please clarify.
If this is referring to cloud radar simulators, the statement is not correct.
QuickBeam, as an example, uses scattering properties obtained from discrete
dipole simulations of realistic snow particle shapes from the Liu (2004).  It
does not use an "equivalent spherical shape" for snow particles.

L82-85:  Technically, all of these steps are not part of the "forward
modeling".  The "forward model" consists only of the component that takes in
the simulated cloud and precipitation fields from WRF and outputs the
simulated reflectivity profiles.  The activities listed here actually compose the
entire research method.

L83:  Should be "Weather Research and Forecasting (WRF) model".  Also, no citation
is provided for the model.

L92:  Be a bit more specific here.  Which "real observation data"?

L96:  Should be "refractive index", not "reflective index".

L99:  Need citation for Liebe model.

L143:  I'm not sure what is meant by "direction of raindrop particles".
Please clarify.

L162:  Can you provide a citation that supports this statement?  I don't
recall ever seeing an exponential distribution used for cloud ice.

L177–178:  This statement explicitly contradicts the actual findings in Nowell
et al, 2013.  Nowell et al. find that "the backscatter cross section is not
well duplicated by the soft or solid spherical/spheroid approximations" in
comparison to DDA results for realistic particles.  This quote from Nowell et
al. applies to particles with size parameters larger than "x ˜ 0.75", which is
true for most snowflakes at 94 GHz.

This is the root of my concern raised in my original comments about the need
for using more realistic shape and scattering models for snow particles.

L204–205:  I'm not sure how the comment on graupel altitudes is relevant to
this work.

L289:  Is this equation reference correct?  None of these variables appear in
equation 9.

L294:  Same comment as above for L289.

L300–325:  This is a long paragraph and covers several different topics.
Perhaps split it into two or three shorter ones.

L302:  I still object to this use of 'dB'.  Using the units 'dB' for this
quantity is equivalent to using the units 'mm' for a variable that is
measured in meters.  It is misleading, confusing, and shouldn't be done in a
professional publication.

L306–308:  It is not clear how this statement about changes in N0 through
natural aggregation processes is relevant to the sensitivity study.

L308:  What is "among them" referring to?  This isn't clear.

L357–358:  I don't think a comparison of reflectivities calculated using
sphere and spheroid shapes will adequately evaluate the sensitivitiy of radar
reflectivity to snow particle shape.  The actual uncertainty at 94 GHz is much larger
than 1.6 dB.  See for example, Wood et al. (2015) for an evaluation of
different aggregate shape assumptions.

L376:  Do you mean "mixing ratio"?

L390–392:  Citations needed for ERA5 and MODIS products.

L399–400:  I don't think it is possible to unequivocally state that the cloud
scenario simulation results are valid based solely on evaluations of cloud
fraction and cloud top temperature.

L414:  Are these mass-power parameters the "improved microphysical parameter
settings" referenced at L432–433?  If so, it would be good to point out here
that these are "improved" parameters since they are selected to be consistent
with the stratiform conditions specific to this case.

L433–435:  OK, here is a statement about what "conventional" means.
Apparently, "improved" includes the melting layer model.  Are the PSD
parameters given here (D0=1mm, mu=3) for the conventional or improved
settings?  This statement isn't clear, and it's also not clear what is the
basis for selecting the "improved" settings.

L442:  "the PSD parameters for raindrops were based on the assumed value".
This isn't clear because both the "conventional" and "improved" simulations
use "assumed" PSD parameter values.

L447–448:  For both the "conventional" and "improved" cases, aren't there

constraints on the mass-power relation?

L478:  Again, it is unclear what is meant by "conventional" and "improved" settings, but then it is somewhat explained in the following lines, but not clearly.

L510-512:  It is still not clear to me how the mass-diameter relationship affects the shape of the PSD.

L513:  Should be "significant" rather than "significantly".

L513-514:  Revise this to "Variation in a may result in reflectivity uncertainty of approximately 45% for snow and 30% for graupel, mainly due to changes in the particle scattering properties."

L515-516:  Again, I think the approach used to estimate uncertainties due to particle shape significantly underestimates this uncertainty.

L536:  This is the first mention of "multiband measurements".  What is meant by this, and why is it introduced for the first time here?

L537-538:  Similar comment here as above.  It is not clear what is meant by "increasing the polarization function" and how the results of this study support this statement.

---

## Author Response (AR2)

To: Editor, *Atmospheric Measurement Techniques (AMT)*

Re:     Manuscript Number: egusphere-2022-886
Title: Simulation and sensitivity analysis for cloud and precipitation measurements via spaceborne millimeter wave radar
Author: Leilei Kou; Zhengjian Lin; Haiyang Gao; Shujun Liao; Piman Ding

Dear Editor,

Thanks for your attention and the reviewers' evaluation and comments on our manuscript (ID: egusphere-2022-886) once again. We appreciate editor and reviewers very much for the time and effort that they have put into reviewing the manuscript. The suggestions have enabled us to improve our work, as well as guide our research in the future. Based on the revised opinions, we have made targeted revisions to part of the article after discussion. We hope that the revised manuscript is now acceptable for publication in your journal.

\* All the changed contents are highlighted in track change mode in the revised manuscript. More specific revisions against each point are explained as follows.

Thanks very much again for your help to our paper processing.

Best regards
Leilei Kou

**Response to comments by reviewers 1#:**

**I want to thank the authors for their hard work in revising and improving the manuscript. In general, the revised manuscript shows vast improvements, especially in terms of readability, motivation and in clearing up previously ambiguous passages. I stand by considering the study as suitable for AMT. However there are still a few points that should be changed before publishing. Also, one explanation regarding the calculation of mass size parameters depending on rime fraction is puzzling to me. I therefore recommend another round of major revisions.**

Thank you very much for reviewing again as well as your valuable suggestions to improve the paper. We will benefit impressively from your suggestions about writing and technique details. According to the suggestions, I have made careful revisions to this paper. More specific revisions against each point are explained as follows.

**Specific comments**

**● I still find the difference between "conventional" and "optimized" settings to be unclear in the text. Writing "the main difference…" gives the impression that there are other differences that are not mentioned / explained. If that is the case, please write down the other differences. If not, then omit the "main".**

Response: Thanks very much for the valuable suggestion. This expression is inaccurate. We have omitted the "main" in the sentence. Besides, we have restructured the section 4, and the difference between improved and conventional setting was described in the subsection of "experiment design" to make it more clearly.

"4.1.2 Experiment design

  Besides the comparison with the CloudSat observation data, the simulation results with improved and conventional setting were compared as well. For the stratiform case, the PSD parameters were assumed based on the empirical values of land stratiform precipitation clouds (Mason, 1971; Niu and He, 1995; Yin et al., 2011), in which the $D_0$ of cloud water was set to 0.01 mm, the $D_0$ of cloud ice was 0.02 mm, and $\mu$ was set as a constant of 1. As snow in stratiform clouds were mainly unrimed particles in middle and low latitudes (Yin et al., 2017), a mass-power relation representative $m=0.0075D^{2.05}$ of unrimed snow (Moisseev et al., 2017) was used in the simulation, where $D$ was the volume equivalent diameter. During simulation with improved microphysical setting, a melting layer model with a width of 1 km was assumed below 0 ℃ based on the statistical median of melting layer width in stratiform precipitation observed by radars (Liu et al., 2016; Wang et al., 2012), and the PSD parameters of the raindrops were calculated according to the melting model. For conventional setting, the melting model was not included, and the PSD parameters for raindrops were set as $D_0 =1$ mm, $\mu=3$ based on the statistical average values of microphysical parameters of stratiform precipitation in eastern China (Chen et al., 2013; Wen et al., 2019)."

●**It is still not clear to me what mass-size parameter literature values are averaged over. I still recommend including a sentence like, "For a and b we took literature values from -list of studies and calculated the mean" and please go into detail, if you have left out any values that are listed in the literature (for example values for hail).**

Response: Thanks very much for the suggestion. The corresponding sentences have been rewritten. "According to results from observation experiments reported in the literatures, the exponent *b* for snow varies from 1.4 to 2.8, and we derive the mean value of *b* close to 2.1 via averaging literature values of *b* from list of studies (Brandes et al., 2007; Heymsfield et al., 2010; Huang et al., 2019; Sy et al., 2020; Szyrmer and Zawadzki, 2010; Tiira et al., 2016; Wood et al., 2013). For graupel, the exponent *b* varies from 2.1 to 3, and a mean value of approximately 2.6 is derived from the studies in the literatures (Heymsfield et al., 2018; Mason et al., 2018; Von Lerber et al., 2017). Based on the range and mean value of *b* for the Gaussian distribution, we calculated the standard deviation (SD) to be 0.28 and 0.16 for snow and graupel, respectively."

The sentence in Figure 4 has been rewritten as well. "For *a* and *b* we took literature values from list of studies and calculated the mean, and the standard deviation of *b* for snow and graupel are calculated according to the range and average of Gaussian distribution."

●**I don't understand the description of the variable density (variable mass size prefactor a) depending on rime mass fraction. As far as I'm aware the ELWP is the liquid water path along the trajectory of the rimed particle assuming a riming efficiency of 1. It is therefore not equal to the LWP and the equation in L510 (which is the definition of LWP not ELWP as far as I know) is incorrect. In Moissev et al. 2017, ELWP is about two times lower than LWP. If LWP=ELWP is assumed in the study, this must be discussed in more detail. I also don't understand how FR was calculated from ELWP, the formula seems to be missing. It is written that a linear increase of FR with ELWP is assumed and Moissev et al. 2017 is referenced. However, in Moissev Eq. 8 ELWP is proportional to FR/(1-FR) which is not linear. Also the scatter plot in Fig 9 of Moissev et al. 2017 does not show a clear linear behavior between FR and LWP. I would therefore like to ask the authors to clear up these issues and discuss the calculation of the adjustment factor f in more detail. It is fine in my opinion to assume a linear relation between f and LWP, if this is what was done (I am not sure). But this decision must be discussed and possible errors resulting from that approach must be mentioned. In addition, the formula that was used to calculate f should be included (including numerical parameters).**

Response: I am sorry for the mistake. Thanks very much for pointing out. The adjust factor *f* was assumed to increase linearly with LWP. We have revised the relevant content and added more details. "The adjust factor *f* is obtained from *f*=1/(1-*FR*) where FR is the ratio of the rime mass to the snowflake mass. According to Moisseev et al (2017), FR can be expressed as a function of the effective liquid water path (ELWP), $ELWP \approx 4\alpha_u / \pi \cdot FR / (1 - FR)$, given that the rime mass is determined by the mass of swept supercooled liquid droplets. Considering the connection between ELWP and liquid water path LWP (according to Moisseev et al (2017), ELWP is approximately half

of LWP), we assumed that the adjust factor f increased linearly with LWP, and the relation between $f$ and LWP was derived to be $f \approx 0.5\pi LWP / \alpha_u + 1$. The assumption ignores possible changes in particle mass linked to the presence of different crystal habits, and the exponent $b$ in the mass-size relation remains constant. Large uncertainty may occur in the cases where majority of precipitation occurs in the form of crystals."

**Technical corrections (& minor comments)**

**L18 "optimal physical modeling" is misleading**

Response:Thanks for your suggestion. The term of "optimal physical modeling" has been revised as "improved physical modeling".

**L34 Not the best sentence, maybe: "Cloud radars are mainly operated spaceborne, airborne or ground-based."**

Response: Thanks very much for your suggestion. The sentence has been revised as "Cloud radars are mainly operated spaceborne, airborne or ground-based."

**L39 "with a minimum detectable signal of about -30 dBZ"**
**L42 environment → environmental (?)**
**L50 highly important → important**
**L60 satellite simulation**
**L128 was → were (or are)**

Response: Sorry for the poor writing. Thanks for your suggestions. The corresponding expressions and sentences have been modified in the revised manuscript.

**L140 include references, where did you get the knowledge that 10 microns is typical?**

Response: Thanks for pointing out. The references of Mason (1971) and Miles et al (2000) have been added.

**L141 include Mie and Rayleigh citations**

Response: Thanks for your suggestion. Another reference of Bohren and Huffman (1983) has been added. "their scattering characteristics can usually be calculated via Mie theory (Bohren and Huffman, 1983) or Rayleigh approximation (Zhang, 2017) based on the sphere assumption."
Bohren, C.F., and Huffman, D.R.: Absorption and scattering of light by small particles, New York: Wiley, 1983.

**Eq 8 form looks a bit weird, maybe inlcude spacing or dot between number and exp?**

Response: Thanks for the suggestion. We have rewritten Eq.8.

$$D_{max} = \begin{cases} 11\exp(0.069T) & \textit{stratiform} \\ 21\exp(0.070T) & \textit{convective} \end{cases}$$

**L200 I don't think mature is the right term. Also what do you mean by that?**

Response: Thanks for your suggestion. The term of "mature" has been revised as "large".

**L215 & L219 it's misleading that "prefactor a can vary considerably" and "relations vary slightly" are written.**

Response: Sorry for the unclear description. Thanks for pointing out. "relations vary slightly" refers to the mass-power relations in different literatures vary slightly. The prefactor $a$ can vary considerably, and the prefactor $a$ varies between 0.005 and 0.014 cgs units. In different literatures, the prefactor $a$ almost all varies in this range.

The sentence has been rewritten. "In different studies, the statistical results of mass-size relations vary slightly (Brandes et al., 2007; Mason et al., 2018; Tiira et al., 2016; Wood et al., 2015), with the primary difference being the diameter expression for the maximum dimension diameter, $D_m$, median volume diameter, $D_0$, or volume equivalent diameter, $D$."

**L229 why is 0.4 the typical value? Explanation on how that was derived is missing.**

Response: Thanks for your suggestion. The sentence has been rewritten. "The density is generally between 0.2 and 0.9 g/cm$^3$, with the typical value of approximately 0.4 g/cm$^3$ from the statistical results in observation experiments (Heymsfield et al., 2018; Ryzhkov and Zrnic, 2019)."

**L231 graupels → graupel particles**

Response: Thanks for the suggestion. The term has been revised.

**L240 unclear sentence**

Response: Thanks for pointing out. The sentence has been revised. "Similar mass relations can be found for graupel, and its exponent $b$ is larger than that for snow."

**L258 uniform bin size set → I still think that the bin size should be mentioned somewhere earlier**

Response: Thanks for your suggestion. The information about the bin size has been added. "Uniform bin sizes are set for hydrometeors, for example, $dD$ is 0.01 mm for cloud water."

**L282 by the T-matrix**

Response: Thanks for pointing out. The mm$^2$ following the T-matrix has been removed.

**L284 I don't understand. Is the look-up table for backscattering cross-sections of individual particles?**

Response: Yes, the backscattering cross-sections for particles of different diameters are stored in the look-up table.

**L301 change with dominating microphysical processes (?)**

Response: Yes. The term of "microphysical process" has been revised as "dominating microphysical processes".

**L311 do you mean Rayleigh regime?**

Response: Yes, the linear growth stage in the Mie scattering region means the Rayleigh regime.

**L335 leads to**

Response: Thanks for the suggestion. The term of "lead to" has been revised as "leads to".

**L342 "most of the mass relations have the mean value of b close to…" → misleading / unclear; maybe better write: When averaging literature values of the exponent b from – list of studies – we derive b = 2.1 for particles classified as snow.**

**L346 see L342 but for graupel**

Response: Thanks very much for the valuable suggestion. The relevant sentences have been revised. "According to results from observation experiments reported in the literatures, the exponent *b* for snow varies from 1.4 to 2.8, and we derived the mean value of *b* close to 2.1 via averaging literature values of *b* from list of studies (Brandes et al., 2007; Heymsfield et al., 2010; Huang et al., 2019; Sy et al., 2020; Szyrmer and Zawadzki, 2010; Tiira et al., 2016; Wood et al., 2013). For graupel, the exponent *b* varies from 2.1 to 3, and a mean value of approximately 2.6 was derived from the studies in the literatures (Heymsfield et al., 2018; Mason et al., 2018; Von Lerber et al., 2017)."

**L358 represented as different models → represented by different distributions**

**L369 represents for the**

Response: Thanks for the suggestion. The relevant terms have been revised.

**L395 became apparent (?)**

Response: Yes. The sentence has been revised. "For raindrops, the backscattering difference became apparent after the equivalent diameter was 2 mm."

**L413 Maybe better "We then selected two typical weather cases: …." And maybe include a bit more explanation on why these were chose as reviewer#2 has suggested.**

Response: Thanks very much for your valuable suggestion. More explanation has been added. "The cases were selected by combining historical CloudSat data and typical weather processes observed on the ground."

**L420 reference to appendix?**

Response: Thanks very much for your valuable suggestion. We have added this sentence. "More details about model setup can refer to Appendix A."

**L425 the maximum total water content was at approximately 3 km with ~0.9 g/m³.**

Response: Thank you. The sentence has been revised.

**L429 please include the resolution of ERA5 and MODIS (maybe in brackets?) here**

Response: Thanks very much for your suggestion. The information has been added. "Considering the resolution of ERA5 data (0.25º) and the MODIS scanning track (2330 km), the outermost grid in the WRF simulation data was used for comparison."

**L447 the empirical**

Response: The term has been "the empirical".

**L449 maybe include geographic region. In high latitudes (arctic) riming is observed also in stratiform clouds**

Response: Thanks for your suggestion. The geographic has been added in the sentence. "As snow in stratiform clouds were mainly unrimed particles in middle and low latitudes (Yin et al., 2017)."

**L467 with attenuation → in the attenuated signal ?**

Response: Yes, "with attenuation" refers to the attenuated signal.

**L478 what does mostly mean? Either write xx% was within yy% or omit mostly and write the (higher) uncertainty that holds for all data**

Response: Thanks very much for your valuable suggestion. The term of mostly has been omitted and the sentence has been rewritten. "The relative error ($|Z_{sim} - Z_{obs}| / Z_{obs}$, where $Z_{sim}$ represents the simulated reflectivity and $Z_{obs}$ represents the observations, the units of $Z_{sim}$ and $Z_{obs}$ are converted to $mm^6/m^3$) at each height was within 20 %."

**L503 include "typically" or "commonly" otherwise statement is incorrect.**

Response: Thanks for the suggestion. The term of "commonly" has been added in the sentence. "Unlike stratiform clouds, a large percentage of heavily aggregated and/or rimed snow commonly exist in convective clouds (Yin et al., 2017)."

**L546 nine sub modules → eight sub modules?**

Response: Thanks for pointing out. "nine sub modules" has been revised as "eight sub modules".

**L557 significantly greater than … what?**

Response: Thanks for pointing out. The sentence has been rewritten. "Using the exponential PSD with a power-law mass spectrum for snow and graupel, we found that the effects of prefactor *a* on radar reflectivity were significant."

**L570 is it % or rather percent points?**

Response: It should be percent points. The sentence has been rewritten. "The average relative errors in radar reflectivity profile between the simulation and CloudSat data were within 20 %, which improved by 20–80 percent points compared with the conventional setting."
* * *
Special thanks to the reviewer for his/her good comments.
* * *
**Response to comments by reviewers 2#:**

===========

**1. Summary**

===========

This is a review of the revised version of Kou et al., "Simulation and sensitivity analysis for cloud and precipitation measurements via spaceborne millimeter wave radar". The authors evaluate the sensitivity of a forward model for radar reflectivity to its microphysical input variables. The forward model includes cloud ice and water, melting mixed-phase precipitation, snow, graupel and rain. They then perform comparisons of reflectivities that are forward-modeled for two WRF simulations (one stratiform and one convective event) against CloudSat observations of the same events. Although the authors have provided sufficient responses to most of my original concerns, there are still two substantial issues that have not been addressed adequately.

Issue 1 could be addressed by deferring the particle shape and orientation part of this study to a future, more complete study.

Issue 2 could be addressed by following the revisions I've suggested below. I believe that addressing these issues and that by addressing the remaining comments on this revision of the paper, the paper will be suitable for publication.

We appreciate you very much for the time and effort that you have put into reviewing the manuscript. Special thanks to you for the valuable suggestions about issue 1 and issue 2. According to the comments, I have made careful revisions to this paper. More specific revisions against each point are addressed in the notes below.

================

**2. Main issues**

================

Issue 1: Particle shape sensitivities

The authors responded to my original comment by performing DDA simulations of the scattering properties of their chosen particle shapes (sphere, spheroid, cylinder). The point of my comment wasn't that DDA needed to be applied to these shapes. Instead the point was that more realistic shape variations are needed and that DDA is the method usually used to calculate scattering properties for more realistic shapes. The use of realistic shapes and DDA (or perhaps Raleigh-Gans) to calculate scattering properties is the current standard for evaluating the shape sensitivity for millimeter-wavelength radar reflectivity in snow. I think that the authors cannot claim to be assessing shape sensitivity accurately when using only

**spheres, spheroids and cylinders.**

Response: Thanks very much for your valuable suggestion. Yes, the discussion of shape sensitivity is not sufficient. In this study, we mainly discuss the shape sensitivity of spheres and spheroids, which is not sufficient to evaluate the effects of particle shapes on backscattering properties. In the future research, we will consider more realistic variations in particle shapes to evaluate sensitivity of the scattering properties to hydrometeor shapes more comprehensively.

We have added relevant information in the revised manuscript. "Here we mainly discuss the backscattering difference between spheres and spheroids. In future research, we will consider more realistic variations in particle shapes to evaluate sensitivity of the scattering properties to hydrometeor shapes more comprehensively."

**Issue 2: "Conventional" versus "improved"**

**After going through this version of the paper thoroughly, I still find it difficult to discern what are the conventional and improved assumptions for the two test cases. This needs to be stated more clearly. Part of the problem is that there is no clear layout of the experimental design (this would usually be included in a methods or objectives section, but section 2.1 is the closest we have to this).**

**I would suggest:**

**a) Add a paragraph just after the first paragraph in section 4. The new paragraph should describe the authors' intentions to test the forward model simulations using both conventional and improved parameter settings and briefly describe in general terms the objective of using the conventional and improved settings.**

**b) Add a section just before section 4.1.2 that contains an outline of what is being tested for the stratiform case. Describe what parameters are changed between the conventional and improved radar simulations for this case and the scientific justification for those parameter changes. Then proceed to describe the radar reflectivity simulation results.**

**c) Do the same thing in section 4.2 for the convective case. Also, structure section 4.2 similar to the way section 4.1 is structured: A subsection for the WRF simulation description, a subsection for the experiment design (describing the conventional and improved parameter settings), and a subsection for the results.**

Response: Thank you very much for the valuable suggestion. The section 4 has been restructured. An example of the stratiform case is presented below. The structure of the convective case is similar to this.

"**4 Simulation results for typical cases**

Based on the sensitivity analysis of typical cloud physical parameters, we simulated the radar reflectivity of typical cloud scenes by assuming appropriate physical parameters for different hydrometeors and cloud precipitation types with the hydrometeor mixture ratio from the WRF as input. The simulation results were compared with CloudSat observation data.

Two typical weather cases of a cold front stratiform cloud and a deep convective process were

shown, which were simulated with improved setting accounting for the particle shapes, melting modeling, and mass-power relations for snow and graupel. The cases were selected by combining historical CloudSat data and typical weather processes observed on the ground. For comparison, the results with conventional simulation were also shown.

**4.1 Stratiform case**

**4.1.1 WRF scenario simulation**

**4.1.2 Experiment design**

For comparison with CloudSat data, the two-dimensional (2-D) hydrometeor profile from the WRF model on the track matching CloudSat was selected as the input for the radar reflectivity simulation. The WRF data at 04:30 AM was selected. Owing to the uneven output height layer of the WRF, data for the WRF simulation results were interpolated in the vertical direction. The vertical grid of the interpolated data was 240 m, corresponding to the CloudSat CPR data.

Figure 9a–e shows the latitude-height cross-section of the hydrometeors in the stratiform case simulated by the WRF for cloud water, cloud ice, snow, rain, and the total hydrometeors. The vertical extent of snow is widely distributed, ranging from 3 to 10 km. Rain is mainly below 3 km, with water contents between 0.1 and 0.2 $g/m^3$. At approximately 0 ºC, the water content for cloud water, snow, and rain were large, which led to a high total water content, with a maximum of 0.57 $g/m^3$.

Besides the comparison with the CloudSat observation data, the simulation results with improved and conventional setting were compared as well. For the stratiform case, the PSD parameters were assumed based on the empirical values of land stratiform precipitation clouds (Mason, 1971; Niu and He, 1995; Yin et al., 2011), in which the $D_0$ of cloud water was set to 0.01 mm, the $D_0$ of cloud ice was 0.02 mm, and $\mu$ was set as a constant of 1. As snow in stratiform clouds were mainly unrimed particles in middle and low latitudes (Yin et al., 2017), a mass-power relation representative $m=0.0075D^{2.05}$ of unrimed snow (Moisseev et al., 2017) was used in the simulation, where $D$ was the volume equivalent diameter. During simulation with improved microphysical setting, a melting layer with a width of 1 km was assumed below 0 ºC based on the statistical median of melting layer width in stratiform precipitation observed by radars (Liu et al., 2016; Wang et al., 2012), and the PSD parameters of the raindrops were calculated according to the melting model. For conventional setting, the melting model was not included, and the PSD parameters for raindrops were set as $D_0=1$ mm, $\mu=3$ based on the statistical average values of microphysical parameters of stratiform precipitation in eastern China (Chen et al., 2013; Wen et al., 2019).

**4.1.3 Radar reflectivity simulation results"**

===============================

**3.   Responses to prior comments**

===============================

**These notes provide my assessment of the authors' responses to my original comments (egusphere-2022-886-author_response-version1.pdf)**

**Prior comments, overall**

**1. Thanks for providing these additional details. They are sufficient for explaining the perturbations in b.**

**2. This revision addresses my original comment, thanks. There are some additional comments that apply to these revisions, please see the specific comments section that follows.**

**3. Thanks, this additional text resolves my comment.**

**Prior comments, WRF model simulations**

**Thanks for this response and the details provided in the new Appendix A. This addresses my concern, but please also see the specific comments section that follows.**

**Prior comments, particle shape and orientation**

**This doesn't really address the point of my original comment. The meaning of the original comment is that using soft spheres, spheroids and cylinders doesn't give a realistic representation of how scattering properties for snow particles vary with shape at 94 GHz. This is true regardless of whether the spherical/spheroidal particles' scattering properties are calculated using DDA or T-matrix theory. See for example, Figure 12 and the related discussion in Wood et al. (2015).**

**In order to accurately assess sensitivities of radar reflectivity to particle shape variations, more realistic partical shape variations must be used. And in order to evaluate the scattering properties of more realistic particle shapes, a technique such as DDA must be used. The authors comment: "We mainly considered the difference between sphere and spheroid with with different orientations in this study. In future research, we will consider the influence of more particle shapes on radar reflectivity." I think this is not sufficient to support the authors claims of evaluating particle shape and orientation effects in this study.**

Response: We appreciate you very much for the time and effort that you have put into reviewing the manuscript. The suggestions have enabled us to improve our work, as well as guide our research in the future.

Thanks for your valuable suggestion about evaluating particle shape and orientation. In this study, we mainly consider the shapes of spheres and spheroids, which is not sufficient to evaluate the effects of particle shapes on backscattering properties. In future research, we will consider more realistic variations in particle shapes to evaluate sensitivity of the scattering properties to hydrometeor shapes more comprehensively.

===============================================

**4. Specific comments from review of version**

===============================================

**Note that the ATC document and the version2 paper are not consistent in their revisions. For example, L21 of the ATC uses the phrase "brightness band" while the corresponding line in**

the version 2 paper (also L21) uses the term "bright band". L39 in the ATC gives CloudSat minimum detectable signal of -30dBZ, while L38 of the version2 paper gives -29 dBZ.

Response: Thanks very much for pointing out. This may be related to the manuscript version. We will check the manuscript carefully before submitting.

**L11: Should be "improve" rather than "improving".**

Response: Thank you very much. The term has been revised.

**L22-23: Relative error in the vertical profile of what variable?**

Response: Thanks for pointing out. The sentence has been rewritten. "The average relative error of radar reflectivity in the vertical profile was within 20 %."

**L51-53: To be correct, QuickBeam doesn't compare modeled clouds to observations, it is a radar simulator package. It is up to the users to make the comparisons. Also, QuickBeam is capable of simulating radar reflectivities for radars other than CloudSat. Finally, to say that QuickBeam does not simulate mixed-phase melting particles is entirely incorrect. See section 4 of the Haynes et al. paper you have referenced.**

Response: Sorry for the inaccurate statement. The sentence has been rewritten. "QuickBeam is a user-friendly radar simulation package that converts modeled clouds to the equivalent radar reflectivities measured by a wide range of meteorological radars."

**L57: No citation for WRF-SBM.**

Response: Thanks for pointing out. The citation has been added.

"Iguchi, T., Matsui, T., Shi, J.J., Tao, W.-K., Khain, A.P., Hou, A., Cifelli, R., Heymsfield, A., and Tokay, A.: Numerical analysis using WRF-SBM for the cloud microphysical structures in the C3VP field campaign: impacts of supercooled droplets and resultant riming on snow microphysics, J. Geophys. Res., 117, D23206, doi: 10.1029/2012JD018101."

**L60-61: I am not sure what a "cloud data simulator" is, please clarify. If this is referring to cloud radar simulators, the statement is not correct. QuickBeam, as an example, uses scattering properties obtained from discrete dipole simulations of realistic snow particle shapes from the Liu (2004). It does not use an "equivalent spherical shape" for snow particles.**

Response: Sorry for the inaccurate statement. Thanks very much for pointing out. The sentence about the traditional cloud radar simulator has been removed.

**L82-85: Technically, all of these steps are not part of the "forward modeling". The "forward model" consists only of the component that takes in the simulated cloud and precipitation fields from WRF and outputs the simulated reflectivity profiles. The activities listed here actually compose the entire research method.**

Response: Thanks for pointing out. This includes not only forward modeling, but also sensitivity analysis and results comparison. The sentence has been rewritten. "The framework of forward modeling and simulation for spaceborne millimeter radar was composed of eight sub modules."

**L83: Should be "Weather Research and Forecasting (WRF) model". Also, no citation is provided for the model.**

Response: Thanks for pointing out. "Weather Research and Forecasting (WRF)" has been revised as "Weather Research and Forecasting (WRF) model". The reference of Skamarock et al (2019) has been cited.

**L92: Be a bit more specific here. Which "real observation data"?**

Response: Thanks very much for the suggestion. The sentence has been rewritten. "The WRF simulation results were then validated by using the real satellite and ground observation data such as ground-based radar data."

**L96: Should be "refractive index", not "reflective index".**

Response: Sorry for the mistake. Thanks for pointing out. The term has been revised.

**L99: Need citation for Liebe model.**

Response: Thanks for the suggestion. The reference of Liebe (1981) has been added.

"Liebe, H.J.: Modeling attenuation and phase of radio waves in air at frequencies below 1000 GHz, Radio Sci., 16, 1183-1199, doi: 10.1029/RS016i006p01183, 1981."

**L143: I'm not sure what is meant by "direction of raindrop particles". Please clarify.**

Response: Sorry for the unclear statement. The term of "direction of raindrop particles" has been revised as "particle orientation".

**L162: Can you provide a citation that supports this statement? I don't recall ever seeing an exponential distribution used for cloud ice.**

Response: The reference of Ryzhkov and Zrnic (2019) has been provided.

Ryzhkov, A. V., and Zrnic, D. S.: Radar polarimetry for weather observations, Cham, Switzerland: Springer Press, 2019.

**L177-178: This statement explicitly contradicts the actual findings in Nowell et al, 2013. Nowell et al. find that "the backscatter cross section is not well duplicated by the soft or solid spherical/spheroid approximations" in comparison to DDA results for realistic particles. This quote from Nowell et al. applies to particles with size parameters larger than "x ˜ 0.75", which is true for most snowflakes at 94 GHz. This is the root of my concern raised in my original comments about the need for using more realistic shape and scattering models for snow particles.**

Response: Thanks very much for the good comment. This citation is really inaccurate. The citation of Nowell et al (2013) in this sentence has been removed.

**L204-205: I'm not sure how the comment on graupel altitudes is relevant to this work.**

Response: Yes, this sentence makes no sense. Thanks for pointing out. This sentence has been removed.

**L289: Is this equation reference correct? None of these variables appear in equation 9.**

**L294: Same comment as above for L289.**

Response: Thanks for pointing out. This should be equation (6)-(9), to be exact. The sentences have been rewritten. "For cloud ice, $D_0$ is calculated from Eqs. (6)-(9) given $W$ and $T$; $\mu$ is the only parameter that needs to be assumed." "Based on Eqs. (6)-(9), $D_0$ varied from 0.1–0.5 mm at –60 °C and 0.2–0.8 mm at –20 °C when $W$ ranged from 0 to 0.5 g/m³."

**L300-325: This is a long paragraph and covers several different topics. Perhaps split it into two or three shorter ones.**

Response: Thanks very much for the valuable suggestion. This paragraph has been split into three shorter ones.

**L302: I still object to this use of 'dB'. Using the units 'dB' for this quantity is equivalent to using the units 'mm' for a variable that is measured in meters. It is misleading, confusing, and shouldn't be done in a professional publication.**

Response: The $dBN_0$ has been omitted. The $\log_{10}(N_0)$ has been used in the revised manuscript. Correspondingly, the $dBN_0$ in Figure 4 has been modified to $\log_{10}(N_0)$.

**L306-308: It is not clear how this statement about changes in N0 through natural aggregation processes is relevant to the sensitivity study.**

**L308: What is "among them" referring to? This isn't clear.**

Response: Sorry for the unclear statement. The two sentences are meaningless to the sensitivity analysis. These have been removed. "An increase in $a$ leads to an obvious increase in the corresponding particle scattering properties, and then causes the reflectivity change. Using an average mass-power relation assumption, the variation in $a$ as a result of the degree of aggregation and riming, and particle shapes may result in the reflectivity uncertainty of approximately 45 % and 30 % for snow and graupel, respectively."

**L357-358: I don't think a comparison of reflectivities calculated using sphere and spheroid shapes will adequately evaluate the sensitivitiy of radar reflectivity to snow particle shape. The actual uncertainty at 94 GHz is much larger than 1.6 dB. See for example, Wood et al. (2015) for an evaluation of different aggregate shape assumptions.**

Response: Thanks very much for the comment. The comparison of reflectivities using spheres and spheroids is not sufficient to evaluate the shape effects for snow. In future research, we will consider more realistic variations in snow shapes to evaluate sensitivity of the scattering properties to particle shapes more comprehensively.

**L376: Do you mean "mixing ratio"?**

Response: Yes, it is mixing ratio. The term of "mixture ratio" has been revised as "mixing ratio".

**L390-392: Citations needed for ERA5 and MODIS products.**

Response: Thanks for the suggestion. The citations for ERA5 and MODIS products have been added.

"Menzel, W.P., Frey, R.A., Zhang, H., Wylie, D.P., Moeller, C.C., Holz, R.E., Maddux, B., Baum, B.A., Strabala, K.I., and Gumley, L.E.: MODIS global cloud-top pressure and amount estimation: algorithm description and results, J. Appl. Meteorol. Climatol., 47, 1175-1198, doi: 10.1175/2007JAMC1705.1, 2008.

Hersbach, H., Bell, B., Berrisford, P., Hirahara, S., Horányi, A., Moñoz-Sabater, J., Nicolas, J., Peubey, C., Radu, R., Schepers, D., Simmons, A., Soci, C., Abdalla, S., Abellan, X., Balsamo, G., Bechtold, P., Biavati, G., Bidlot, J., Bonavita, M., Chiara, G.D., Dahlgren, P., Dee, D., Diamantakis, M., Dragani, R., Flemming, J., Forbes, R., Fuentes, M., Geer, A., Haimberger, L., Healy, S., Hogan, R.J., Hólm, E., Janisková, M., Keeley, S., Laloyaux, P., Lopez, P., Lupu, C., Radnoti, G., Rosnay, P., Rozum, I., Vamborg, F., Villaume, S., Thépaut, J-N.: The ERA5 global reanalysis, Q. J. Roy. Meteor. Soc., 146, 1999-2049, doi: 10.1002/qj.3803, 2020."

**L399-400: I don't think it is possible to unequivocally state that the cloud scenario simulation results are valid based solely on evaluations of cloud fraction and cloud top temperature.**

Response: Thanks for the suggestion. This sentence has been removed.

**L414: Are these mass-power parameters the "improved microphysical parameter settings" referenced at L432-433? If so, it would be good to point out here that these are "improved" parameters since they are selected to be consistent with the stratiform conditions specific to this case.**

Response: Sorry for the unclear statement. About improved and conventional microphysical parameter settings, we have added a subsection for the experiment design. Detailed description can be found in the next comment of "L433-435".

**L433-435: OK, here is a statement about what "conventional" means. Apparently, "improved" includes the melting layer model. Are the PSD parameters given here (D0=1mm, mu=3) for the conventional or improved settings? This statement isn't clear, and it's also not clear what is the basis for selecting the "improved" settings.**

Response: Sorry for the unclear statement. Thanks for pointing out. We have added a subsection of the experiment design to describe the conventional and improved parameter settings.

"Besides the comparison with the CloudSat observation data, the simulation results with improved and conventional setting were compared as well. For the stratiform case, the PSD parameters were assumed based on the empirical values of land stratiform precipitation clouds (Mason, 1971; Niu and He, 1995; Yin et al., 2011), in which the $D_0$ of cloud water was set to 0.01 mm, the $D_0$ of cloud ice was 0.02 mm, and $\mu$ was set as a constant of 1. As snow in stratiform clouds were mainly unrimed particles in middle and low latitudes (Yin et al., 2017), a mass-power relation representative $m=0.0075D^{2.05}$ of unrimed snow (Moisseev et al., 2017) was used in the simulation, where $D$ was the volume equivalent diameter. During simulation with improved microphysical setting, a melting layer with a width of 1 km was assumed below 0 ℃ based on the statistical median of melting layer width in stratiform precipitation observed by radars (Liu et al., 2016; Wang et al.,

2012), and the PSD parameters of the raindrops were calculated according to the melting model. For conventional setting, the melting model was not included, and the PSD parameters for raindrops were set as $D_0$ =1 mm, $\mu$=3 based on the statistical average values of microphysical parameters of stratiform precipitation in eastern China (Chen et al., 2013; Wen et al., 2019).

Chen, B.J., Yang, J., and Pu, J.P.: Statistical characteristics of raindrop size distribution in the Meiyu season observed in Eastern China, J. Meteorol. Soc. Jpn, 91, 215-227, doi: 10.2151/jmsj.2013-208, 2013.

Liu, L.P., Zhou, M.: Characteristics of bright band and automatic detection algorithm with vertical pointed Ka band cloud radar, Plateau Meteor., 35, 734-744, doi: 10.7522/j.issn.1000-0534.2014.00160, 2016.

Wang, D.W., Liu, L.P., Zhong, L.Z., Wei, Y.Q., and Wang, X.B.: Analysis of the characters of melting layer of cloud radar data and its identification, Meteor. Mon., 38, 712-721, doi: CNKI:SUN:QXXX.0.2012-06-010, 2012.

Wen, L., Zhao, K., Yang, Z.L., Chen, H.N., Huang, H., Chen, G., and Yang, Z.W.: Microphysics of stratiform and convective precipitation during Meiyu season in eastern China, J. Geophys. Res., 125, e2020JD032677, doi: 10.1029/2020JD032677, 2020.”

**L442: "the PSD parameters for raindrops were based on the assumed value". This isn't clear because both the "conventional" and "improved" simulations use "assumed" PSD parameter values.**

Response: Sorry for the inaccurate statement. Thanks very much for pointing out. The PSD parameters of the raindrops in improved setting were calculated according to the melting model. For conventional setting, the melting model was not included, and the PSD parameters for raindrops were set as $D_0$ =1 mm, $\mu$=3 based on the statistical average values of stratiform precipitation in eastern China (Chen et al., 2013; Wen et al., 2019).

This sentence has been rewritten. “Without the melting model, the PSD parameters for raindrops were based on the assumed fixed value.”

**L447-448: For both the "conventional" and "improved" cases, aren't the reconstraints on the mass-power relation?**

Response: Thanks very much for pointing out. The mass-power relation was used for both "conventional" and "improved" cases. The sentence of “in other words, simulations with mass-power constrained PSD of snow and the melting model are more similar to the observations” has been removed.

**L478: Again, it is unclear what is meant by "conventional" and "improved" settings, but then it is somewhat explained in the following lines, but not clearly.**

Response: Thanks for pointing out. We have restructured the section 4, and the difference between improved and conventional setting was described in the subsection of “experiment design”.

“4.2.2 Experiment design

In the convective case, snow and graupel were abundant. Unlike stratiform clouds, a large percentage of heavily aggregated and/or rimed snow commonly exist in convective clouds (Yin et al., 2017); therefore, rimed particles were assumed for convective clouds modeling. Considering the effect of riming, a varying mass-power relationship was assumed in the simulation with improved setting. As the prefactor $a$ in the mass-power relations increases with the riming degree (Mason et al., 2018; Moisseev et al., 2017; Ryzhkov and Zrnic, 2019), an adjustment factor $f$ was considered in the simulation process, i.e., $a=a_u f$, where $a_u$ is the density prefactor for unrimed snow. $f$ is obtained from $f=1/(1-FR)$ where FR is the ratio of the rime mass to the snowflake mass. According to Moisseev et al (2017), FR can be expressed as a function of the effective liquid water path (ELWP), $ELWP \approx 4\alpha_u / \pi \cdot FR / (1-FR)$, given that the rime mass is determined by the mass of swept supercooled liquid droplets. Considering the connection between ELWP and liquid water path LWP (according to Moisseev et al (2017), ELWP is approximately half of LWP), we assumed that the adjust factor $f$ increased linearly with LWP, and the relation between $f$ and LWP was derived to be $f \approx 0.5\pi LWP / \alpha_u +1$. The assumption ignores possible changes in particle mass linked to the presence of different crystal habits, and the exponent $b$ in the mass-size relation remains constant. Large uncertainty may occur in the cases where majority of precipitation occurs in the form of crystals. The exponent $b$ for snow was assumed to be the mean value of 2.1 based on the sensitivity analysis. Then, the corresponding scattering properties and PSD for snow and graupel were calculated according to the mass-power relations.

The effect of riming was not considered in the conventional simulation. In the simulation with the conventional microphysical setting, a mass-power relation of $m=0.0075D^{2.05}$ of unrimed snow (Moisseev et al., 2017) was used for simulation of snow particles, and a constant density of 0.4 g/cm$^3$ was assumed for graupel particles."

**L510-512: It is still not clear to me how the mass-diameter relationship affects the shape of the PSD.**

Response: Sorry for unclear statement. The statement about the effect on the shape of the PSD in this sentence has been removed. "The mass-diameter relationships for snow and graupel differ substantially for different particle habit types. Using the exponential PSD with a power-law mass spectrum for snow and graupel, we found that the effects of prefactor $a$ on radar reflectivity were significant."

**L513: Should be "significant" rather than "significantly".**

Response: Thanks for the suggestion. The term of "significantly" has been revised as "significant".

**L513-514: Revise this to "Variation in a may result in reflectivity uncertainty of approximately 45% for snow and 30% for graupel, mainly due to changes in the particle scattering properties."**

Response: Thank you very much for the valuable suggestion. The sentence has been revised.

**L515-516: Again, I think the approach used to estimate uncertainties due to particle shape significantly underestimates this uncertainty.**

Response: Thanks for your good comment. This study mainly considers the shapes of sphere and spheroid. More realistic shape variations will be considered in the future research. The sentence has been rewritten. "The assumption of sphere and spheroid could lead to an average reflectivity difference of approximately 4–14 %."

**L536: This is the first mention of "multiband measurements". What is meant by this, and why is it introduced for the first time here?**

Response: This is about the application research prospect. This may not be appropriate at the end of this sentence. The sentence about the "multiband measurements" has been removed.

**L537-538: Similar comment here as above. It is not clear what is meant by "increasing the polarization function" and how the results of this study support this statement.**

Response: Thanks for the comment. This sentence has been removed.
* * *
Special thanks to the reviewer for his/her good comments.

---

## Author Response (AR3)

To: Editor, *Atmospheric Measurement Techniques (AMT)*

Re:      Manuscript Number: egusphere-2022-886
Title: Simulation and sensitivity analysis for cloud and precipitation measurements via spaceborne millimeter wave radar
Author: Leilei Kou; Zhengjian Lin; Haiyang Gao; Shujun Liao; Piman Ding

Dear Editor,

Thanks for your attention on our manuscript (ID: egusphere-2022-886), as well as your valuable suggestions to improve the paper. We appreciate editor and reviewers very much for the time and effort that they have put into reviewing the manuscript. Based on the suggestions, we have added more details about the shape/orientation and dielectric assumption in 2.26. Also, we have checked the typos, equations, figures and tables, authors and their affiliations, and references and citations carefully throughout this manuscript. We hope that the revised manuscript is now acceptable for publication in your journal.

* All the changed contents are highlighted in track change mode in the revised manuscript. More specific revisions against each point are explained as follows.

Thanks very much again for your help to our paper processing.

Best regards
Leilei Kou

**Response to comments:**

**I found one detail lacking for understanding in 2.26 (Melting Modeling). The authors provide much detail on the change in density and particle size are calculated throughout the melting process, but there is no corresponding detail on either the shape/orientation (which are provided for the other particle types) and dielectric assumptions (for example, was the liquid water assumed to be a shell around a solid core, or homogenous mixture with the ice?). Please provide these details for consistency with the other sections on the hydrometeor microphysical modeling.**

Response: Thanks very much for your valuable suggestion. We have added more details about the shape/orientation and dielectric assumption in 2.26 (Melting Modeling).

"Dielectric constant of melting snow depends on snow density and water fraction $f_w$. Here, we use the model that water is considered as background and snow is treated as inclusions, and compute the dielectric constant based on Maxwell-Garnett formulas for the mixture of snow and water (Ryzhkov et al., 2011; Zhang, 2017).

The scattering characteristics of melting particles are still calculated by T-matrix. It is assumed that the shape of melted ice particles gradually changes with the increase of mass water fraction $f_w$, so as to finally obtain the shape of raindrops with the same mass. We can introduce the axis ratio ($\gamma_{ms}$) relationship and the relationship of SD of the canting angle ($\delta_{mr}$) for melting particles as (Ryzhkov and Zrnic, 2019):

$$\begin{aligned} \gamma_{ms} &= \gamma_s + f_w(\gamma_w - \gamma_s) \\ \delta_{ms} &= \delta_s + f_w(\delta_r - \delta_s) \end{aligned} \quad , \tag{20}$$

where $\gamma_s$ is the axis ratio of dry snow, $\gamma_w$ is the axis ratio of raindrop of diameter which is produced as a result of snow melting, $\delta_r$ is the SD of the canting angle distribution of raindrops, whereas $\delta_s$ is the corresponding SD of the distribution of dry snow."

Besides, we have checked the typos, equations, figures and tables, authors and their affiliations, and references and citations carefully throughout this manuscript. Several typos have been modified in the revised manuscript. The changes are shown in the marked-up manuscript version.